# Flatness-Aware Stochastic Gradient Langevin Dynamics

Stefano Bruno [* 1]  Youngsik Hwang [* 2]  Jaehyeon An [3]  Sotirios Sabanis [4 5 6]  Dong-Young Lim [† 2 3]

## Abstract

Flatness of the loss landscape has been widely studied as an important perspective for understanding the behavior and generalization of deep learning algorithms. Motivated by this view, we propose Flatness-Aware Stochastic Gradient Langevin Dynamics (fSGLD), a first-order optimization method that biases learning its dynamics toward flat basins while retaining the computational and memory efficiency of SGD and SGLD. We provide a non-asymptotic theoretical analysis showing that fSGLD targets a flatness-biased Gibbs distribution under a theoretically prescribed coupling between the noise scale $\sigma$ and the inverse temperature $\beta$, together with explicit excess risk guarantees. We empirically evaluate fSGLD across standard optimizer benchmarks, Bayesian image classification, uncertainty quantification, and out-of-distribution detection, demonstrating consistently strong performance and reliable uncertainty estimates. Additional experiments confirm the effectiveness of the theoretically prescribed $\beta$–$\sigma$ coupling compared to decoupled choices. The code for all the experiments is available at https://github.com/youngsikhwang/Flatness-aware-SGLD.

## 1. Introduction

A central challenge in deep learning is ensuring that models generalize effectively to unseen data. In high-dimensional and highly non-convex loss landscapes, stochastic optimizers often encounter numerous local minima, each exhibiting significantly different generalization capabilities (He et al., 2019; Jiang et al., 2020; Damian et al., 2021). Among various attempts to characterize "good" solutions, the *flat minima hypothesis* (Hochreiter & Schmidhuber, 1997), which associates low-curvature regions of the loss landscape with superior generalization, has emerged as a prominent research direction. This perspective has motivated a large body of flatness-aware optimizers, most notably Entropy-SGD (Chaudhari et al., 2017), Sharpness-Aware Minimization (SAM) (Foret et al., 2021), and their variants (Xie et al., 2024; Li et al., 2024b; Tahmasebi et al., 2024; Luo et al., 2024; Chen et al., 2024; Kang et al., 2025; Wei et al., 2025; Liu et al., 2022a;b; Du et al., 2022b; Li et al., 2025).

Despite their empirical success, SAM and related methods are inherently local in nature. By exploiting geometric information only within a restricted vicinity of the current iterate, they can struggle to escape sharp basins in multimodal landscapes. Accordingly, theoretical guarantees for such methods are largely limited to local convergence properties (Andriushchenko & Flammarion, 2022; Bartlett et al., 2023; Si & Yun, 2023; Yu et al., 2024; Khanh et al., 2024; Oikonomou & Loizou, 2025; Zhang et al., 2024; Li et al., 2024a) except for a notable exception (Gatmiry et al., 2024).

In contrast, sampling-based optimizers derived from Langevin dynamics provide a theoretically grounded mechanism for global exploration of the parameter space. Formally, consider the overdamped Langevin dynamics governed by the stochastic differential equation (SDE):

$$\mathrm{d}Y_t = -\nabla u(Y_t)\mathrm{d}t + \sqrt{2\beta^{-1}}\mathrm{d}B_t, \qquad (1)$$

which admits a unique invariant (Gibbs) measure $\pi_\beta^{\mathrm{SGLD}}(\theta) \propto \exp(-\beta u(\theta))$, where $\beta > 0$ is the inverse temperature and $(B_t)_{t\geq 0}$ is a $d$-dimensional Brownian motion. As $\beta$ increases, this measure $\pi_\beta^{\mathrm{SGLD}}(\theta)$ concentrates on the global minimizers of $u$, establishing a direct link between Langevin dynamics and global optimization. Building on this property, Stochastic Gradient Langevin Dynamics (SGLD) (Welling & Teh, 2011; Raginsky et al., 2017)

---

[*]Equal contribution [1]UNIST InnoCORE AI-Space Solar Initiative, Ulsan National Institute of Science and Technology (UNIST), Ulsan, 44919, Republic of Korea [2]Artificial Intelligence Graduate School, Ulsan National Institute of Science and Technology (UNIST), Ulsan, 44919, Republic of Korea [3]Department of Industrial Engineering, Ulsan National Institute of Science and Technology (UNIST), Ulsan, 44919, Republic of Korea [4]School of Mathematics, University of Edinburgh, Edinburgh, United Kingdom [5]Department of Mathematics, National Technical University of Athens, Athens, Greece [6]Archimedes, Athena Research and Innovation Centre, Marousi, Greece. Correspondence to: Dong-Young Lim <dlim@unist.ac.kr>.

*Proceedings of the 43rd International Conference on Machine Learning*, Seoul, South Korea. PMLR 306, 2026. Copyright 2026 by the author(s).

applies an Euler–Maruyama discretization of the Langevin SDE, replacing the exact gradient $\nabla u$ with a stochastic gradient. SGLD has attracted considerable attention as an optimization algorithm for nonconvex problems, and under mild regularity conditions a series of works has established non-asymptotic global convergence guarantees (Raginsky et al., 2017; Xu et al., 2018; Majka et al., 2020; Chau et al., 2021; Zhang et al., 2023). However, the invariant measure $\pi_\beta^{\text{SGLD}}$ targeted by SGLD is purely objective-driven and is largely indifferent to landscape geometry. This creates a critical gap: the lack of a systematic framework that simultaneously offers global exploration capabilities and an inductive bias toward low-curvature regions.

To the best of our knowledge, only a few studies have attempted to bridge this gap by biasing MCMC sampling toward flat basins. For example, Entropy-MCMC (Li & Zhang, 2024) leverages local entropy to smooth the posterior and bias sampling toward low-curvature regions. While this represents an important step forward, Entropy-MCMC augments the sampling state space with auxiliary variables to approximate the flat posterior, introducing additional algorithmic complexity. Moreover, its theoretical analysis is primarily established under strongly convex assumptions.

In this work, we introduce Flatness-Aware Stochastic Gradient Langevin Dynamics (fSGLD), a Langevin-based algorithm that enables global exploration of the parameter space and induces an implicit bias toward flat regions of the loss landscape without additional computational or memory overhead. fSGLD is driven by the interplay between two sources of noise: Langevin noise for global exploration, controlled by the inverse temperature $\beta$; and perturbation noise for identifying flat basins via perturbed gradient evaluations, controlled by the perturbation scale $\sigma$. Our key theoretical finding is that, when the inverse temperature $\beta$ and the perturbation scale $\sigma$ are properly coupled, the invariant measure of fSGLD is close to a flatness-biased Gibbs distribution (specifically, one induced by a Hessian-trace regularized objective) that assigns higher probability mass to low-curvature regions of the loss landscape. Moreover, we establish non-asymptotic estimates in Wasserstein distance and an explicit excess-risk bound for the associated Hessian-trace regularized objective. Importantly, this mechanism requires no explicit second-order computations, additional gradient evaluations, or auxiliary state variables. That is, fSGLD matches the computational and memory cost of SGD and SGLD. We empirically evaluate fSGLD in Bayesian inference, uncertainty quantification and out-of-distribution detection, as well as standard optimization settings. Across diverse experiments, fSGLD consistently achieves superior or comparable performance without incurring any additional computational or memory cost. Moreover, in line with our theoretical analysis, we empirically confirm the effect of the proposed $\beta$-$\sigma$ coupling, and observe that fSGLD tends to prefer flat minima.

## 2. Problem Setting and FSGLD Algorithm

**Notation.** Let $(\Omega, \mathcal{F}, \mathbb{P})$ be a fixed probability space. We denote the probability law of a random variable $Z$ by $\mathcal{L}(Z)$. Fix integers $d, m \geq 1$. Let $I_d$ be the identity matrix of dimension $d$. The Euclidean scalar product is denoted by $\langle \cdot, \cdot \rangle$, with $|\cdot|$ standing for the corresponding norm. Let $f : \mathbb{R}^d \to \mathbb{R}$ be a continuously differentiable function, and we denote its gradient by $\nabla f$. For any integer $q \geq 1$, let $\mathcal{P}(\mathbb{R}^q)$ be the set of probability measures on $\mathcal{B}(\mathbb{R}^q)$. For $\mu, \nu \in \mathcal{P}(\mathbb{R}^d)$, let $\mathcal{A}(\mu, \nu)$ denote the set of probability measures $\varrho$ on $\mathcal{B}(\mathbb{R}^{2d})$ such that its respective marginals are $\mu$ and $\nu$. For any $\mu$ and $\nu \in \mathcal{P}(\mathbb{R}^d)$, the Wasserstein distance of order $p \geq 1$ is defined as

$$W_p(\mu, \nu) = \left( \inf_{\varrho \in \mathcal{A}(\mu, \nu)} \int_{\mathbb{R}^d} \int_{\mathbb{R}^d} |x - z|^p \, \mathrm{d}\varrho(x, z) \right)^{\frac{1}{p}}.$$

### 2.1. Curvature-Regularized Objective for Flat Minima

We consider the following nonconvex stochastic optimization problem:

$$\min_{\theta \in \mathbb{R}^d} u(\theta) := \min_{\theta \in \mathbb{R}^d} \mathbb{E}\big[U(\theta, X)\big], \qquad (2)$$

where $u : \mathbb{R}^d \to \mathbb{R}$ is a four times continuously differentiable function, $U : \mathbb{R}^d \times \mathbb{R}^m \to \mathbb{R}$ is a measurable function satisfying $\mathbb{E}[|U(\theta, X)|] < \infty$ for all $\theta \in \mathbb{R}^d$, and $X$ is a random variable with probability law $\mathcal{L}(X)$. In practice, the gradient $\nabla u$ is usually unknown and one only has access to its unbiased estimate, i.e. $\nabla u(\theta) = \mathbb{E}[\nabla U(\theta, X)]$.

The flat minima hypothesis (Hochreiter & Schmidhuber, 1997) suggests that generalization is closely linked to the local geometry of the loss landscape. Among various notions of curvature, the trace of the Hessian provides a natural measure of average curvature and has been widely adopted to characterize flat minima in both empirical and theoretical studies (Ahn et al., 2024; Gatmiry et al., 2023; Liu et al., 2023; Wen et al., 2023a; Wang et al., 2021; Damian et al., 2021; Zhang et al., 2024; Li et al., 2024a). Following these works, we adopt the trace of the Hessian as a curvature-based measure of flatness and define the following *Hessian-trace regularized objective*:

$$v(\theta) := u(\theta) + \frac{\sigma^2}{2} \operatorname{tr}(H(\theta)), \qquad (3)$$

where $\operatorname{tr}(H(\theta))$ is the trace of the Hessian of $u$ evaluated at $\theta$, and $\sigma > 0$ controls the strength of the Hessian-trace regularization. The global minimizers of $v$ represent a trade-off between low loss and low curvature. We refer to these minimizers as the *global flat minima*, i.e., $\arg\min_{\theta \in \mathbb{R}^d} v(\theta)$.

However, formulations that explicitly incorporate such second-order information are often computationally expensive in high-dimensional settings. Furthermore, alternative strategies that seek flat minima without explicit Hessian computation also incur extra computational or memory overhead. For instance, SAM (Foret et al., 2021) relies on a min-max formulation that requires double gradient computations in each iteration. On the other hand, Entropy-SGD and Entropy-MCMC (Chaudhari et al., 2017; Li & Zhang, 2024) augment the parameter space with auxiliary variables, resulting in double memory costs and increased state-space complexity.

## 2.2. fSGLD Algorithm

To efficiently seek the global flat minima defined in (3) without additional gradient computation or memory costs, we propose the fSGLD algorithm. Formally, let $\theta_0$ be an $\mathbb{R}^d$-valued random variable representing the initial value, $(X_k)_{k \in \mathbb{N}}$ be an i.i.d sequence of data, $(\epsilon_k)_{k \in \mathbb{N}}$ be i.i.d copies of the Gaussian perturbation $\epsilon \sim \mathcal{N}(0, \sigma^2 I_d)$, and $(\xi_k)_{k \in \mathbb{N}}$ be an independent sequence of standard $d$-dimensional Gaussian random variables. We assume that $\theta_0$, $(\epsilon_k)_{k \in \mathbb{N}}$, and $(\xi_k)_{k \in \mathbb{N}}$ are all mutually independent. In fSGLD, the perturbation scale is fully determined by the inverse temperature $\beta > 0$:

$$\sigma := \beta^{-\frac{1+\eta}{4}}, \qquad \text{for a fixed } \eta \in (0, 1). \quad (4)$$

Then, the fSGLD algorithm is given by

$$\begin{aligned} \theta_{k+1}^{\text{fSGLD}} = \theta_k^{\text{fSGLD}} &- \lambda \nabla_\theta U(\theta_k^{\text{fSGLD}} + \epsilon_{k+1}, X_{k+1}) \\ &+ \sqrt{2\lambda\beta^{-1}} \xi_{k+1}, \qquad k \in \mathbb{N}, \end{aligned} \quad (5)$$

with $\theta_0^{\text{fSGLD}} := \theta_0$ where $\lambda > 0$ is the step size. We highlight several key properties of the proposed fSGLD algorithm. *First*, fSGLD employs perturbed stochastic gradients $\nabla_\theta U(\theta_k^{\text{fSGLD}} + \epsilon_{k+1}, X_{k+1})$ instead of unbiased stochastic gradient evaluated at the current iterate. *Second*, fSGLD requires only a single gradient computation per iteration and updates only the $d$-dimensional parameter vector without introducing auxiliary variables. *Third*, fSGLD features a structured interaction between two sources of randomness: the parameter perturbation noise scaled by $\sigma$ and the Langevin noise scaled by $\beta^{-1}$. The perturbation scale $\sigma$ is explicitly coupled with the inverse temperature $\beta$ through (4). As a result, fSGLD requires tuning only a single parameter $\beta$, as in standard SGLD. The parameter $\eta \in (0, 1)$ is fixed throughout, set to $\eta = 0.1$ in all experiments, and is introduced solely for technical reasons in the convergence analysis. *Lastly*, fSGLD provably targets a Gibbs distribution corresponding to the Hessian-trace regularized objective (3), which will be rigorously established in Section 3. This behavior is fundamentally different from that of standard SGLD, whose Gibbs measure $\pi_\beta^{\text{SGLD}}$ concentrates on the global minimizers of the original objective $u$.

## 2.3. Why Does fSGLD Induce a Flatness Bias?

This subsection provides an intuitive explanation of how the fSGLD algorithm induces a flatness bias without additional computation, by identifying the intermediate objective implicitly optimized by fSGLD.

The key observation is that the gradient term of fSGLD in (5) corresponds to an unbiased stochastic gradient of a randomized-smoothing surrogate objective. Specifically, let $\epsilon \sim \mathcal{N}(0, \sigma^2 I_d)$ be a Gaussian perturbation independent of the data variable $X$, and define the *randomized-smoothing surrogate objective*

$$g_\epsilon(\theta) := \mathbb{E}\big[u(\theta + \epsilon)\big] = \mathbb{E}\big[\mathbb{E}_X\big[U(\theta + \epsilon, X)\big]\big]. \quad (6)$$

where the outer expectation is taken with respect to the noise $\epsilon$ and $\mathbb{E}_X[\cdot]$ denotes the conditional expectation given $\epsilon$. Then, the gradient of $g_\epsilon$ satisfies

$$\nabla g_\epsilon(\theta) = \mathbb{E}\big[\mathbb{E}_X\big[\nabla_\theta U(\theta + \epsilon, X)\big]\big], \quad (7)$$

which shows that the perturbed stochastic gradient $\nabla_\theta U(\theta + \epsilon, X)$ used in fSGLD is an unbiased estimator of $\nabla g_\epsilon(\theta)$. Consequently, fSGLD can be interpreted as performing standard SGLD on the surrogate objective $g_\epsilon$ rather than directly on the original objective $u$.

This surrogate serves as a tractable way to access curvature information. By Taylor's theorem, we have

$$\begin{aligned} u(\theta + \epsilon) = u(\theta) &+ \nabla u(\theta)^\top \epsilon + \frac{1}{2}\epsilon^\top H(\theta)\epsilon \\ &+ \frac{1}{6}\sum_{i,j,k=1}^d \frac{\partial^3 u}{\partial\theta_i \partial\theta_j \partial\theta_k}(\theta)\, \epsilon_i \epsilon_j \epsilon_k + \mathcal{R}(\theta, \epsilon), \end{aligned}$$

where $\mathcal{R}(\theta, \epsilon)$ is the remainder term. Taking the expectation over $\epsilon \sim \mathcal{N}(0, \sigma^2 I_d)$ yields the key connection:

$$\begin{aligned} g_\epsilon(\theta) &= u(\theta) + \frac{\sigma^2}{2}\text{tr}\,(H(\theta)) + \mathbb{E}[\mathcal{R}(\theta, \epsilon)] \\ &= v(\theta) + \mathbb{E}[\mathcal{R}(\theta, \epsilon)]. \end{aligned} \quad (8)$$

This derivation reveals that fSGLD implicitly minimizes the Hessian-trace regularized objective $v(\theta)$ by optimizing the randomized-smoothing surrogate objective $g_\epsilon$, provided that the expected remainder term $\mathbb{E}[\mathcal{R}(\theta, \epsilon)]$ is negligible.

However, in general high-dimensional nonconvex settings, the higher-order remainder term $\mathbb{E}[\mathcal{R}(\theta, \epsilon)]$ need not be negligible a priori, and can potentially corrupt the desired flatness bias. The coupling between $\sigma$ and $\beta$ in (4) ensures that the contribution of these higher-order terms is controlled. By linking the magnitude of the perturbation to the temperature of the Langevin dynamics, the surrogate objective $g_\epsilon$ becomes an effective intermediate mechanism through which fSGLD recovers Hessian-trace regularization without explicitly computing second-order information.

Finally, fSGLD admits an alternative interpretation as standard SGLD combined with random weight perturbations (RWP). While this viewpoint is useful for intuition, there are important distinctions. In the implementation of RWP, the perturbation scale is treated as an additional tuning parameter. In contrast, fSGLD determines the perturbation scale entirely through the inverse temperature $\beta$, and therefore introduces no extra hyperparameter beyond those of SGLD. Empirically, perturbation-based methods have been used to improve generalization, but rigorous guarantees for RWP remain limited, except for (Ahn et al., 2024), which studies the algorithmic complexity of reaching flat local minima. In this sense, our results complement this line of work by providing a global, non-asymptotic perspective on how RWP shapes optimization landscapes in Langevin-based dynamics.

## 3. Theoretical Results

In this section, we present the main theoretical results that rigorously validate the fSGLD algorithm. We begin by stating the formal assumptions for our analysis. We then prove that the invariant measure of fSGLD is close to an ideal target distribution over flat minima when $\beta$ and $\sigma$ are properly coupled. Building on this, we obtain non-asymptotic $W_1$ error bounds for the fSGLD iterates and excess risk bounds for the Hessian-trace regularized objective.

### 3.1. Assumptions

Our assumptions impose standard conditions on: (i) regularity of the original objective $u$, unbiasedness of the stochastic gradient, and moments of the initial parameters; (ii) a Lipschitz condition on the stochastic gradient; and (iii) a dissipativity condition to ensure the stability of the Langevin dynamics.

**Assumption 3.1** (Regularity of $u$, moments of the initial parameter). We assume that $u : \mathbb{R}^d \to \mathbb{R}$ is a four times continuously differentiable function, that the stochastic gradient $\nabla_\theta U(\theta, \cdot)$ is unbiased, i.e., $\nabla u(\theta) = \mathbb{E}[\nabla_\theta U(\theta, X)]$, and that the initial parameter $\theta_0$ has a finite fourth moment, i.e. $\mathbb{E}[|\theta_0|^4] < \infty$. Furthermore, there exists a compact set $\mathcal{K} \subseteq \mathbb{R}^d$ such that

$$\sup_{\theta \notin \mathcal{K}} \mathbb{E}[\mathcal{R}(\theta, \epsilon)] \leq C_\triangle \sigma^4 d^2, \tag{9}$$

for some $C_\triangle < \infty$ independent of $\sigma \in (0, 1)$.

The last condition (9) of Assumption 3.1 requires that, for large values of $\theta$, the contribution of the fourth-order derivatives of $u$ in the expected remainder term is absorbed by the fourth order moment of the Gaussian perturbation $\epsilon$. This condition is *weaker* than global uniform control ($\sup_{\theta \in \mathbb{R}^d} \mathbb{E}[\mathcal{R}(\theta, \epsilon)] \leq C_\triangle \sigma^4 d^2$) or a bound that holds outside every compact set ($\forall$ compact $\mathcal{K}$, $\sup_{\theta \notin \mathcal{K}} \mathbb{E}[\mathcal{R}(\theta, \epsilon)] \leq$

$C_\triangle \sigma^4 d^2$). Because the notion of flatness already depends on second-order information of $u$ (see, e.g., (3)), it is natural for our theoretical analysis to assume additional higher-order regularity. Such higher-order assumptions are common in implicit regularization analyses (Arora et al., 2022; Wen et al., 2023b), in complexity results for escaping from sharp minima (Ahn et al., 2024), and in recent studies of flat-minima optimization (Zhang et al., 2025). We stress that these higher-order requirements are purely analytical tools: the fSGLD algorithm in (5) itself makes use only of first-order gradient information.

**Assumption 3.2** (Lipschitzness). There exists $\varphi : \mathbb{R}^m \to [1, \infty)$ with $\mathbb{E}[|(1 + |X_0|)\varphi(X_0)|^4] < \infty$, and $L_1, L_2 > 0$ such that, for all $x, x' \in \mathbb{R}^m$ and $\theta, \theta' \in \mathbb{R}^d$,

$$|\nabla_\theta U(\theta, x) - \nabla_{\theta'} U(\theta', x)| \leq L_1 \varphi(x)|\theta - \theta'|,$$

and

$$\begin{aligned} &|\nabla_\theta U(\theta, x) - \nabla_\theta U(\theta, x')| \\ &\leq L_2(\varphi(x) + \varphi(x'))(1 + |\theta|)|x - x'|. \end{aligned}$$

**Assumption 3.3** (Dissipativity). There exist a measurable function (symmetric matrix-valued) function $\bar{A} : \mathbb{R}^m \to \mathbb{R}^{d \times d}$ and a measurable function $\hat{b} : \mathbb{R}^m \to \mathbb{R}$ such that for any $x \in \mathbb{R}^m$, $z \in \mathbb{R}^d$, $\langle z, \bar{A}(x)z \rangle \geq 0$ and for all $\theta \in \mathbb{R}^d$ and $x \in \mathbb{R}^m$,

$$\langle \nabla_\theta U(\theta, x), \theta \rangle \geq \langle \theta, \bar{A}(x)\theta \rangle - \hat{b}(x).$$

The smallest eigenvalue of $\mathbb{E}[\bar{A}(X_0)]$ is a positive real number $\bar{a} > 0$ and $\mathbb{E}[\hat{b}(X_0))] = \bar{b} > 0$.

Note that the dissipativity condition is a standard requirement for analysis of SGLD in the literature; e.g., see (Raginsky et al., 2017; Xu et al., 2018; Deng et al., 2020a;b; 2022; Futami & Fujisawa, 2023). Our version in Assumption 3.3, together with the Lipschitz condition in Assumption 3.2, follows the more general formulation of (Zhang et al., 2023), which allows for dependency on the data $X$. Moreover, several direct consequences of these assumptions, which are useful for our subsequent analysis, are detailed in Appendix C. Remarks C.1 and C.2 shows that introducing the Gaussian perturbation $\epsilon$ modifies the dimension dependence of the relevant constants associated with the overdamped Langevin dynamics (1). In particular, the linear-growth constant for $\nabla g_\epsilon$, which is $O(1)$ for $\nabla u$, becomes $O(1 + \sigma\sqrt{d})$, while the corresponding offset in the dissipativity condition changes from $O(1)$ to $O(1 + \sigma^2 d)$.

### 3.2. Target Gibbs Measure for Global Flat Minima

At the core of our analysis lies an ideal Gibbs distribution associated with the Hessian-trace regularized objective $v$, which biases toward flat regions of the loss landscape. We

denote this target distribution by $\pi_{\beta,\sigma}^\star$:

$$\pi_{\beta,\sigma}^\star(d\theta) \propto \exp(-\beta v(\theta))\, d\theta. \tag{10}$$

Our central question is whether the law of the fSGLD iterates is sufficiently close to this target Gibbs measure $\pi_{\beta,\sigma}^\star$, and if so, how this discrepancy can be quantified. To this end, we first analyze the invariant measure induced by the fSGLD dynamics, denoted by $\pi_\beta^{fSGLD}$, and investigate its relationship to $\pi_{\beta,\sigma}^\star$. For these two Gibbs measures to align, the expected remainder term $\mathbb{E}[\mathcal{R}(\theta,\epsilon)]$ in (8) must be negligible. In high-dimensional nonconvex problems, this is a non-trivial condition, as higher-order terms can be substantial and unpredictable, potentially corrupting the intended regularization effect.

Under the coupling condition in (4), we show that the invariant measure of fSGLD, $\pi_\beta^{fSGLD}$, is close to the target Gibbs measure $\pi_{\beta,\sigma}^\star$ concentrating on global flat minima.

**Proposition 3.4.** *Let Assumptions 3.1, 3.2, and 3.3 hold, and let $\sigma = \beta^{-\frac{1+\eta}{4}}$ for $\eta \in (0,1)$. Then*

$$W_2(\pi_\beta^{fSGLD}, \pi_{\beta,\sigma}^\star) \leq \underline{D},$$

*where $\underline{D} = O(\beta^{-\frac{\eta}{4}}\sqrt{d} + \beta^{-\frac{\eta}{2}}d + \beta^{-\frac{1+\eta}{2}}d^2)$, whose explicit expression is given in (35).*

The proof of Proposition 3.4 is postponed to Appendix D.1. Due to Proposition 3.4, controlling the discrepancy term $\underline{D}$ between the invariant measure of fSGLD, $\pi_\beta^{fSGLD}$ and the target Gibbs measure $\pi_{\beta,\sigma}^\star$ requires increasing $\beta$. Since $\sigma$ is coupled to $\beta$, larger values of $\beta$ simultaneously reduce the perturbation scale and hence the strength of the flatness bias. This highlights a trade-off between approximation accuracy and flatness induction. We emphasize that fSGLD is not designed as a simulated annealing scheme (Pelletier, 1998) and does not seek the limit $\beta \to \infty$. Rather, it operates in a finite-$\beta$ regime, where the prescribed coupling balances these two effects and yields a flatness-biased Gibbs distribution $\pi_{\beta,\sigma}^\star$. We refer to Appendix B for the formal relationship between two Gibbs measures $\pi_\beta^{fSGLD}$ and $\pi_{\beta,\sigma}^\star$ and for the intuition behind the coupling condition (4).

### 3.3. Non-asymptotic error bounds for fSGLD

Our first main result provides non-asymptotic error bounds on the Wasserstein-1 distance between the law of the fSGLD iterates and the target Gibbs measure showing that the overall error can be made arbitrarily small by appropriately choosing the fSGLD algorithm hyperparameters, i.e. $\lambda, k$, and $\beta$. All proofs for the results in this section are provided in Appendix D.2.

**Theorem 3.5.** *Let Assumptions 3.1, 3.2, and 3.3 hold, and let $\sigma = \beta^{-\frac{1+\eta}{4}}$ for $\eta \in (0,1)$. Then, there exist constants*

$\dot{c}, D_1, D_2, D_3, \underline{D} > 0$ *such that, for every $\beta > 0$, for $0 < \lambda \leq \lambda_{max}$ with $\lambda_{max}$ given in (36), and $k \in \mathbb{N}$,*

$$W_1(\mathcal{L}(\theta_k^{fSGLD}), \pi_{\beta,\sigma}^\star) \leq D_1 e^{-\dot{c}\lambda k/2}(1 + \mathbb{E}[|\theta_0|^4])$$
$$+ (D_2 + D_3)\sqrt{\lambda} + \underline{D}, \tag{11}$$

*where $\dot{c}$ is given in Lemma D.9, and*

$$D_1 = O(e^{D_\star(\beta + d + d\beta^{(1-\eta)/2} + d^2\beta^{-\eta} + 1)}(1 + (1 - e^{-\dot{c}})^{-1})),$$

$$D_2 = O\left(1 + \sqrt{d/\beta^{(1+\eta)/2}}\right),$$

$$D_3 = O(e^{D_\star(\beta + d + d\beta^{(1-\eta)/2} + d^2\beta^{-\eta} + 1)}(1 + (1 - e^{-\dot{c}})^{-1})),$$

*with $D_\star > 0$ independent of $d, \beta, k$. The explicit expressions of $D_1$, $D_2$, $D_3$ are given in (61), and $\underline{D}$ is given in (35). Furthermore, let $\beta_{\bar{\delta}}$, $\lambda_{\bar{\delta}}$, $k_{\bar{\delta}}$ be as in (63), (64), and (65) respectively. For any $\bar{\delta} > 0$, if we choose $\beta \geq \beta_{\bar{\delta}}$, $\lambda \leq \lambda_{\bar{\delta}}$, and $k \geq k_{\bar{\delta}}$, then*

$$W_1(\mathcal{L}(\theta_k^{fSGLD}), \pi_{\beta,\sigma}^\star) \leq \bar{\delta}.$$

The bound (11) in Theorem 11 consists of three components. The term $D_1 e^{-\dot{c}\lambda k/2}$ captures exponential mixing of the underlying overdamped Langevin diffusion, with rate $O(\lambda k)$ leading to a step-size exponent 1/2; the term $(D_2 + D_3)\sqrt{\lambda}$ corresponds to the Euler–Maruyama discretization error, which is of order $O(\lambda^{1/2})$, and the term $\underline{D}$, which does not appear in standard SGLD analyses, is the bound for the distance between the invariant measure of fSGLD and the flatness-aware target, and depends on $\beta$, $d$, and $\eta$ (Proposition 3.4). The fSGLD convergence rate is governed by the discretization error $O(\lambda^{1/2})$, matching the best-known non-asymptotic rate for the discretization error in $W_1$ under the comparable Assumptions 3.2 and 3.3, see, e.g., Theorem 2.4 in (Zhang et al., 2023). A high-level overview of the proof strategy is provided at the beginning of Appendix D. The corresponding Wasserstein-2 bound for Theorem 3.5 is given in Corollary D.16.

*Remark* 3.6. Theorem 3.5 recovers the best known non-asymptotic rate for the discretization error in $W_1$ for SGLD under comparable assumptions, see e.g. Theorem 2.4 in (Zhang et al., 2023). Unfortunately, the constants $D_1$ and $D_3$ have exponential dependence on $d$ and $\beta$ due to the coupling arguments of (Eberle et al., 2019), which are standard in the SGLD literature (Chau et al., 2021; Zhang et al., 2023; Deng et al., 2020a;b; Lovas et al., 2023; Lim et al., 2025a; Neufeld et al., 2025), and therefore this behavior does not reflect a limitation of our approach. In this setting, any improvement in the dimension dependence would necessitate substantially strengthening the contraction-rate estimates in Theorem 2.2 of (Eberle et al., 2019).

*Remark* 3.7. Non-asymptotic error bounds for SGLD targeting the Gibbs measure associated with the original objective

$u$ are well studied in the literature (Raginsky et al., 2017; Xu et al., 2018; Zhang et al., 2023; Lim & Sabanis, 2024; Neufeld et al., 2025). In contrast, our analysis addresses a fundamentally different problem: we explicitly define flat solutions through the Hessian-trace regularized objective $v$ and derive non-asymptotic bounds on the distance between the law of the fSGLD iterates and the corresponding flatness-aware Gibbs measure $\pi_{\beta,\sigma}^{\star}$. This requires controlling the discrepancy between the invariant measure of fSGLD and the ideal Gibbs measure $\pi_{\beta,\sigma}^{\star}$, which introduces analytical challenges that do not arise in standard SGLD analyses.

While the previous results provides a theoretical analysis from a sampling perspective, our final result analyzes fSGLD as an optimizer. The following theorem provides a non-asymptotic bound on the expected excess risk with respect to the Hessian-trace regularized objective $v$.

**Theorem 3.8.** *Let Assumption 3.1, 3.2 and 3.3 hold, and let $\sigma = \beta^{-\frac{1+\eta}{4}}$ for $\eta \in (0,1)$. Then, there exist constants $\dot{c}$, $D_1^{\Diamond}, D_2^{\Diamond}, D_3^{\Diamond} > 0$ such that, for every $\beta > 0$, $0 < \lambda \leq \lambda_{max}$ with $\lambda_{max}$ given in (36), $k \in \mathbb{N}$,*

$$\mathbb{E}[v(\theta_k^{fSGLD})] - \inf_{\theta \in \mathbb{R}^d} v(\theta) \qquad (12)$$
$$\leq D_1^{\Diamond} e^{-\dot{c}\lambda k/4} + D_2^{\Diamond}\lambda^{1/4} + D_3^{\Diamond},$$

*where $\dot{c}$ is given in Lemma D.9, and*

$$D_1^{\Diamond} = O(e^{D_{\star}(\beta + d + d\beta^{(1-\eta)/2} + d^2\beta^{-\eta} + 1)}(1 + (1 - e^{-\frac{\dot{c}}{2}})^{-1})),$$

$$D_2^{\Diamond} = O(e^{D_{\star}(\beta + d + d\beta^{(1-\eta)/2} + d^2\beta^{-\eta} + 1)}(1 + (1 - e^{-\frac{\dot{c}}{2}})^{-1})),$$

$$D_3^{\Diamond} = O\left(\frac{d}{\beta}\log\left(D_{\star}\left(\frac{\beta}{d} + \beta^{\frac{1-\eta}{2}} + 1\right)\right) + \frac{d^2}{\beta^{1+\eta}}\right),$$

*with $D_{\star} > 0$ independent of $d$, $\beta$, $k$. The explicit expressions of $D_1^{\Diamond}$ and $D_2^{\Diamond}$ are given in (74), while $D_3^{\Diamond}$ is defined in (79). Moreover, let $\beta_{\delta}$, $\lambda_{\delta}$, $k_{\underline{\delta}}$ be as in (80), (81), and (82) respectively. For any $\underline{\delta} > 0$, if we choose $\beta \geq \beta_{\underline{\delta}}$, $\lambda \leq \lambda_{\underline{\delta}}$, and $k \geq k_{\underline{\delta}}$, then*

$$\mathbb{E}[v(\theta_k^{fSGLD})] - \inf_{\theta \in \mathbb{R}^d} v(\theta) \leq \underline{\delta}.$$

The bound (12) in Theorem 3.8 admits a similar interpretation as the bound (11) in Theorem 11, with the rate governed by a discretization term of order $O(\lambda^{1/4})$ due to use of the Wasserstein-2 bound in Corollary D.16, matching the best-known rate for the discretization error for excess risk bounds for SGLD under similar Assumptions 3.2 and 3.3, see e.g., Corollary 2.8 in (Zhang et al., 2023). This result shows that the iterates $\theta_k^{fSGLD}$ increasingly favor global minimizers of the Hessian-trace regularized objective $v$.

# 4. Numerical Experiments

In this section, we evaluate fSGLD across a range of tasks to assess its behavior from both Bayesian sampling and optimization perspectives. Section 4.1 studies Bayesian image classification under posterior sampling, while Section 4.2 examines predictive uncertainty estimation and out-of-distribution detection. Section 4.3 evaluates generalization performance under noisy labels in standard optimization settings, including both training from scratch and fine-tuning. Section 4.4 investigates the effect of the proposed $\beta$–$\sigma$ coupling through targeted ablation studies, and Section 4.5 analyzes the curvature of the obtained solutions via the Hessian spectrum. Detailed experimental settings and hyperparameter configurations are provided in Appendix E.

## 4.1. Bayesian Image Classification

We first evaluate fSGLD in Bayesian image classification on CIFAR-10 and CIFAR-100 to assess its effectiveness as a posterior sampling method, following the evaluation protocol of Entropy-MCMC (Li & Zhang, 2024). We adopt the same network architecture and training setup as in prior work for a fair comparison. Following their Bayesian marginalization procedure, we construct the predictor by averaging network outputs over multiple parameter snapshots, enabling a unified inference scheme across both samplers and standard optimizers. As shown in Table 1, fSGLD is consistently competitive in classification accuracy on both CIFAR-10 and CIFAR-100, while achieving the lowest NLL across the compared methods, indicating improved probabilistic prediction under Bayesian model averaging.

## 4.2. Uncertainty Quantification and OOD Detection

We evaluate predictive uncertainty estimation and out-of-distribution (OOD) detection. Using models trained on CIFAR-10 and CIFAR-100 in Section 4.1, we first quantify predictive uncertainty via the entropy of the Bayesian predictive distribution (Malinin & Gales, 2018). This uncertainty score is used for OOD detection on SVHN, which is evaluated using AUROC and AUPR. Table 2 shows that fSGLD achieves strong OOD detection performance on both CIFAR-10–SVHN and CIFAR-100–SVHN, attaining the best AUROC and AUPR on CIFAR-10–SVHN and the best AUPR with the second-best AUROC on CIFAR-100–SVHN.

## 4.3. Generalization on Noisy Label Datasets

This section examines the generalization performance of fSGLD in a standard optimization setting under noisy labels, beyond the Bayesian sampling perspective.

**Training from scratch.** We compare the generalization performance of different optimizers when training ResNet models from scratch. Table 3 presents the results across all dataset-architecture combinations. Overall, fSGLD shows highly competitive performance across all benchmarks. In

*Table 1.* Performance comparison on ResNet-18. Results are reported as mean±std over three different random seeds. The best result is **bold** and the second-best is underlined. Results for all methods except fSGLD and ASAM are sourced from Li & Zhang (2024).

| Method | CIFAR10 | | CIFAR100 | |
|---|---|---|---|---|
| | ACC (%) ↑ | NLL ↓ | ACC (%) ↑ | NLL ↓ |
| SGD | $94.87_{\pm0.04}$ | $0.205_{\pm0.015}$ | $76.49_{\pm0.27}$ | $0.935_{\pm0.021}$ |
| Entropy-SGD | $95.11_{\pm0.09}$ | $0.184_{\pm0.020}$ | $77.45_{\pm0.03}$ | $0.895_{\pm0.009}$ |
| SAM | $95.25_{\pm0.12}$ | $0.166_{\pm0.005}$ | $78.41_{\pm0.22}$ | $0.876_{\pm0.007}$ |
| ASAM | $95.34_{\pm0.23}$ | $\underline{0.150}_{\pm0.008}$ | $78.26_{\pm0.30}$ | $\underline{0.814}_{\pm0.008}$ |
| SGLD | $95.47_{\pm0.11}$ | $0.167_{\pm0.011}$ | $\underline{78.79}_{\pm0.35}$ | $0.854_{\pm0.031}$ |
| Entropy-SGLD | $94.46_{\pm0.24}$ | $0.194_{\pm0.020}$ | $77.98_{\pm0.39}$ | $0.897_{\pm0.027}$ |
| Entropy-MCMC | $\underline{95.69}_{\pm0.06}$ | $0.162_{\pm0.002}$ | $\mathbf{79.16}_{\pm0.07}$ | $0.840_{\pm0.004}$ |
| fSGLD | $\mathbf{95.73}_{\pm0.07}$ | $\mathbf{0.144}_{\pm0.001}$ | $78.53_{\pm0.21}$ | $\mathbf{0.810}_{\pm0.011}$ |

*Table 2.* OOD detection on CIFAR-SVHN. The best result is **bold** and the second-best is underlined. Results for all methods except fSGLD and ASAM are sourced from Li & Zhang (2024).

| Method | CIFAR10-SVHN | | CIFAR100-SVHN | |
|---|---|---|---|---|
| | AUROC (%) ↑ | AUPR (%) ↑ | AUROC (%) ↑ | AUPR (%) ↑ |
| SGD | 98.30 | 99.24 | 71.96 | 84.08 |
| Entropy-SGD | $\underline{98.71}$ | $\underline{99.37}$ | 79.15 | 86.92 |
| SAM | 94.23 | 95.67 | 74.56 | 84.61 |
| ASAM | 97.24 | 98.26 | 79.86 | $\underline{87.93}$ |
| SGLD | 97.66 | 98.64 | 72.51 | 83.35 |
| Entropy-SGLD | 90.07 | 91.80 | 71.83 | 82.89 |
| Entropy-MCMC | 98.15 | 99.04 | $\mathbf{81.14}$ | 87.18 |
| fSGLD | $\mathbf{98.91}$ | $\mathbf{99.44}$ | $\underline{80.52}$ | $\mathbf{88.01}$ |

*Table 3.* Performance comparison on ResNet-34 and ResNet-50. Results are reported as mean±std over five different random seeds. Within each model block, the best result is **bold** and the second-best is underlined. WV-1/WV-5 denote Top-1/Top-5 accuracy on WebVision. The wall-clock time per iteration (s/epoch) measured on CIFAR-10N for each model architecture.

| Model | Optimizer | CIFAR-10N | CIFAR-100N | WV-1 | WV-5 | (s/epoch) |
|---|---|---|---|---|---|---|
| ResNet-34 | SGD | $89.31_{\pm0.84}$ | $58.47_{\pm0.20}$ | $71.87_{\pm0.44}$ | $89.33_{\pm0.30}$ | 22.0 |
| | AdamW | $89.25_{\pm0.66}$ | $56.77_{\pm0.47}$ | $68.69_{\pm0.32}$ | $87.01_{\pm0.24}$ | 22.5 |
| | SAM | $\underline{91.53}_{\pm0.22}$ | $59.18_{\pm0.33}$ | $\underline{73.49}_{\pm0.36}$ | $\mathbf{90.32}_{\pm0.31}$ | 41.3 |
| | ASAM | $\mathbf{91.73}_{\pm0.36}$ | $\underline{60.79}_{\pm0.72}$ | $73.46_{\pm0.24}$ | $\underline{90.14}_{\pm0.42}$ | 41.4 |
| | fSGLD | $91.37_{\pm0.43}$ | $\mathbf{61.51}_{\pm0.65}$ | $\mathbf{73.95}_{\pm0.52}$ | $90.03_{\pm0.36}$ | 23.7 |
| ResNet-50 | SGD | $89.41_{\pm0.26}$ | $57.52_{\pm0.17}$ | $71.11_{\pm0.59}$ | $88.31_{\pm0.40}$ | 31.9 |
| | AdamW | $89.26_{\pm0.31}$ | $57.28_{\pm0.90}$ | $69.92_{\pm0.67}$ | $87.97_{\pm0.34}$ | 32.3 |
| | SAM | $\underline{90.88}_{\pm0.49}$ | $59.01_{\pm0.60}$ | $\underline{72.52}_{\pm0.46}$ | $\underline{89.53}_{\pm0.44}$ | 60.7 |
| | ASAM | $\mathbf{91.25}_{\pm0.67}$ | $\underline{60.47}_{\pm0.90}$ | $71.92_{\pm0.67}$ | $88.48_{\pm0.53}$ | 60.9 |
| | fSGLD | $90.86_{\pm0.34}$ | $\mathbf{61.26}_{\pm1.08}$ | $\mathbf{73.54}_{\pm0.51}$ | $\mathbf{90.34}_{\pm0.21}$ | 34.1 |

*Table 4.* Fine-tuning performance comparison on ViT-B/16.

| Model | ViT-B/16 | | |
|---|---|---|---|
| Dataset | CIFAR-10N | CIFAR-100N | (s/epoch) |
| SGD | 94.64 | 71.80 | 343.2 |
| AdamW | 95.57 | 72.30 | 344.5 |
| SAM | $\mathbf{96.75}$ | 74.66 | 656.7 |
| ASAM | 96.25 | $\underline{74.86}$ | 662.5 |
| fSGLD | $\underline{96.45}$ | $\mathbf{75.67}$ | 345.8 |

particular, fSGLD achieves clear performance gains over standard optimizers on CIFAR-100N and WebVision under the Top-1 metric (WV-1), which represent more challenging settings due to higher label noise and a large number of classes.

In terms of computational cost, the wall-clock time per iteration (s/iter) shows that fSGLD has a training speed comparable to standard optimizers like SGD and AdamW. In contrast, SAM and ASAM incur nearly double the computational overhead due to their min-max formulations requiring two gradient evaluations per step. This highlights a key advantage of our method: fSGLD matches or surpasses SAM and ASAM strong performance with a computational budget similar to standard SGD.

**Fine-tuning.** We further evaluate fSGLD in a fine-tuning setting, using a pre-trained ViT-B/16 model on CIFAR-10N and CIFAR-100N. The results are summarized in Table 4. fSGLD outperforms standard optimizers such as SGD and AdamW, and achieves performance competitive with or superior to SAM and ASAM while requiring roughly half the computational overhead.

### 4.4. Effect of the $\beta$-$\sigma$ Coupling

To assess the effect of the proposed $\beta$-$\sigma$ coupling, we conduct controlled experiments in which the coupling is deliberately violated. Specifically, we consider (i) fixing $\beta$ and varying $\sigma$ over a wide range, and (ii) fixing $\sigma$ and varying $\beta$. For each configuration, we compute the implied exponent $\eta$ via the coupling formula $\sigma = \beta^{-(1+\eta)/4}$.

In Figure 1, the primary (bottom) x-axis corresponds to the varied parameter ($\sigma$ or $\beta$), while the secondary (top) x-axis shows the implied exponent $\eta$ computed from the coupling formula. Across both settings, performance is maximized when the implied $\eta$ lies within the theoretical range $(0, 1)$, and degrades outside this regime, supporting the proposed

$\beta$-$\sigma$ coupling. Moreover, performance remains stable across a broad range of $\eta \in (0, 1)$, indicating that fSGLD is insensitive to the precise choice of $\eta$. This supports the use of a fixed value $\eta = 0.1$ in our experiments.

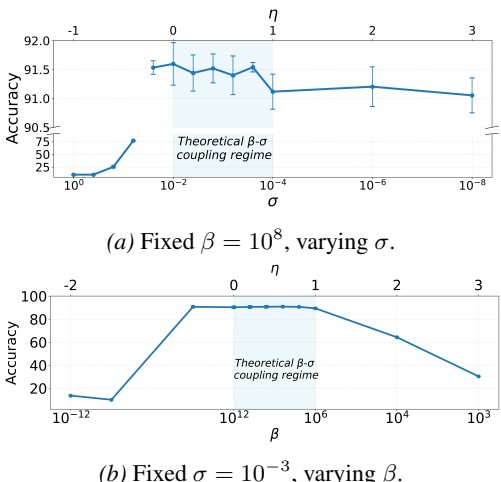

*(a)* Fixed $\beta = 10^8$, varying $\sigma$.

*(b)* Fixed $\sigma = 10^{-3}$, varying $\beta$.

*Figure 1.* Performance of fSGLD as a function of the implied exponent $\eta$, where $\eta$ is computed from the coupling $\sigma = \beta^{-(1+\eta)/4}$. Top: fixing $\beta$ and varying $\sigma$. Bottom: fixing $\sigma$ and varying $\beta$. The bottom x-axis corresponds to the varied parameter ($\sigma$ or $\beta$), while the top x-axis shows the implied exponent $\eta$.

### 4.5. Hessian Spectrum

To empirically verify our theoretical insight that fSGLD finds flat minima by implicitly regularizing the Hessian trace, we analyze the curvature of the solutions obtained by SGD, SGLD, and fSGLD. We measure the maximum Hessian eigenvalue ($\lambda_{\text{top}}$) and the Hessian trace for a ResNet-34 trained on CIFAR-10N, using standard stochastic approximations (Appendix E.5). The results, presented in Figure 2, confirm our hypothesis. fSGLD converges to solutions with a significantly smaller maximum eigenvalue and Hessian trace compared to SGD and SGLD.

## 5. Related Work and Discussions

We review the most relevant literature on SAM, RWP, Hessian-based optimization, and SGLD.

**Flat Minima and Generalization.** Empirical studies (Keskar et al., 2017; Jastrzkebski et al., 2017; Jiang et al., 2020) and theoretical analyses (Dziugaite & Roy, 2017; Neyshabur et al., 2017) consistently show that flatter minima are strongly correlated with better generalization in deep neural networks. However, elucidating precise notions of sharpness and their relationship to generalization remains an open and active area of research (Andriushchenko & Flammarion, 2022; Andriushchenko et al., 2023; Ding et al., 2024; Wen et al., 2023a; Tahmasebi et al., 2024).

**SAM, RWP, and Hessian-regularized Optimizers.** The success of SAM (Foret et al., 2021) has produced a wide range of follow-up work to improve its efficiency, effectiveness, and applicability. Extensions include algorithmic improvements to approximate the inner maximization more efficiently (Liu et al., 2022a; Du et al., 2022a; Kwon et al., 2021; Xie et al., 2024; Li et al., 2024b; Chen et al., 2024; Kang et al., 2025). Beyond these, several Hessian-based regularization approaches have explored flatness from a different angle. For example, (Sankar et al., 2021) studies explicit Hessian-penalty methods that rely on costly approximations. Moreover, (Zhang et al., 2024) proposes Noise-Stability Optimization, and (Li et al., 2024a) studies random weight perturbation with explicit Hessian penalties. Both works focus on PAC-Bayes generalization bounds and local convergence to stationary points, providing algorithm-agnostic guarantees about the perturbed loss rather than the training dynamics of a specific optimizer.

**Flat Posterior Sampling.** Recent Bayesian methods explicitly bias posterior inference toward flat regions of the loss landscape. Flat-seeking Bayesian neural networks (Nguyen et al., 2023) and Flat Posterior-aware Bayesian Model Averaging (Lim et al., 2025b) modify the posterior or inference objective to favor wide, low-curvature minima and improve generalization. Entropy-MCMC (Li & Zhang, 2024) instead uses entropy-regularized MCMC with auxiliary variables to bias sampling toward flat posterior basins.

**SGLD and its Non-asymptotic Analyses.** Following the seminal works of (Welling & Teh, 2011; Raginsky et al., 2017), numerous variants of SGLD have been developed to improve its practical performance, such as variance reduction techniques (Kinoshita & Suzuki, 2022; Dubey et al., 2016; Huang & Becker, 2021), preconditioned SGLD (Li et al., 2016), replica exchange SGLD (Dong & Tong, 2021; Deng et al., 2020a). A parallel line of research has focused on its theoretical properties, particularly its non-asymptotic guarantees. Early results (Raginsky et al., 2017; Xu et al., 2018) showed non-asymptotic error bounds in the Wasserstein-2 distance at a rate dependent on the number of iterations. More recently, the state-of-the-art analyses have established Wasserstein-1 discretization error bounds of order $O(\lambda^{1/2})$ (Zhang et al., 2023). Our discretization error bound matches this state-of-the-art rate. However, a crucial distinction is that prior work shows non-asymptotic error bounds between SGLD iterates and the minimizers of the original objective $u$, whereas our error bounds are between the algorithm iterates and global flat minima. Related to this geometric perspective, (Jules et al., 2023) uses Langevin dynamics with thermal-like noise to analyze the geometry of neural network loss landscapes and the structure of flat regions. While their focus is on landscape analysis rather than optimization, our work complements this perspective by developing an optimization algorithm with explicit guarantees for targeting global flat minima.

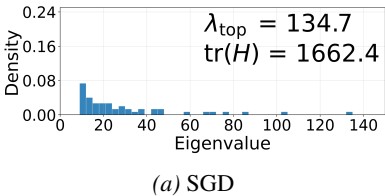 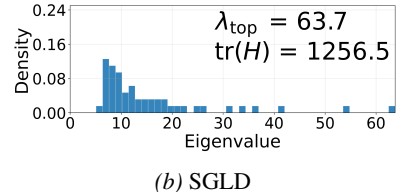 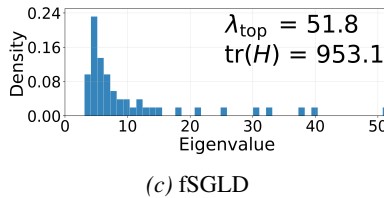

*(a) SGD*      *(b) SGLD*      *(c) fSGLD*

*Figure 2.* The distribution of the leading eigenvalues and Hessian trace of ResNet-34 trained on CIFAR-10N with SGD, SGLD, and fSGLD.

## 6. Conclusion and Limitations

In this work, we proposed Flatness-Aware Stochastic Gradient Langevin Dynamics (fSGLD), a first-order optimization algorithm designed to bias the optimization trajectory toward flat minima without introducing additional memory or extra gradient evaluations. Our main theoretical contribution is a rigorous non-asymptotic analysis of this process. We establish non-asymptotic error bounds in Wasserstein distance and provide the explicit excess risk bound for this class of flatness-aware optimizers. Crucially, our theory shows that the desired regularization effect emerges from a precise coupling of the noise scale $\sigma$ and the inverse temperature $\beta$. Empirically, we conducted an extensive evaluation of fSGLD across a broad range of evaluation protocols and learning settings. We assessed its performance as a standard optimizer in conventional optimization benchmarks, as well as in Bayesian image classification, uncertainty quantification, and out-of-distribution detection. Across these settings, fSGLD demonstrates superior or competitive performance against strong baselines. These gains are achieved at roughly the same gradient evaluation and memory cost as standard SGD and SGLD. We also systematically investigated the effect of the $\beta$-$\sigma$ coupling by varying $\sigma$ and $\beta$ in controlled experiments, clearly demonstrating the effectiveness of the theoretically prescribed coupling compared to decoupled choices of $\beta$ and $\sigma$. Lastly, Hessian spectrum analysis further confirms that fSGLD converges to significantly flatter minima, providing a direct validation of its mechanism.

**Limitations and Future Directions.** Applying fSGLD to diffusion-based generative models (Bruno et al., 2025; Bruno & Sabanis, 2025) is a particularly promising direction; investigating whether its bias towards flatter regions of the loss landscape can lead to more diverse or higher-quality samples is a compelling open question. On the theoretical side, we leave for future work the extension of our analysis to the case where $u$ is semiconvex (i.e., its gradient is one-sided Lipschitz), rather than satisfying Assumption 3.2.

## Acknowledgements

Stefano Bruno was supported by the InnoCORE program of the Ministry of Science and ICT (Grant No. 1.260007.01) and the 2026 Research Fund of UNIST (Grant No. 1.260001.01). Sotirios Sabanis was supported by Innovate UK [grant number 10081810] and was partially supported by project MIS 5154714 of the National Recovery and Resilience Plan Greece 2.0 funded by the European Union through the NextGenerationEU Program. Youngsik Hwang, Jaehyeon An, and Dong-Young Lim were supported by the Ministry of Trade, Industry and Energy (MOTIE) and Korea Institute for Advancement of Technology (KIAT) through the International Cooperative R&D Program (No. P0025828), by the Institute of Information & Communications Technology Planning & Evaluation(IITP) grant by the Korea government(MSIT) (RS-2025-25442824, AI Star Fellowship Program(Ulsan National Institute of Science and Technology)), by the Institute of Information & communications Technology Planning & Evaluation(IITP) grant funded by the Korea government(MSIT) (No.RS-2020-II201336, Artificial Intelligence graduate school support(UNIST)), and by NRF grants funded by MSIT (No. RS-2025-02216640 and NO. RS-2026-25493701). Dong-Young Lim was partially supported by a grant from the Simons Foundation. Part of this work was carried out during his visit to the Isaac Newton Institute.

## Impact Statement

This paper presents work whose goal is to advance the field of Machine Learning. There are many potential societal consequences of our work, none which we feel must be specifically highlighted here.

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

## A. Appendix Roadmap

This appendix is organized as follows:

- Section B establishes the relationship between $\pi_{\beta,\sigma}^\star$ and $\pi_\beta^{\text{fSGLD}}$.

- Section C discusses the direct consequences of our assumptions on the fSGLD dynamics.

- Section D presents an overview of the non-asymptotic error bounds in Wasserstein distance and the expected excess risk bounds.

- Section E reports the implementation details and complements the main empirical results with additional analyses.

## B. Relationship between $\pi_{\beta,\sigma}^\star$ and $\pi_\beta^{\text{fSGLD}}$

We derive the relationship between the target measure $\pi_{\beta,\sigma}^\star$ and the invariant measure $\pi_\beta^{\text{fSGLD}}$ of the fSGLD algorithm, which will be used to prove Proposition 3.4, Theorem 3.5, and Theorem 3.8. By Taylor's theorem, we obtain

$$u(\theta + \epsilon) = u(\theta) + \nabla u(\theta)^T \epsilon + \frac{1}{2}\epsilon^T H(\theta)\epsilon + \frac{1}{6} \sum_{i,j,k=1}^{d} \frac{\partial^3 u}{\partial \theta_i \partial \theta_j \partial \theta_k}(\theta)\, \epsilon_i \epsilon_j \epsilon_k + \mathcal{R}(\theta, \epsilon), \tag{13}$$

where $\mathcal{R}(\theta, \epsilon)$ denotes the remainder term. Taking the expectation over $\epsilon \sim \mathcal{N}(0, \sigma^2 I_d)$ in (13), we have

$$\begin{aligned}
g_\epsilon(\theta) &= u(\theta) + \frac{1}{2}\mathbb{E}[\epsilon^T H(\theta)\epsilon] + \mathbb{E}[\mathcal{R}(\theta, \epsilon)] \\
&= u(\theta) + \frac{1}{2}\text{tr}\left(H(\theta) \cdot \mathbb{E}[\epsilon^T \epsilon]\right) + \mathbb{E}[\mathcal{R}(\theta, \epsilon)] \\
&= v(\theta) + \mathbb{E}[\mathcal{R}(\theta, \epsilon)],
\end{aligned} \tag{14}$$

where

$$v(\theta) = u(\theta) + \frac{\sigma^2}{2}\text{tr}\left(H(\theta)\right).$$

Using Assumption 3.1, in particular $\sigma \in (0, 1)$, and the extreme value theorem on the compact set $\mathcal{K}$, there exists $C_\mathcal{K} < \infty$ such that $\mathbb{E}[\mathcal{R}(\theta, \epsilon)] \leq \sigma^4 d^2 C_\mathcal{K}$ for all $\theta \in \mathcal{K}$. Hence, combining this bound with (9), we obtain

$$\mathbb{E}[\mathcal{R}(\theta, \epsilon)] \leq \sigma^4 d^2 (C_\mathcal{K} + C_\triangle). \tag{15}$$

Let the normalization constants of $\pi_\beta^{\text{fSGLD}}$ and $\pi_{\beta,\sigma}^\star$ be defined as

$$Z_\beta := \int_{\mathbb{R}^d} e^{-\beta g_\epsilon(\theta)}\, \mathrm{d}\theta, \tag{16}$$

and

$$Z_{\beta,\sigma} := \int_{\mathbb{R}^d} e^{-\beta v(\theta)}\, \mathrm{d}\theta. \tag{17}$$

Using (14), (16), and (17), we obtain the following relationship

$$\begin{aligned}
\pi_\beta^{\text{fSGLD}}(\mathrm{d}\theta) &= Z_\beta^{-1}\exp(-\beta g_\epsilon(\theta))\, \mathrm{d}\theta \\
&= Z_\beta^{-1} Z_{\beta,\sigma}\exp(-\beta\, \mathbb{E}[\mathcal{R}(\theta, \epsilon)])\, \pi_{\beta,\sigma}^\star(\mathrm{d}\theta).
\end{aligned} \tag{18}$$

The particular choice $\sigma = \beta^{-\frac{1+\eta}{4}}$ arises naturally from the relation

$$\frac{\pi_{\beta,\sigma}^\star(\mathrm{d}\theta)}{\pi_\beta^{\text{fSGLD}}(\mathrm{d}\theta)} \propto \exp\left(\beta\mathbb{E}[\mathcal{R}(\theta, \epsilon)]\right),$$

and from the contribution of the fourth-order moments of $\epsilon \sim \mathcal{N}(0, \sigma^2 I_d)$ to the expected remainder term, which leads, roughly speaking, to a term of order $\exp\left(O(\beta^{-\eta}d^2)\right)$. As demonstrated in our experiments, this coupling yields meaningful improvements in generalization.

# C. Additional results for Section 3.1

This section collects two technical remarks which are direct consequences of the assumptions presented in Section 3.1.

*Remark* C.1. By Assumption 3.1 and 3.2, the gradient $\nabla u(\theta) = \mathbb{E}[\nabla U(\theta, X)]$ for all $\theta \in \mathbb{R}^d$, is well-defined. In addition, one obtains for all $\theta, \theta' \in \mathbb{R}^d$,

$$|\nabla u(\theta) - \nabla u(\theta')| \leq L_1 \mathbb{E}[\varphi(X_0)]|\theta - \theta'|.$$

The bound of the linear growth of $\nabla u$ is $O(1)$, i.e.

$$|\nabla u(\theta)| \leq L_1 \mathbb{E}[\varphi(X_0)]|\theta| + |\nabla u(0)|.$$

Also, Assumption 3.2 implies, for fixed $\widetilde{\epsilon} \in \mathbb{R}^d$,

$$|\nabla_\theta U(\theta + \widetilde{\epsilon}, x)| \leq L_1 \varphi(x)(|\theta| + |\widetilde{\epsilon}|) + L_2 \bar{\varphi}(x) + |\nabla U(0, 0)|,$$

where $\bar{\varphi}(x) := (\varphi(x) + \varphi(0))|x|$. Using (7) and Lemma 1 in (Nesterov & Spokoiny, 2017), one obtains

$$|\nabla g_\epsilon(\theta)| \leq L_1 \mathbb{E}[\varphi(X_0)] \left(|\theta| + \sigma\sqrt{d}\right) + L_2 \mathbb{E}[\bar{\varphi}(X_0)] + |\nabla U(0, 0)|.$$

Therefore, the introduction of the random weight perturbation $\epsilon$ into the overdamped Langevin dynamics (1) yields the bound of the linear growth of $\nabla g_\epsilon$ to be $O(1 + \sigma\sqrt{d})$.

*Remark* C.2. By Assumption 3.1 and 3.3, one obtains a dissipativity condition of $\nabla u$, i.e., for any $\theta \in \mathbb{R}^d$,

$$\langle \nabla u(\theta), \theta \rangle \geq \bar{a}|\theta|^2 - \bar{b}, \qquad \text{where } \bar{a} = O(1) \text{ and } \bar{b} = O(1). \tag{19}$$

Let $\zeta \in (0, \bar{a}L_1^{-2}(\mathbb{E}[\varphi^2(X_0)])^{-1})$. The inclusion of the random weight perturbation $\epsilon$ into (1) and the use of Assumptions 3.1, 3.2, and 3.3, lead to the following dissipative condition of $\nabla g_\epsilon$, i.e. for any $\theta \in \mathbb{R}^d$,

$$\langle \nabla g_\epsilon(\theta), \theta \rangle \geq a|\theta|^2 - b, \tag{20}$$

where

$$\begin{aligned}
a &:= \bar{a} - \zeta L_1^2 \mathbb{E}[\varphi^2(X_0)] > 0, \\
b &:= ((2\zeta)^{-1} + 2\zeta L_1^2 \mathbb{E}[\varphi^2(X_0)])\sigma^2 d + 4\zeta L_2^2 \mathbb{E}[\bar{\varphi}^2(X_0)] + 4\zeta|\nabla U(0, 0)|^2 + \bar{b} > 0,
\end{aligned} \tag{21}$$

with $\bar{\varphi}$ given in Remark C.1. In particular, the dissipative constants of $\nabla g_\epsilon$ are $a = O(1)$ and $b = O(1 + \sigma^2 d)$.

*Proof of Remark C.2.* Using Assumption 3.3 and Remark C.1, and Young's inequality, one obtains, for fixed $\widetilde{\epsilon} \in \mathbb{R}^d$

$$\begin{aligned}
\langle \nabla_\theta U(\theta + \widetilde{\epsilon}, x), \theta \rangle &= \langle \nabla_\theta U(\theta + \widetilde{\epsilon}, x), \theta + \widetilde{\epsilon} \rangle - \langle \nabla_\theta U(\theta + \widetilde{\epsilon}, x), \widetilde{\epsilon} \rangle \\
&\geq \langle \theta + \widetilde{\epsilon}, A(x)\theta + \widetilde{\epsilon} \rangle - \hat{b}(x) - \zeta 2^{-1}|\nabla_\theta U(\theta + \widetilde{\epsilon}, x)|^2 - (2\zeta)^{-1}|\widetilde{\epsilon}|^2 \\
&\geq \langle \theta, (A(x) - \zeta L_1^2 \varphi^2(x))\theta \rangle + \langle \theta, A(x)\widetilde{\epsilon} \rangle + \langle \widetilde{\epsilon}, A(x)\theta \rangle + \langle \widetilde{\epsilon}, A(x)\widetilde{\epsilon} \rangle \\
&\quad - 2\zeta L_1^2 \varphi^2(x)|\widetilde{\epsilon}|^2 - 4\zeta L_2^2 \bar{\varphi}^2(x) - 4\zeta|\nabla U(0, 0)|^2 - \hat{b}(x) - (2\zeta)^{-1}|\widetilde{\epsilon}|^2.
\end{aligned} \tag{22}$$

Therefore,

$$\begin{aligned}
\nabla g_\epsilon(\theta) &= \mathbb{E}[\mathbb{E}_X[\nabla_\theta U(\theta + \epsilon, X)]] \\
&\geq (\bar{a} - \zeta L_1^2 \mathbb{E}[\varphi^2(X_0)])|\theta|^2 + (\bar{a} - (2\zeta)^{-1} - 2\zeta L_1^2 \mathbb{E}[\varphi^2(X_0)])\sigma^2 d - 4\zeta L_2^2 \mathbb{E}[\bar{\varphi}^2(X_0)] \\
&\quad - 4\zeta|\nabla U(0, 0)|^2 - \bar{b} \\
&\geq a|\theta|^2 - b,
\end{aligned}$$

where $a$ and $b$ are defined in (21). $\qquad\square$

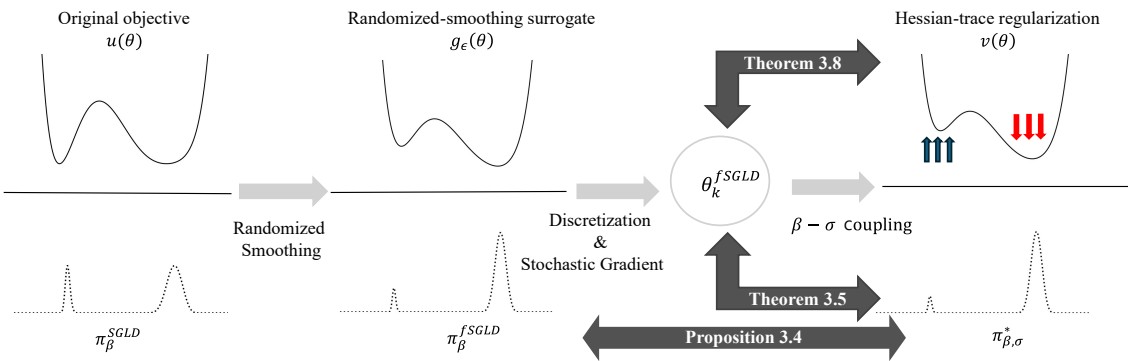

*Figure 3.* A schematic overview of the theoretical framework of fSGLD. The process begins with the **original objective** $u(\theta)$ and its associated Gibbs measure $\pi_\beta^{\text{SGLD}}$ (left). **Randomized smoothing** transforms this into a tractable **surrogate objective**, $g_\epsilon(\theta)$, which is the basis for the fSGLD algorithm and its invariant measure, $\pi_\beta^{\text{fSGLD}}$ (center). This highlights a key distinction: while the Gibbs measure of standard SGLD, $\pi_\beta^{\text{SGLD}}$, is indifferent to the flatness of the minima, the fSGLD framework is designed such that its invariant measure, $\pi_\beta^{\text{fSGLD}}$, targets the distribution over the flattest minima. Our ultimate goal is to target the **Hessian-trace regularized objective** $v(\theta)$ and its corresponding measure $\pi_{\beta,\sigma}^\star$, which concentrates on the desired global flat minima (right).

## D. Overview of the non-asymptotic Wasserstein analysis and error bound for the expected excess risk

In this section, we derive the results introduced in Sections 3.2 and 3.3. To provide a high-level picture of the framework, Figure 3 summarizes the logical flow of our main results.

The proofs of Theorem 3.5 and the resulting $W_2$ bound (Corollary D.16) rely on the following decomposition:

$$W_p(\mathcal{L}(\theta_k^{\text{fSGLD}}), \pi_{\beta,\sigma}^\star) \leq W_p(\mathcal{L}(\theta_k^{\text{fSGLD}}), \mathcal{L}(Y_t^{\lambda,\text{fSGLD}})) + W_p(\mathcal{L}(Y_t^{\lambda,\text{fSGLD}}), \pi_\beta^{\text{fSGLD}}) + W_p(\pi_\beta^{\text{fSGLD}}, \pi_{\beta,\sigma}^\star), \quad (23)$$

for $p = \{1, 2\}$, and $t \in (kT, (k+1)T]$. The first term on the right-hand side of (23) corresponds to the discretization error between the fSGLD recursion (5) and the time-rescaled version of flatness Langevin SDE (38) associated with the randomized-smoothing surrogate objective $g_\epsilon$ defined in (6). The second term captures the convergence error between this SDE and its invariant measure $\pi_\beta^{\text{fSGLD}}$. The third term is the distance between the two measures provided in Proposition 3.4. The proofs of the first two error terms follow the general structure of (Chau et al., 2021; Zhang et al., 2023), but require substantial adaptation to handle the fSGLD update (instead of SGLD) as well as the surrogate objective function $g_\epsilon$ (instead of the original objective $u$). The proof of the third error term is new and is provided in Appendix D.1.

The proof of Theorem 3.8 relies on the $W_2$ bounds of the first and second error terms on the right-hand side of (23), and is given in Appendix D.2.5.

### D.1. Proof of Proposition 3.4

To prove Proposition 3.4, we will use the following result stated in Lemma D.1.

**Lemma D.1.** *Let Assumption 3.3 hold. Then, the following set*

$$\mathfrak{B} := \left\{ \theta \in \mathbb{R}^d : |\theta| \leq \sqrt{\frac{b}{a}} \right\}, \quad (24)$$

*contains all the minimizers of $u(\theta)$ and $g_\epsilon(\theta)$, where $a$ and $b$ are given in (21).*

*Proof of Lemma D.1.* Let $\theta_{g_\epsilon}^\star$ and $\theta_u^\star$ be a minimizer of $g_\epsilon(\theta)$ and $u(\theta)$, respectively. By Remark C.2, we have

$$0 = \langle \nabla u(\theta_u^\star), \theta_u^\star \rangle \geq \overline{a}|\theta_u^\star|^2 - \overline{b}, \quad (25)$$

which implies

$$|\theta_u^\star| \leq \sqrt{\frac{\overline{b}}{\overline{a}}} \leq \sqrt{\frac{b}{a}}.$$

Again, by Remark C.2, we have

$$0 = \langle \nabla g_\epsilon(\theta^\star_{g_\epsilon}), \theta^\star_{g_\epsilon} \rangle \geq a|\theta^\star_{g_\epsilon}|^2 - b, \tag{26}$$

which implies

$$|\theta^\star_{g_\epsilon}| \leq \sqrt{\frac{b}{a}}.$$

$\square$

*Proof of Proposition 3.4.* Recall that for any $\mu$ and $\nu \in \mathcal{P}(\mathbb{R}^d)$, the Kullbak-Leibler divergence (or relative entropy) between $\mu$ and $\nu$ is defined as

$$\mathrm{KL}(\mu||\nu) = \begin{cases} \int_{\mathbb{R}^d} \log\left(\frac{\mathrm{d}\mu}{\mathrm{d}\nu}\right) \mathrm{d}\mu, & \text{if } \mu \ll \nu, \\ \infty, & \text{otherwise.} \end{cases} \tag{27}$$

We use the following result from Corollary 2.3 of (Bolley & Villani, 2005): For any two Borel probability measures $\mu$ and $\nu$ with finite second moments, one obtains

$$W_2(\mu, \nu) \leq C_\nu \left[ \sqrt{\mathrm{KL}(\mu||\nu)} + \left(\frac{\mathrm{KL}(\mu||\nu)}{2}\right)^{1/4} \right], \tag{28}$$

where

$$C_\nu := 2 \inf_{\widetilde{\kappa}>0} \left( \frac{1}{\widetilde{\kappa}} \left( \frac{3}{2} + \log \int_{\mathbb{R}^d} e^{\widetilde{\kappa}|\theta|^2} \nu(\mathrm{d}\theta) \right) \right)^{1/2}. \tag{29}$$

Let $\mu = \pi^{\mathrm{fSGLD}}_\beta$ and $\nu = \pi^\star_{\beta,\sigma}$. Using (18), we have

$$\begin{aligned} \mathrm{KL}(\pi^{\mathrm{fSGLD}}_\beta||\pi^\star_{\beta,\sigma}) &= \int_{\mathbb{R}^d} \log\left(\frac{\pi^{\mathrm{fSGLD}}_\beta(\mathrm{d}\theta)}{\pi^\star_{\beta,\sigma}(\mathrm{d}\theta)}\right) \pi^{\mathrm{fSGLD}}_\beta(\mathrm{d}\theta) \\ &= \int_{\mathbb{R}^d} \log\left(Z^{-1}_\beta Z_{\beta,\sigma} \exp(-\beta\,\mathbb{E}[\mathcal{R}(\theta,\epsilon)])\right) \pi^{\mathrm{fSGLD}}_\beta(\mathrm{d}\theta) \\ &= \log\left(\frac{Z_{\beta,\sigma}}{Z_\beta}\right) - \beta \int_{\mathbb{R}^d} \mathbb{E}[\mathcal{R}(\theta,\epsilon)]\, \pi^{\mathrm{fSGLD}}_\beta(\mathrm{d}\theta). \end{aligned} \tag{30}$$

We focus on the first term on the right-hand side of (30). We denote the complementary set of $\mathcal{K}$ in Assumption 3.1 by $\mathcal{K}^c$. Using (16) and (17), one obtains

$$\begin{aligned} \log\left(\frac{Z_{\beta,\sigma}}{Z_\beta}\right) &= \log\left(\frac{\int_\mathcal{K} e^{-\beta v(\theta)}\mathrm{d}\theta}{\int_\mathcal{K} e^{-\beta g_\epsilon(\theta)}\mathrm{d}\theta + \int_{\mathcal{K}^c} e^{-\beta g_\epsilon(\theta)}\mathrm{d}\theta} + \frac{\int_{\mathcal{K}^c} e^{-\beta v(\theta)}\mathrm{d}\theta}{\int_\mathcal{K} e^{-\beta g_\epsilon(\theta)}\mathrm{d}\theta + \int_{\mathcal{K}^c} e^{-\beta g_\epsilon(\theta)}\mathrm{d}\theta}\right) \\ &\leq \log\left(\frac{\int_\mathcal{K} e^{-\beta g_\epsilon(\theta)+\beta\mathbb{E}[\mathcal{R}(\theta,\epsilon)]}\mathrm{d}\theta}{\int_\mathcal{K} e^{-\beta g_\epsilon(\theta)}\mathrm{d}\theta} + \frac{\int_{\mathcal{K}^c} e^{-\beta g_\epsilon(\theta)+\beta\mathbb{E}[\mathcal{R}(\theta,\epsilon)]}\mathrm{d}\theta}{\int_{\mathcal{K}^c} e^{-\beta g_\epsilon(\theta)}\mathrm{d}\theta}\right) \\ &\leq \log\left(e^{\beta^{-\eta}d^2 C_\mathcal{K}} + e^{\beta^{-\eta}d^2 C_\Delta}\right), \end{aligned} \tag{31}$$

where the second inequality holds by Assumption 3.1, by (15), and by $\sigma = \beta^{-\frac{1+\eta}{4}}$. Using (31) and (15) in (30) yields

$$\begin{aligned} \mathrm{KL}(\pi^{\mathrm{fSGLD}}_\beta||\pi^\star_{\beta,\sigma}) &\leq \log\left(e^{\beta^{-\eta}d^2 C_\mathcal{K}} + e^{\beta^{-\eta}d^2 C_\Delta}\right) + \beta\frac{\int_{\mathbb{R}^d}[|\mathbb{E}[\mathcal{R}(\theta,\epsilon)]|1_\mathcal{K} + |\mathbb{E}[\mathcal{R}(\theta,\epsilon)]|1_{\mathcal{K}^c}]\,e^{-\beta g_\epsilon(\theta)}\,\mathrm{d}\theta}{\int_{\mathbb{R}^d} e^{-\beta g_\epsilon(\theta)}\,\mathrm{d}\theta} \\ &\leq \log\left(e^{\beta^{-\eta}d^2 C_\mathcal{K}} + e^{\beta^{-\eta}d^2 C_\Delta}\right) + \beta^{-\eta}d^2\,(C_\mathcal{K} + C_\Delta) \\ &=: C_1. \end{aligned} \tag{32}$$

Thus, $C_1 = O(\beta^{-\eta} d^2)$. We provide a bound on the constant $C_{\pi^\star_{\beta,\sigma}}$ in (28) using the following inequality. We denote the complementary set of $\mathfrak{B}$ in Lemma D.1 by $\mathfrak{B}^c$. For $\theta \in \mathfrak{B}^c$ and $\widetilde{c} \in (0,1)$, we have, by Remark C.2,

$$
\begin{aligned}
g_\epsilon(\theta) &= g_\epsilon(\widetilde{c}\theta) + \int_{\widetilde{c}}^1 \langle \theta, \nabla g_\epsilon(t\theta) \rangle \, \mathrm{d}t \\
&\geq g_\epsilon(\theta^\star_{g_\epsilon}) + \int_{\widetilde{c}}^1 t^{-1} \langle t\theta, \nabla g_\epsilon(t\theta) \rangle \, \mathrm{d}t \\
&\geq g_\epsilon(\theta^\star_{g_\epsilon}) + \int_{\widetilde{c}}^1 t^{-1} (a|t\theta|^2 - b) \, \mathrm{d}t \\
&\geq \frac{a(1-\widetilde{c}^2)}{2} |\theta|^2 + b \log \widetilde{c} + g_\epsilon(\theta^\star_{g_\epsilon}) \\
&= \bar{c}|\theta|^2 + b \log \widetilde{c} + g_\epsilon(\theta^\star_{g_\epsilon}),
\end{aligned}
\tag{33}
$$

where $\bar{c} := \frac{a(1-\widetilde{c}^2)}{2} > 0$. We use (28) with $\widetilde{\kappa} = \beta\bar{c} - \gamma > 0$, for $\gamma \in (0, 1 \wedge \beta\bar{c})$. Let $c_v := \min_{\theta \in \mathfrak{B}} e^{-\beta v(\theta)}$. Using (33), the inequality $\log(x+y) \leq \log(2) + \max(\log(x), \log(y))$ for $x, y > 0$, the bound (15), and $\sigma = \beta^{-\frac{1+\eta}{4}}$, we obtain

$$
\begin{aligned}
C^2_{\pi^\star_{\beta,\sigma}} &\leq \frac{6}{\beta\bar{c} - \gamma} + \frac{4}{\beta\bar{c} - \gamma} \log \left( \frac{\int_{\mathbb{R}^d} e^{(\beta\bar{c}-\gamma)|\theta|^2 - \beta v(\theta)} \, \mathrm{d}\theta}{\int_{\mathbb{R}^d} e^{-\beta v(\theta)} \, \mathrm{d}\theta} \right) \\
&\leq \frac{6}{\beta\bar{c} - \gamma} + \frac{4}{\beta\bar{c} - \gamma} \log \left( \frac{\int_{\mathfrak{B}} e^{(\beta\bar{c}-\gamma)|\theta|^2 - \beta v(\theta)} \, \mathrm{d}\theta}{\int_{\mathfrak{B}} e^{-\beta v(\theta)} \, \mathrm{d}\theta} + \frac{\int_{\mathfrak{B}^c} e^{(\beta\bar{c}-\gamma)|\theta|^2 - \beta g_\epsilon(\theta) + \beta \mathbb{E}[\mathcal{R}(\theta,\epsilon)]} \, \mathrm{d}\theta}{\int_{\mathfrak{B}} e^{-\beta v(\theta)} \, \mathrm{d}\theta} \right) \\
&\leq \frac{6}{\beta\bar{c} - \gamma} + \frac{4}{\beta\bar{c} - \gamma} \log \left( e^{\frac{(\beta\bar{c}-\gamma)b}{a}} + \frac{e^{-\beta b \log \widetilde{c} - \beta g_\epsilon(\theta^\star_{g_\epsilon})} \int_{\mathfrak{B}^c} e^{-|\theta|^2 \gamma + \beta \mathbb{E}[\mathcal{R}(\theta,\epsilon)](1_{\mathcal{K}^c} + 1_{\mathfrak{B}^c \setminus \mathcal{K}^c})} \, \mathrm{d}\theta}{c_v (\frac{b}{a})^{\frac{d}{2}} \frac{\pi^{d/2}}{\Gamma(d/2+1)}} \right) \\
&\leq \frac{6}{\beta\bar{c} - \gamma} + \frac{4}{\beta\bar{c} - \gamma} \log \left( e^{\frac{(\beta\bar{c}-\gamma)b}{a}} + \frac{e^{-\beta b \log \widetilde{c} - \beta g_\epsilon(\theta^\star_{g_\epsilon}) + \beta \sigma^4 d^2 (C_\Delta + C_\mathcal{K})} a^{\frac{d}{2}} \Gamma(\frac{d}{2} + 1)}{c_v \gamma^{\frac{d}{2}} b^{\frac{d}{2}}} \right) \\
&\leq \frac{6}{\beta\bar{c} - \gamma} + \frac{4\log(2)}{\beta\bar{c} - \gamma} + 4 \max \left( \frac{b}{a}, \frac{\beta b \log \frac{1}{\widetilde{c}}}{\beta\bar{c} - \gamma} - \frac{\beta g_\epsilon(\theta^\star_{g_\epsilon})}{\beta\bar{c} - \gamma} + \frac{\beta^{-\eta} d^2 (C_\Delta + C_\mathcal{K})}{\beta\bar{c} - \gamma} + \frac{d \log(a)}{2(\beta\bar{c} - \gamma)} \right. \\
&\qquad\qquad \left. + \frac{\log(\Gamma(d/2+1))}{\beta\bar{c} - \gamma} - \frac{\log(c_v)}{\beta\bar{c} - \gamma} - \frac{d \log(\gamma b)}{2(\beta\bar{c} - \gamma)} \right) \\
&:= C_2.
\end{aligned}
\tag{34}
$$

By Stirling's formula, we have $\log(\Gamma(d/2+1)) = O(d \log d)$ and by (21), we have $b$ is $O(1 + d\beta^{-\frac{1+\eta}{2}})$. Therefore, $C_2 = O(1 + \beta^{-\frac{1+\eta}{2}} d + d^2 \beta^{-(1+\eta)} + \beta^{-1} d \log(d))$. Applying (28) with (32) and (34), we obtain

$$
\begin{aligned}
&W_2(\pi^{\mathrm{fSGLD}}_\beta, \pi^\star_{\beta,\sigma}) \\
&\leq C_2^{\frac{1}{2}} \left[ \left( \sqrt{C_1} + 2^{-\frac{1}{4}} (C_1)^{1/4} \right) \right] \\
&= \left[ \frac{6}{\beta\bar{c} - \gamma} + \frac{4\log(2)}{\beta\bar{c} - \gamma} + 4 \max \left( \frac{b}{a}, \frac{\beta b \log \frac{1}{\widetilde{c}}}{\beta\bar{c} - \gamma} - \frac{\beta g_\epsilon(\theta^\star_{g_\epsilon})}{\beta\bar{c} - \gamma} + \frac{\beta^{-\eta} d^2 (C_\Delta + C_\mathcal{K})}{\beta\bar{c} - \gamma} + \frac{d \log(a)}{2(\beta\bar{c} - \gamma)} \right.\right. \\
&\qquad\qquad \left.\left. + \frac{\log(\Gamma(d/2+1))}{\beta\bar{c} - \gamma} - \frac{\log(c_v)}{\beta\bar{c} - \gamma} - \frac{d \log(\gamma b)}{2(\beta\bar{c} - \gamma)} \right) \right]^{\frac{1}{2}} \\
&\quad \times \left[ \sqrt{\log \left( e^{\beta^{-\eta} d^2 C_\mathcal{K}} + e^{\beta^{-\eta} d^2 C_\Delta} \right) + \beta^{-\eta} d^2 (C_\mathcal{K} + C_\Delta)} \right. \\
&\qquad\qquad \left. + 2^{-\frac{1}{4}} \left( \log \left( e^{\beta^{-\eta} d^2 C_\mathcal{K}} + e^{\beta^{-\eta} d^2 C_\Delta} \right) + \beta^{-\eta} d^2 (C_\mathcal{K} + C_\Delta) \right)^{\frac{1}{4}} \right] \\
&=: \underline{D},
\end{aligned}
\tag{35}
$$

where $\underline{D} = O(\beta^{-\frac{\eta}{4}}\sqrt{d} + \beta^{-\frac{\eta}{2}}d + \beta^{-\frac{(1+\eta)}{2}}d^2)$. $\qquad\qquad\square$

## D.2. Proof of the results in Section 3.3

We begin by presenting the framework behind this section. The 'data' process $(X_k)_{k\in\mathbb{N}}$ in (5) is adapted to a given filtration $(\mathcal{X}_k)_{k\in\mathbb{N}}$ representing the flow of past information, and we denote the sigma-algebra of $\cup_{k\in\mathbb{N}}\mathcal{X}_k$ by $\mathcal{X}_\infty$. In addition, we assume that $\theta_0$, $\mathcal{X}_\infty$, $(\epsilon_k)_{k\in\mathbb{N}}$, and $(\xi_k)_{k\in\mathbb{N}}$ are all independent among themselves.

We define

$$\lambda_{\max} = \min\left\{ \frac{\min\{a, a^{\frac{1}{3}}\}}{16(1+L_1)^2(\mathbb{E}[(1+\varphi(X_0))^4])^{1/2}}, \frac{1}{a}\right\},\tag{36}$$

where $L_1$, $\varphi$ and $a$ are defined in Assumptions 3.2 and Remark C.2, respectively.

We define the process $(Y_t^{\text{fSGLD}})_{t\geq 0}$ as the solution of the *flatness* Langevin SDE

$$
\begin{aligned}
Y_0^{\text{fSGLD}} &= \theta_0 \in \mathbb{R}^d,\\
\mathrm{d}Y_t^{\text{fSGLD}} &= -\nabla g_\epsilon(Y_t^{\text{fSGLD}})\mathrm{d}t + \sqrt{2\beta^{-1}}\,\mathrm{d}B_t,
\end{aligned}
\tag{37}
$$

where $B_t$ is a standard $d$-dimensional Brownian motion. Denote by $(\mathcal{F}_t)_{t\geq 0}$ the natural filtration of $(B_t)_{t\geq 0}$ and by $\Sigma_{\theta_0}$ the sigma-algebra generated by $\theta_0$, and we assume that $(\mathcal{F}_t)_{t\geq 0}$ is independent of $\mathcal{X}_\infty \vee \Sigma_{\theta_0}$. Furthermore, denote by $\mathcal{F}_\infty$ the sigma-algebra of $\bigcup_{t\geq 0}\mathcal{F}_t$. Since $\nabla g_\epsilon$ is Lipschitz-continuous as a consequence of Assumptions 3.1 and 3.2, the flatness Langevin SDE (37) has a unique solution adapted to $(\mathcal{F}_t)_{t\geq 0}$.

To facilitate the convergence analysis, we consider the time-rescaled version of the process (37). For each $\lambda > 0$, let $Y_t^{\lambda,\text{fSGLD}} := Y_{\lambda t}^{\text{fSGLD}}$, $t \geq 0$ with

$$
\begin{aligned}
Y_0^{\lambda,\text{fSGLD}} &= \theta_0\\
\mathrm{d}Y_t^{\lambda,\text{fSGLD}} &= -\lambda\nabla g_\epsilon(Y_t^{\lambda,\text{fSGLD}})\,\mathrm{d}t + \sqrt{2\lambda\beta^{-1}}\,\mathrm{d}\widehat{B}_t^\lambda,
\end{aligned}
\tag{38}
$$

where $\widehat{B}_t^\lambda := B_{\lambda t}/\sqrt{\lambda}$, $t \geq 0$ is a Brownian motion. The natural filtration of $(\widehat{B}_t^\lambda)_{t\geq 0}$ is denoted by $(\mathcal{F}_t^\lambda)_{t\geq 0}$ with $\mathcal{F}_t^\lambda := \mathcal{F}_{\lambda t}$, $t \in \mathbb{R}_+$ and is independent of $\mathcal{X}_\infty \vee \Sigma_{\theta_0}$. For a positive real number $a$, we denote its integer part by $\lfloor a\rfloor$ and its ceiling by $\lceil a\rceil$. Then, we define the continuous-time interpolation of fSGLD algorithm (5) as the process $(\bar{\theta}_t^{\text{fSGLD}})_{t\geq 0}$ solving

$$
\begin{aligned}
\bar{\theta}_0^{\text{fSGLD}} &= \theta_0,\\
\mathrm{d}\bar{\theta}_t^{\text{fSGLD}} &= -\lambda\nabla_\theta U(\bar{\theta}_{\lfloor t\rfloor}^{\text{fSGLD}} + \epsilon_{\lceil t\rceil}, X_{\lceil t\rceil})\,\mathrm{d}t + \sqrt{2\lambda\beta^{-1}}\mathrm{d}\widehat{B}_t^\lambda.
\end{aligned}
\tag{39}
$$

For the convergence analysis, we work with the continuous-time interpolation (39) instead of the discrete update (5), since both share the same law at the grid points, i.e., $\mathcal{L}(\bar{\theta}_k^{\text{fSGLD}}) = \mathcal{L}(\theta_k^{\text{fSGLD}})$ for all $k \in \mathbb{N}$. Moreover, we consider the following continuous-time process $(\Phi_t^{s,u,\lambda,\text{fSGLD}})_{t\geq s}$ defined as the solution of

$$
\begin{aligned}
\Phi_s^{s,u,\lambda,\text{fSGLD}} &= u \in \mathbb{R}^d\\
\mathrm{d}\Phi_t^{s,u,\lambda,\text{fSGLD}} &= -\lambda\nabla g_\epsilon(\Phi_t^{s,u,\lambda,\text{fSGLD}})\,\mathrm{d}t + \sqrt{2\lambda\beta^{-1}}\,\mathrm{d}\widehat{B}_t^\lambda.
\end{aligned}
\tag{40}
$$

Fix $k \in \mathbb{N}$. For any $t \geq kT$, define the auxiliary process

$$\bar{\Phi}_t^{\lambda,k,\text{fSGLD}} := \Phi_t^{kT,\bar{\theta}_{kT}^{\text{fSGLD}},\lambda,\text{fSGLD}}, \qquad \text{where} \quad T := \lfloor 1/\lambda\rfloor.\tag{41}$$

In other words, $\bar{\Phi}_t^{\lambda,k,\text{fSGLD}}$ in (41) is a process started from the value of the continuous-time interpolation fSGLD process (39) at time $kT$ and run until time $t \geq kT$ with the continuous-time flatness Langevin dynamics.

We use the following triangle inequality to establish the non-asymptotic bounds in Theorem 3.5 and its $W_2$ consequence (Corollary D.16):

$$
\begin{aligned}
W_p(\mathcal{L}(\theta_k^{\text{fSGLD}}), \pi_{\beta,\sigma}^\star) &\leq W_p(\mathcal{L}(\bar{\theta}_t^{\text{fSGLD}}), \mathcal{L}(\bar{\Phi}_t^{\lambda,k,\text{fSGLD}})) + W_p(\mathcal{L}(\bar{\Phi}_t^{\lambda,k,\text{fSGLD}}), \mathcal{L}(Y_t^{\lambda,\text{fSGLD}}))\\
&\quad + W_p(\mathcal{L}(Y_t^{\lambda,\text{fSGLD}}), \pi_\beta^{\text{fSGLD}}) + W_p(\pi_\beta^{\text{fSGLD}}, \pi_{\beta,\sigma}^\star), \qquad \text{with} \quad p \in \{1,2\}.
\end{aligned}
\tag{42}
$$

We control the four terms on the right-hand side of (42) separately. The bounds for the first three terms follow the general methodology of (Chau et al., 2021; Zhang et al., 2023), with Assumption 1 in (Zhang et al., 2023) replaced by Assumption 3.1, and using the bounds for the linear growth and the dissipativity of $\nabla g_\epsilon$ established in Remarks C.1 and C.2, respectively. For completeness, we reproduce these proofs here to make the theoretical analysis of fSGLD self-contained. The bound for the fourth term on the right-hand side of (42) is provided in Proposition 3.4.

### D.2.1. BOUND FOR $W_2(\mathcal{L}(\bar{\theta}_t^{\text{fSGLD}}), \mathcal{L}(\bar{\Phi}_t^{\lambda,k,\text{fSGLD}}))$

We will use the results stated in Lemma D.2 and Lemma D.6 to derive the bound for the first term on the right-hand side of (42) in Lemma D.7.

**Lemma D.2** (Moment bounds of (39)). *Let Assumptions 3.1, 3.2 and 3.3 hold. For any $0 < \lambda \le \lambda_{max}$ given in (36), one obtains*

$$\sup_{t>0} \mathbb{E}[|\bar{\theta}_t^{fSGLD}|^2] \le \mathbb{E}[|\theta_0|^2] + \alpha_1(\lambda_{max} + a^{-1}) < \infty,$$

*where*

$$\begin{aligned}
\alpha_1 &:= 4\lambda_{\max}L_1^2\mathbb{E}\left[\varphi^2(X_0)\right]\sigma^2 d + 4\lambda_{\max}L_2^2\mathbb{E}\left[\bar{\varphi}^2(X_0)\right] + 4\lambda_{\max}|\nabla U(0,0)|^2 + 2b + 2d\beta^{-1} \\
&= O\left(d(\sigma^2 + \beta^{-1}) + 1\right),
\end{aligned} \tag{43}$$

*with $a$ and $b$ given in Remark C.2. In addition, one obtains for $t \in (k, k+1]$ with $k \in \mathbb{N}$,*

$$\mathbb{E}\left[|\bar{\theta}_t^{fSGLD}|^4\right] \le (1 - a\lambda(t-k))(1 - a\lambda)^k \,\mathbb{E}[|\bar{\theta}_0^{fSGLD}|^4] + \alpha_3(\lambda_{max} + a^{-1}),$$

*where*

$$\begin{aligned}
M &:= \max\{(8ba^{-1} + 96a^{-1}\lambda_{\max}(\sigma^2 dL_1^2\mathbb{E}\left[\varphi^2(X_0)\right] + L_2^2\mathbb{E}\left[\bar{\varphi}^2(X_0)\right] + |\nabla_\theta U(0,0)|^2))^{1/2}, \\
&\qquad (256a^{-1}\lambda_{\max}^2(L_2^3\mathbb{E}\left[\bar{\varphi}^3(X_0)\right] + |\nabla_\theta U(0,0)|^3))^{1/3}\} \\
&= O(\sigma\sqrt{d}(\sigma\sqrt{d} \vee 1) + 1), \\
\alpha_2 &:= 4bM^2 + 304(1 + \lambda_{\max})^3 \\
&\qquad \times \left(8(1 + L_1)^4(1 + d(d+2)\sigma^4) + (1 + L_2)^4\mathbb{E}\left[(1 + \bar{\varphi}(X_0))^4\right] + [(1 + |\nabla_\theta U(0,0)|)^4]\right)(1 + M)^2 \\
&= O(\sigma^4 d(\sigma^2 d \vee 1) + 1), \\
\alpha_3 &:= (1 + a\lambda_{max})\alpha_2 + 12d^2\beta^{-2}(\lambda_{max} + 9a^{-1}) \\
&= O(\sigma^4 d(\sigma^2 d \vee 1) + d^2\beta^{-2} + 1).
\end{aligned} \tag{44}$$

*In particular, this implies $\sup_{t>0} \mathbb{E}[|\bar{\theta}_t^{fSGLD}|^4] < \infty$.*

*Proof.* This follows along the same lines as Lemma 4.2 of (Zhang et al., 2023) under our own Assumptions 3.1, 3.2, and 3.3, and using the estimates in Remark C.1 and C.2. □

We define, for each $p \ge 1$, the Lyapunov function $\widetilde{V}_p$ by $\widetilde{V}_p(\theta) := (1 + |\theta|^2)^{p/2}$, $\theta \in \mathbb{R}^d$, and similarly $\widetilde{v}_p(\omega) := (1 + \omega^2)^{p/2}$, for any real $\omega \ge 0$. We establish a drift condition for the flatness Langevin SDE (37), which will be instrumental in deriving moment bounds for the continuous-time process $\bar{\Phi}_t^{\lambda,k,\text{ fSGLD}}$ in Lemma D.11.

**Lemma D.3.** *Let Assumptions 3.1 and 3.3 hold. Then, for each $p \ge 2$, $\theta \in \mathbb{R}^d$,*

$$\Delta\widetilde{V}_p(\theta)\beta^{-1} - \langle\nabla g_\epsilon(\theta), \nabla\widetilde{V}_p(\theta)\rangle \le -\bar{\alpha}(p)\widetilde{V}_p(\theta) + \tilde{\alpha}(p),$$

*where $\bar{\alpha}(p) := ap/4$ and $\tilde{\alpha}(p) := (3/4)ap\,\widetilde{v}_p(\bar{M}_p)$ with $\bar{M}_p := (1/3 + 4b/(3a) + 4d/(3a\beta) + 4(p-2)/(3a\beta))^{1/2}$.*

*Proof.* This follows by replacing the Langevin SDE (1) with the *flatness* Langevin SDE (37) and using Remark C.2 in the arguments of the proof of Lemma 3.5 of (Chau et al., 2021). □

**Lemma D.4.** *Let $\mathcal{F}, \mathcal{X}, \mathcal{H} \subset \mathcal{M}$ be sigma-algebras. Let $X, Y$ be $\mathbb{R}^d$-valued random vectors in $L^2(\Omega)$ such that $Y$ is measurable with respect to $\mathcal{F} \vee \mathcal{X} \vee \mathcal{H}$. Then,*

$$\mathbb{E}^{1/2}\left[\left.|X - \mathbb{E}[X|\mathcal{F} \vee \mathcal{X} \vee \mathcal{H}]|^2\right| \mathcal{X} \vee \mathcal{H}\right] \leq 2\mathbb{E}^{1/2}\left[\left.|X - Y|^2\right| \mathcal{X} \vee \mathcal{H}\right].$$

*Proof.* This follows by applying Lemma 6.1 of (Chau et al., 2019) to $\mathcal{F} \vee \mathcal{N}$, where the sigma-algebra $\mathcal{N} := \mathcal{X} \vee \mathcal{H}$. □

**Lemma D.5.** *Let Assumptions 3.1, 3.2 and 3.3 hold. For any $t \in (kT, (k+1)T]$, with $k, N \in \mathbb{N}$ and $n = 1, \dots, N+1$, where $N + 1 \leq T$, one obtains*

$$\mathbb{E}[|\nabla g_\epsilon(\bar{\Phi}_t^{\lambda,k,\text{fSGLD}}) - \nabla_\theta U(\bar{\Phi}_t^{\lambda,k,\text{fSGLD}} + \epsilon_{kT+n}, X_{kT+n})|^2] \leq e^{-a\lambda t/2}\bar{\psi}_Z \mathbb{E}[\widetilde{V}_2(\theta_0)] + \widetilde{\psi}_Z,$$

*where*

$$
\begin{aligned}
\bar{\psi}_Z &= 16L_2^2\mathbb{E}[(\varphi(X_0) + \varphi(\mathbb{E}[X_0]))^2|X_0 - \mathbb{E}[X_0]|^2] \\
&= O(1), \\
\widetilde{\psi}_Z &= 16L_2^2\mathbb{E}[(\varphi(X_0) + \varphi(\mathbb{E}[X_0]))^2|X_0 - \mathbb{E}[X_0]|^2](3\widetilde{v}_2(\bar{M}_2) + \alpha_1(\lambda_{max} + a^{-1}) + 1) + 8L_1^2\mathbb{E}[\varphi^2(X_0)]\sigma^2 d \\
&= O\left(d(\sigma^2 + \beta^{-1}) + 1\right),
\end{aligned}
\tag{45}
$$

*with $\bar{M}_2$ and $\alpha_1$ given in Lemma D.3 and Lemma D.2, respectively.*

*Proof.* We adapt the proof of Lemma A.1 of (Zhang et al., 2023) to the process (41) and our framework. The perturbation $(\epsilon_k)_{k\in\mathbb{N}}$ in (5) is adapted to a given filtration $(\mathcal{H}_k)_{k\in\mathbb{N}}$, and we denote the sigma-algebra of $\cup_{k\in\mathbb{N}}\mathcal{H}_k$ by $\mathcal{H}_\infty$. Then, we define the filtration $\mathcal{J}_t = \mathcal{F}_\infty^\lambda \vee \mathcal{X}_{\lfloor t \rfloor} \vee \mathcal{H}_{\lfloor t \rfloor}$. Then, the result follows by an application of Lemma D.4, Assumption 3.2, and Lemma D.11

$$
\begin{aligned}
&\mathbb{E}\left[\left|\nabla g_\epsilon(\bar{\Phi}_t^{\lambda,k,\text{fSGLD}}) - \nabla_\theta U(\bar{\Phi}_t^{\lambda,k,\text{fSGLD}} + \epsilon_{kT+n}, X_{kT+n})\right|^2\right] \\
&= \mathbb{E}\left[\mathbb{E}\left[\left.\left|\nabla g_\epsilon(\bar{\Phi}_t^{\lambda,k,\text{fSGLD}}) - \nabla_\theta U(\bar{\Phi}_t^{\lambda,k,\text{fSGLD}} + \epsilon_{kT+n}, X_{kT+n})\right|^2\right| \mathcal{J}_{kT}\right]\right] \\
&= \mathbb{E}\left[\mathbb{E}\left[\left|\mathbb{E}\left[\nabla_\theta U(\bar{\Phi}_t^{\lambda,k,\text{fSGLD}} + \epsilon_{kT+n}, X_{kT+n})\right| \mathcal{J}_{kT}\right]\right.\right. \\
&\qquad\qquad \left.\left.\left. - \nabla_\theta U(\bar{\Phi}_t^{\lambda,k,\text{fSGLD}} + \epsilon_{kT+n}, X_{kT+n})\right|^2\right| \mathcal{J}_{kT}\right]\right] \\
&\leq 4\mathbb{E}\left[\mathbb{E}\left[\left|\nabla_\theta U(\bar{\Phi}_t^{\lambda,k,\text{fSGLD}} + \epsilon_{kT+n}, X_{kT+n})\right.\right.\right. \\
&\qquad\qquad \left.\left.\left. - \nabla_\theta U(\bar{\Phi}_t^{\lambda,k,\text{fSGLD}} + \mathbb{E}\left[\epsilon_{kT+n}| \mathcal{J}_{kT}\right], \mathbb{E}\left[X_{kT+n}| \mathcal{J}_{kT}\right])\right|^2\right| \mathcal{J}_{kT}\right]\right] \\
&\leq 8L_1^2\mathbb{E}[\varphi^2(X_0)]\sigma^2 d + 8L_2^2\mathbb{E}\left[(\varphi(X_0) + \varphi(\mathbb{E}[X_0]))^2|X_0 - \mathbb{E}[X_0]|^2\right]\mathbb{E}\left[\left(1 + \left|\bar{\Phi}_t^{\lambda,k,\text{fSGLD}}\right|^2\right)\right] \\
&\leq 8L_1^2\mathbb{E}[\varphi^2(X_0)]\sigma^2 d + 16L_2^2\mathbb{E}\left[(\varphi(X_0) + \varphi(\mathbb{E}[X_0]))^2|X_0 - \mathbb{E}[X_0]|^2\right] \\
&\qquad \times \left(e^{-\lambda ta/2}\mathbb{E}[\widetilde{V}_2(\theta_0)] + \alpha_1(\lambda_{\max} + a^{-1}) + 3\widetilde{v}_2(\bar{M}_2) + 1\right).
\end{aligned}
$$

□

**Lemma D.6.** *Let Assumptions 3.1, 3.2 and 3.3 hold, and let $\lambda_{max}$ be given in (36). Then, for any $t > 0$,*

$$\mathbb{E}\left[|\bar{\theta}_{\lfloor t \rfloor}^{\text{fSGLD}} - \bar{\theta}_t^{\text{fSGLD}}|^2\right] \leq \lambda\left[e^{-\lambda a\lfloor t \rfloor}\bar{\psi}_Y \mathbb{E}[\widetilde{V}_2(\theta_0)] + \widetilde{\psi}_Y\right],$$

*where*

$$\begin{aligned}
\bar{\psi}_Y &:= 4\lambda_{max}L_1^2\mathbb{E}[\varphi^2(X_0)] \\
&= O(1), \\
\widetilde{\psi}_Y &:= 4\alpha_1 L_1^2\lambda_{max}\mathbb{E}[\varphi^2(X_0)](\lambda_{max} + a^{-1}) + 4\lambda_{max}(L_1^2\sigma^2 d\mathbb{E}[\varphi^2(X_0)] + L_2^2\mathbb{E}[\bar{\varphi}^2(X_0)] + |\nabla_\theta U(0,0)|^2) + 2d\beta^{-1} \\
&= O\left(1 + d(\sigma^2 + \beta^{-1})\right),
\end{aligned} \tag{46}$$

*with $\alpha_1$ given in Lemma D.2.*

*Proof.* This follows by applying Remark C.1 and Lemma D.2 in the proof of Lemma A.2 of (Zhang et al., 2023). $\qquad\square$

We now proceed to bound the first term in (42).

**Lemma D.7.** *Let Assumptions 3.1, 3.2, and 3.3 hold. For any $0 < \lambda < \lambda_{max}$ given in (36), $t \in (kT, (k+1)T]$,*

$$W_2(\mathcal{L}(\bar{\theta}_t^{fSGLD}), \mathcal{L}(\bar{\Phi}_t^{\lambda,k,fSGLD})) \leq \sqrt{\lambda}\left(e^{-ak/4}\bar{D}_{2,1}\mathbb{E}[\widetilde{V}_2(\theta_0)] + \bar{D}_{2,2}\right)^{1/2},$$

*where*

$$\begin{aligned}
\bar{D}_{2,1} &:= 4e^{4L_1^2\mathbb{E}[\varphi^2(X_0)]}(L_1^2\mathbb{E}[\varphi^2(X_0)]\bar{\psi}_Y + \bar{\psi}_Z) \\
&= O(1), \\
\bar{D}_{2,2} &:= 4e^{4L_1^2\mathbb{E}[\varphi^2(X_0)]}(L_1^2\mathbb{E}[\varphi^2(X_0)]\widetilde{\psi}_Y + \widetilde{\psi}_Z) \\
&= O\left(d(\sigma^2 + \beta^{-1}) + 1\right),
\end{aligned} \tag{47}$$

*with $\bar{\psi}_Y$, $\widetilde{\psi}_Y$ given in (46), and $\bar{\psi}_Z$, $\widetilde{\psi}_Z$ given in (45).*

*Proof.* This follows by applying Lemma D.6 together with the argument used in the proof of Lemma 4.7 of (Zhang et al., 2023). We summarize the main steps in the following. Using (39), the process (41), Remark C.1, and it follows that for any $t \in (kT, (k+1)T]$,

$$\begin{aligned}
\left|\bar{\Phi}_t^{\lambda,k,fSGLD} - \bar{\theta}_t^{fSGLD}\right| &\leq \lambda\left|\int_{kT}^t \left[\nabla_\theta U(\bar{\theta}_{\lfloor s\rfloor}^{fSGLD} + \epsilon_{\lceil s\rceil}, X_{\lceil s\rceil}) - \nabla_\theta U(\bar{\Phi}_s^{\lambda,k,fSGLD} + \epsilon_{\lceil s\rceil}, X_{\lceil s\rceil})\right]\mathrm{d}s\right| \\
&\quad + \lambda\left|\int_{kT}^t \left[\nabla g_\epsilon(\bar{\Phi}_s^{\lambda,k,fSGLD}) - \nabla_\theta U(\bar{\Phi}_s^{\lambda,k,fSGLD} + \epsilon_{\lceil s\rceil}, X_{\lceil s\rceil})\right]\mathrm{d}s\right| \\
&\leq \lambda L_1\int_{kT}^t \varphi(X_{\lceil s\rceil})\left|\bar{\theta}_{\lfloor s\rfloor}^{fSGLD} - \bar{\Phi}_s^{\lambda,k,fSGLD}\right|\mathrm{d}s \\
&\quad + \lambda\left|\int_{kT}^t \left[\nabla g_\epsilon(\bar{\Phi}_s^{\lambda,k,fSGLD}) - \nabla_\theta U(\bar{\Phi}_s^{\lambda,k,fSGLD} + \epsilon_{\lceil s\rceil}, X_{\lceil s\rceil})\right]\mathrm{d}s\right|.
\end{aligned} \tag{48}$$

Squaring both sides of (48) and taking expectations, we obtain using $\lambda T \leq 1$ and Lemma D.6

$$\begin{aligned}
\mathbb{E}\left[\left|\bar{\Phi}_t^{\lambda,k,fSGLD} - \bar{\theta}_t^{fSGLD}\right|^2\right] &\leq 4\lambda L_1^2\mathbb{E}\left[\varphi^2(X_0)\right]\int_{kT}^t \mathbb{E}\left[\left|\bar{\theta}_{\lfloor s\rfloor}^{fSGLD} - \bar{\theta}_s^{fSGLD}\right|^2\right]\mathrm{d}s \\
&\quad + 4\lambda L_1^2\mathbb{E}\left[\varphi^2(X_0)\right]\int_{kT}^t \mathbb{E}\left[\left|\bar{\theta}_s^{fSGLD} - \bar{\Phi}_s^{\lambda,k,fSGLD}\right|^2\right]\mathrm{d}s \\
&\quad + 2\lambda^2\mathbb{E}\left[\left|\int_{kT}^t \left[\nabla g_\epsilon(\bar{\Phi}_s^{\lambda,k,fSGLD}) - \nabla_\theta U(\bar{\Phi}_s^{\lambda,k,fSGLD} + \epsilon_{\lceil s\rceil}, X_{\lceil s\rceil})\right]\mathrm{d}s\right|^2\right] \\
&\leq 4\lambda L_1^2\mathbb{E}\left[\varphi^2(X_0)\right](e^{-\lambda akT}\bar{\psi}_Y\mathbb{E}[\widetilde{V}_2(\theta_0)] + \widetilde{\psi}_Y) \\
&\quad + 4\lambda L_1^2\mathbb{E}\left[\varphi^2(X_0)\right]\int_{kT}^t \mathbb{E}\left[\left|\bar{\theta}_s^{fSGLD} - \bar{\Phi}_s^{\lambda,k,fSGLD}\right|^2\right]\mathrm{d}s \\
&\quad + 2\lambda^2\mathbb{E}\left[\left|\int_{kT}^t \left[\nabla g_\epsilon(\bar{\Phi}_s^{\lambda,k,fSGLD}) - \nabla_\theta U(\bar{\Phi}_s^{\lambda,k,fSGLD} + \epsilon_{\lceil s\rceil}, X_{\lceil s\rceil})\right]\mathrm{d}s\right|^2\right].
\end{aligned} \tag{49}$$

We now bound the last term on the right-hand side of (49) by splitting the final integral. Let $kT + N < t \leq kT + N + 1$ with $N + 1 \leq T, N \in \mathbb{N}$. It follows that

$$2\lambda^2 \left| \int_{kT}^t \left[ \nabla g_\epsilon(\bar{\Phi}_s^{\lambda,k,\text{fSGLD}}) - \nabla_\theta U(\bar{\Phi}_s^{\lambda,k,\text{fSGLD}} + \epsilon_{\lceil s \rceil}, X_{\lceil s \rceil}) \right] \mathrm{d}s \right|^2$$

$$= 2\lambda^2 \sum_{n=1}^N \left| \int_{kT+(n-1)}^{kT+n} [\nabla g_\epsilon(\bar{\Phi}_s^{\lambda,k,\text{fSGLD}}) - \nabla_\theta U(\bar{\Phi}_s^{\lambda,k,\text{fSGLD}} + \epsilon_{kT+n}, X_{kT+n})] \mathrm{d}s \right|^2$$

$$+ 4\lambda^2 \sum_{n=2}^N \sum_{j=1}^{n-1} \left\langle \int_{kT+(n-1)}^{kT+n} [\nabla g_\epsilon(\bar{\Phi}_s^{\lambda,k,\text{fSGLD}}) - \nabla_\theta U(\bar{\Phi}_s^{\lambda,k,\text{fSGLD}} + \epsilon_{kT+n}, X_{kT+n})] \mathrm{d}s, \right.$$

$$\left. \int_{kT+(j-1)}^{kT+j} [\nabla g_\epsilon(\bar{\Phi}_s^{\lambda,k,\text{fSGLD}}) - \nabla_\theta U(\bar{\Phi}_s^{\lambda,k,\text{fSGLD}} + \epsilon_{kT+j}, X_{kT+j})] \mathrm{d}s \right\rangle$$

$$+ 4\lambda^2 \sum_{n=1}^N \left\langle \int_{kT+(n-1)}^{kT+n} [\nabla g_\epsilon(\bar{\Phi}_s^{\lambda,k,\text{fSGLD}}) - \nabla_\theta U(\bar{\Phi}_s^{\lambda,k,\text{fSGLD}} + \epsilon_{kT+n}, X_{kT+n})] \mathrm{d}s, \right.$$

$$\left. \int_{kT+N}^t [\nabla g_\epsilon(\bar{\Phi}_s^{\lambda,k,\text{fSGLD}}) - \nabla_\theta U(\bar{\Phi}_s^{\lambda,k,\text{fSGLD}} + \epsilon_{kT+N+1}, X_{kT+N+1})] \mathrm{d}s \right\rangle$$

$$+ 2\lambda^2 \left| \int_{kT+N}^t [\nabla g_\epsilon(\bar{\Phi}_s^{\lambda,k,\text{fSGLD}}) - \nabla_\theta U(\bar{\Phi}_s^{\lambda,k,\text{fSGLD}} + \epsilon_{kT+N+1}, X_{kT+N+1})] \mathrm{d}s \right|^2,$$

Let the filtration $\mathcal{J}_t$ be as in the proof of Lemma D.5. Observe that for any $n = 2, \ldots, N, j = 1, \ldots, n-1$,

$$4\lambda^2 \mathbb{E} \left[ \left\langle \int_{kT+(n-1)}^{kT+n} [\nabla g_\epsilon(\bar{\Phi}_s^{\lambda,k,\text{fSGLD}}) - \nabla_\theta U(\bar{\Phi}_s^{\lambda,k,\text{fSGLD}} + \epsilon_{kT+n}, X_{kT+n})] \mathrm{d}s, \right. \right.$$

$$\left. \left. \int_{kT+(j-1)}^{kT+j} [\nabla g_\epsilon(\bar{\Phi}_s^{\lambda,k,\text{fSGLD}}) - \nabla_\theta U(\bar{\Phi}_s^{\lambda,k,\text{fSGLD}} + \epsilon_{kT+j}, X_{kT+j})] \mathrm{d}s \right\rangle \right]$$

$$= 4\lambda^2 \mathbb{E} \left[ \mathbb{E} \left[ \left\langle \int_{kT+(n-1)}^{kT+n} [\nabla g_\epsilon(\bar{\Phi}_s^{\lambda,k,\text{fSGLD}}) - \nabla_\theta U(\bar{\Phi}_s^{\lambda,k,\text{fSGLD}} + \epsilon_{kT+n}, X_{kT+n})] \mathrm{d}s, \right. \right. \right.$$

$$\left. \left. \left. \int_{kT+(j-1)}^{kT+j} [\nabla g_\epsilon(\bar{\Phi}_s^{\lambda,k,\text{fSGLD}}) - \nabla_\theta U(\bar{\Phi}_s^{\lambda,k,\text{fSGLD}} + \epsilon_{kT+j}, X_{kT+j})] \mathrm{d}s \right\rangle \right| \mathcal{J}_{kT+j} \right] \right]$$

$$= 4\lambda^2 \mathbb{E} \left[ \left\langle \int_{kT+(n-1)}^{kT+n} \mathbb{E} \left[ \nabla g_\epsilon(\bar{\Phi}_s^{\lambda,k,\text{fSGLD}}) - \nabla_\theta U(\bar{\Phi}_s^{\lambda,k,\text{fSGLD}} + \epsilon_{kT+n}, X_{kT+n}) \right| \mathcal{J}_{kT+j} \right] \mathrm{d}s, \right.$$

$$\left. \int_{kT+(j-1)}^{kT+j} [\nabla g_\epsilon(\bar{\Phi}_s^{\lambda,k,\text{fSGLD}}) - \nabla_\theta U(\bar{\Phi}_s^{\lambda,k,\text{fSGLD}} + \epsilon_{kT+j}, X_{kT+j})] \mathrm{d}s \right\rangle \right] = 0.$$

By the same reasoning, we obtain for all $1 \leq n \leq N$

$$4\lambda^2 \mathbb{E} \left[ \left\langle \int_{kT+(n-1)}^{kT+n} [\nabla g_\epsilon(\bar{\Phi}_s^{\lambda,k,\text{fSGLD}}) - \nabla_\theta U(\bar{\Phi}_s^{\lambda,k,\text{fSGLD}} + \epsilon_{kT+n}, X_{kT+n})] \mathrm{d}s, \right. \right.$$

$$\left. \left. \int_{kT+N}^t [\nabla g_\epsilon(\bar{\Phi}_s^{\lambda,k,\text{fSGLD}}) - \nabla_\theta U(\bar{\Phi}_s^{\lambda,k,\text{fSGLD}} + \epsilon_{kT+N+1}, X_{kT+N+1})] \mathrm{d}s \right\rangle \right] = 0.$$

Combining these results, we can bound the last term on the right-hand side of (49) using Lemma D.5

$$
2\lambda^2 \mathbb{E}\left[\left|\int_{kT}^t \left[\nabla g_\epsilon(\bar{\Phi}_s^{\lambda,k,\text{fSGLD}}) - \nabla_\theta U(\bar{\Phi}_s^{\lambda,k,\text{fSGLD}} + \epsilon_{\lceil s \rceil}, X_{\lceil s \rceil})\right] \mathrm{d}s\right|^2\right]
$$

$$
= 2\lambda^2 \sum_{n=1}^N \mathbb{E}\left[\left|\int_{kT+(n-1)}^{kT+n} \left[\nabla g_\epsilon(\bar{\Phi}_s^{\lambda,k,\text{fSGLD}}) - \nabla_\theta U(\bar{\Phi}_s^{\lambda,k,\text{fSGLD}} + \epsilon_{kT+n}, X_{kT+n})\right]\mathrm{d}s\right|^2\right]
$$

$$
+ 2\lambda^2 \mathbb{E}\left[\left|\int_{kT+N}^t \left[\nabla g_\epsilon(\bar{\Phi}_s^{\lambda,k,\text{fSGLD}}) - \nabla_\theta U(\bar{\Phi}_s^{\lambda,k,\text{fSGLD}} + \epsilon_{kT+N+1}, X_{kT+N+1})\right]\mathrm{d}s\right|^2\right]
$$

$$
\leq 4e^{-a\lambda kT/2}\lambda(\bar{\psi}_Z \mathbb{E}[\widetilde{V}_2(\theta_0)] + \widetilde{\psi}_Z).
$$

Consequently, (49) is bounded as follows

$$
\mathbb{E}\left[\left|\bar{\Phi}_t^{\lambda,k,\text{fSGLD}} - \bar{\theta}_t^{\text{fSGLD}}\right|^2\right] \leq 4\lambda L_1^2 \mathbb{E}\left[\varphi^2(X_0)\right] \int_{kT}^t \mathbb{E}\left[\left|\bar{\theta}_s^{\text{fSGLD}} - \bar{\Phi}_s^{\lambda,k,\text{fSGLD}}\right|^2\right]\mathrm{d}s
$$
$$
+ 4e^{-a\lambda kT/2}\lambda(L_1^2 \mathbb{E}\left[\varphi^2(X_0)\right] \bar{\psi}_Y + \bar{\psi}_Z)\mathbb{E}[\widetilde{V}_2(\theta_0)]
$$
$$
+ 4\lambda(L_1^2 \mathbb{E}\left[\varphi^2(X_0)\right] \widetilde{\psi}_Y + \widetilde{\psi}_Z).
$$

Applying Grönwall's inequality yields

$$
\mathbb{E}\left[\left|\bar{\Phi}_t^{\lambda,k,\text{fSGLD}} - \bar{\theta}_t^{\text{fSGLD}}\right|^2\right] \leq \lambda e^{4L_1^2 \mathbb{E}[\varphi^2(X_0)]}\left[4e^{-a\lambda kT/2}(L_1^2 \mathbb{E}\left[\varphi^2(X_0)\right] \bar{\psi}_Y + \bar{\psi}_Z)\mathbb{E}[\widetilde{V}_2(\theta_0)]\right.
$$
$$
\left. + 4(L_1^2 \mathbb{E}\left[\varphi^2(X_0)\right] \widetilde{\psi}_Y + \widetilde{\psi}_Z)\right].
$$

Finally, we obtain using $\lambda T \geq 1/2$,

$$
W_2^2(\mathcal{L}(\bar{\theta}_t^{\text{fSGLD}}), \mathcal{L}(\bar{\Phi}_t^{\lambda,k,\text{fSGLD}})) \leq \mathbb{E}\left|\bar{\Phi}_t^{\lambda,k,\text{fSGLD}} - \bar{\theta}_t^{\text{fSGLD}}\right|^2 \tag{50}
$$
$$
\leq \lambda(e^{-an/4}\bar{D}_{2,1}\mathbb{E}[\widetilde{V}_2(\theta_0)] + \bar{D}_{2,2}),
$$

where

$$
\bar{D}_{2,1} := 4e^{4L_1^2 \mathbb{E}[\varphi^2(X_0)]}(L_1^2 \mathbb{E}\left[\varphi^2(X_0)\right] \bar{\psi}_Y + \bar{\psi}_Z),
$$
$$
\bar{D}_{2,2} := 4e^{4L_1^2 \mathbb{E}[\varphi^2(X_0)]}(L_1^2 \mathbb{E}\left[\varphi^2(X_0)\right] \widetilde{\psi}_Y + \widetilde{\psi}_Z).
$$

$\square$

### D.2.2. BOUND FOR $W_p(\mathcal{L}(\bar{\Phi}_t^{\lambda,k,\text{FSGLD}}), \mathcal{L}(Y_t^{\lambda,\text{FSGLD}}))$

Let $\mathcal{P}_{\widetilde{V}_p}$ denote the set of $\mu \in \mathcal{P}(\mathbb{R}^d)$ satisfying $\int_{\mathbb{R}^d} \widetilde{V}_p(\theta)\, \mu(\mathrm{d}\theta) < \infty$. For $\mu, \nu \in \mathcal{P}_{\widetilde{V}_2}$, let

$$
w_{1,2}(\mu,\nu) := \inf_{\varrho \in \mathcal{A}(\mu,\nu)} \int_{\mathbb{R}^d} \int_{\mathbb{R}^d} [1 \wedge |\theta - \theta'|](1 + \widetilde{V}_2(\theta) + \widetilde{V}_2(\theta'))\, \varrho(\mathrm{d}\theta, \mathrm{d}\theta'), \tag{51}
$$

where we recall that $\mathcal{A}(\mu,\nu)$ denotes the set of probability measures $\varrho$ on $\mathcal{B}(\mathbb{R}^{2d})$ such that its respective marginals are $\mu$ and $\nu$.

**Proposition D.8.** *Let Assumptions 3.1, 3.2, and 3.3 hold. Let $(\widetilde{Y}_t^{fSGLD})_{t \in \mathbb{R}_+}$ be the solution of (37) with initial condition $\widetilde{Y}_0^{fSGLD} = \widetilde{\theta}_0$ which is independent of $\mathcal{F}_\infty$ and satisfies $\mathbb{E}[|\widetilde{\theta}_0|^2] < \infty$. Then,*

$$
w_{1,2}(\mathcal{L}(Y_t^{fSGLD}), \mathcal{L}(\widetilde{Y}_t^{fSGLD})) \leq \hat{c}e^{-\dot{c}t}w_{1,2}(\mathcal{L}(\theta_0), \mathcal{L}(\widetilde{\theta}_0)),
$$

*where the constants $\dot{c}$ and $\hat{c}$ are given in Lemma D.9.*

*Proof.* From Assumptions 3.1 and 3.2, one can deduce

$$|\nabla g_\epsilon(\theta) - \nabla g_\epsilon(\theta')| \leq L_1 \mathbb{E}[\varphi(X_0)]|\theta - \theta'|. \tag{52}$$

The rest of the proof follows using Assumptions 3.1, 3.2, and 3.3, (52), and Lemma D.3 in the proof of Proposition 4.6 in (Zhang et al., 2023). □

Next, we provide explicitly the constants $\dot{c}$ and $\hat{c}$ appearing in Proposition D.8.

**Lemma D.9.** *The contraction constant $\dot{c} > 0$ in Proposition D.8 is given by*

$$\dot{c} := \min\left\{\bar{\phi}, a/2, 3a^2 v_2(\bar{M}_2)\varepsilon\right\}/2 \tag{53}$$

*where $a$ is from Remark C.2, $\bar{M}_2$ is from Lemma D.3, and*

$$\begin{aligned}
\bar{\phi} := &\left(2\sqrt{(3v_2(\bar{M}_2)(1+a/2)-1)(8\pi/(\beta L_1 \mathbb{E}[\varphi(X_0)]))}\right)^{-1} \\
&\times \left(\exp\left(\left(\sqrt{(3v_2(\bar{M}_2)(1+a/2)-1)(\beta L_1 \mathbb{E}[\varphi(X_0)]/8)} + \sqrt{8/(\beta L_1 \mathbb{E}[\varphi(X_0)])}\right)^2\right)\right)^{-1},
\end{aligned} \tag{54}$$

*Furthermore, $\varepsilon > 0$ in (53) is chosen such that*

$$\varepsilon \leq 1 \wedge \left(6av_2(\bar{M}_2)\sqrt{\frac{2\beta\pi}{L_1 \mathbb{E}[\varphi(X_0)]}}\int_0^{2\sqrt{6v_2(\bar{M}_2)-1}} \exp\left(s\sqrt{\frac{\beta L_1 \mathbb{E}[\varphi(X_0)]}{8}} + \sqrt{\frac{8}{\beta L_1 \mathbb{E}[\varphi(X_0)]}}\right)^2 \mathrm{d}s\right)^{-1}. \tag{55}$$

*In addition, the constant $\hat{c} > 0$ is given by*

$$\begin{aligned}
\hat{c} := &2\left(1 + 2\sqrt{3v_2(\bar{M}_2)(1+a/2)-1}\right) \\
&\times \exp\left(\beta L_1 \mathbb{E}[\varphi(X_0)](3v_2(\bar{M}_2)(1+a/2)-1)/2 + 4\sqrt{3v_2(\bar{M}_2)(1+a/2)-1}\right)/\varepsilon.
\end{aligned} \tag{56}$$

*In particular,*

$$\begin{aligned}
\dot{c} = &O\left(\frac{32\sqrt{\pi}(1+a^2)(1+\beta)}{a^2\sqrt{\beta}}\left(1 + \frac{1}{\sqrt{L_1 \mathbb{E}[\varphi(X_0)]}}\right)\right. \\
&\times \left. e^{\left(8(1+2/a)(1+a+((2\zeta)^{-1}+2\zeta L_1^2 \mathbb{E}[\varphi^2(X_0)])\sigma^2 d + 4\zeta L_2^2 \mathbb{E}[\bar{\varphi}^2(X_0)])(1+\beta L_1 \mathbb{E}[\varphi(X_0)])(1+d(\sigma^2+\beta^{-1}))+\frac{16}{\beta L_1 \mathbb{E}[\varphi(X_0)]}\right)}\right)^{-1},
\end{aligned}$$

*and*

$$\begin{aligned}
\hat{c} = &O\left(\sqrt{\frac{\beta}{L_1 \mathbb{E}[\varphi(X_0)]}}\left(1 + d(\sigma^2+\beta^{-1})\right)^2\right. \\
&\times \left. e^{\left(12(1+2/a)(1+a+((2\zeta)^{-1}+2\zeta L_1^2 \mathbb{E}[\varphi^2(X_0)])\sigma^2 d + 4\zeta L_2^2 \mathbb{E}[\bar{\varphi}^2(X_0)])(1+\beta L_1 \mathbb{E}[\varphi(X_0)])(1+d(\sigma^2+\beta^{-1}))+\frac{16}{\beta L_1 \mathbb{E}[\varphi(X_0)]}\right)}\right).
\end{aligned}$$

*Proof.* This follows by adapting the arguments in the proof of Lemma 4.11 of (Zhang et al., 2023) to the *flatness* Langevin SDE (37), using (52) together with Lemma D.3. □

Next, we establish uniform bounds for $\widetilde{V}_4(\bar{\theta}_t^{fSGLD})$, $\widetilde{V}_2(\bar{\Phi}_t^{\lambda,k,\text{fSGLD}})$, and $\widetilde{V}_4(\bar{\Phi}_t^{\lambda,k,\text{fSGLD}})$ which will be used to control $W_p(\mathcal{L}(\bar{\Phi}_t^{\lambda,k,\text{fSGLD}}), \mathcal{L}(Y_t^{\lambda,\text{fSGLD}}))$.

**Corollary D.10.** *Let Assumptions 3.1, 3.2 and 3.3 hold. For any $0 < \lambda < \lambda_{max}$, $k \in \mathbb{N}$, $t \in (k, k+1]$,*

$$\mathbb{E}[\widetilde{V}_4(\bar{\theta}_t^{fSGLD})] \leq 2(1-a\lambda)^{\lfloor t \rfloor}\mathbb{E}[\widetilde{V}_4(\bar{\theta}_0^{fSGLD})] + 2\alpha_3(\lambda_{max} + a^{-1}) + 2,$$

*where $\alpha_3$ is given in Lemma D.2.*

*Proof.* This follows from the definition of the Lyapunov function $\widetilde{V}_4$ together with Lemma D.2. $\qquad\square$

**Lemma D.11.** *Let Assumptions 3.1, 3.2 and 3.3 hold. For any $0 < \lambda < \lambda_{max}$, $t \geq kT$, with $k \in \mathbb{N}$, the following inequality holds*

$$\mathbb{E}[\widetilde{V}_2(\bar{\Phi}_t^{\lambda,k,fSGLD})] \leq e^{-\lambda ta/2}\mathbb{E}[\widetilde{V}_2(\theta_0)] + \alpha_1(\lambda_{max} + a^{-1}) + 3\widetilde{v}_2(\bar{M}_2) + 1,$$

*where $\alpha_1$ is given in Lemma D.2. In addition, the following inequality holds*

$$\mathbb{E}[\widetilde{V}_4(\bar{\Phi}_t^{\lambda,k,fSGLD})] \leq 2e^{-a\lambda t}\mathbb{E}[\widetilde{V}_4(\bar{\theta}_0^{fSGLD})] + 3\widetilde{v}_4(\bar{M}_4) + 2\alpha_3(\lambda_{max} + a^{-1}) + 2,$$

*where $\bar{M}_2$ and $\bar{M}_4$ are given in Lemma D.3, and $\alpha_3$ is given in Lemma D.2.*

*Proof.* This follows by applying Lemma D.2, Corollary D.10, and Lemma D.3 in the proof of Lemma 4.5 of (Zhang et al., 2023). $\qquad\square$

The bound for the second term on the right-hand side of (42) is established in the following lemma and corollary.

**Lemma D.12.** *Let Assumptions 3.1, 3.2, and 3.3 hold, and let $\sigma = \beta^{-\frac{1+\eta}{4}}$ for $\eta \in (0,1)$. For any $0 < \lambda < \lambda_{max}$ given in (36), $t \in (kT, (k+1)T]$,*

$$W_1(\mathcal{L}(\bar{\Phi}_t^{\lambda,k,fSGLD}), \mathcal{L}(Y_t^{\lambda,fSGLD})) \leq \sqrt{\lambda}(e^{-\dot{c}k/2}\bar{D}_{2,3}\mathbb{E}[\widetilde{V}_4(\theta_0)] + \bar{D}_{2,4}),$$

*where*

$$
\begin{aligned}
\bar{D}_{2,3} &= \hat{c}\left(1 + \frac{2}{\dot{c}}\right)(e^{a/2}\bar{D}_{2,1} + 12) \\
&= O\left(e^{D_\star(\beta + d + d\beta^{(1-\eta)/2} + d^2\beta^{-\eta} + 1)}\left(1 + \frac{1}{\dot{c}}\right)\right), \\
\bar{D}_{2,4} &= \frac{\hat{c}}{1 - \exp(-\dot{c})}(\bar{D}_{2,2} + 12\alpha_3(\lambda_{max} + a^{-1}) + 9\widetilde{v}_4(\bar{M}_4) + 15) \\
&= O\left(e^{D_\star(\beta + d + d\beta^{(1-\eta)/2} + d^2\beta^{-\eta} + 1)}(1 - e^{-\dot{c}})^{-1}\right),
\end{aligned}
\tag{57}
$$

*with $\bar{D}_{2,1}$, $\bar{D}_{2,2}$ given in (47), $\hat{c}$, $\dot{c}$ given in Lemma D.9, $\alpha_3$ given in (44), and $\bar{M}_4$ given in Lemma D.3.*

*Proof.* This follows by applying Proposition D.8, Lemma D.7, Corollary D.10, and Lemma D.11 together with the arguments in the proof of Lemma 4.8 of (Zhang et al., 2023). $\qquad\square$

**Corollary D.13.** *Let Assumptions 3.1, 3.2, and 3.3 hold, and let $\sigma = \beta^{-\frac{1+\eta}{4}}$ for $\eta \in (0,1)$. For any $0 < \lambda < \lambda_{max}$ given in (36), $t \in (kT, (k+1)T]$,*

$$W_2(\mathcal{L}(\bar{\Phi}_t^{\lambda,k,fSGLD}), \mathcal{L}(Y_t^{\lambda,fSGLD})) \leq \lambda^{1/4}(e^{-\dot{c}k/4}\bar{D}_{2,3}^\star(\mathbb{E}[\widetilde{V}_4(\theta_0)])^{1/2} + \bar{D}_{2,4}^\star),$$

*where*

$$
\begin{aligned}
\bar{D}_{2,3}^\star &:= \sqrt{2\hat{c}}(1 + 4/\dot{c})(e^{a/8}\bar{D}_{2,1}^{1/2} + 2\sqrt{2}) \\
&= O\left(e^{D_\star(\beta + d + d\beta^{(1-\eta)/2} + d^2\beta^{-\eta} + 1)}(1 + 1/\dot{c})\right), \\
\bar{D}_{2,4}^\star &:= \frac{\sqrt{2\hat{c}}}{1 - \exp(-\dot{c}/2)}(\bar{D}_{2,2}^{1/2} + 2\sqrt{2\alpha_3}(\lambda_{max} + a^{-1})^{1/2} + \sqrt{3}\widetilde{v}_4^{1/2}(\bar{M}_4) + \sqrt{15}) \\
&= O\left(e^{D_\star(\beta + d + d\beta^{(1-\eta)/2} + d^2\beta^{-\eta} + 1)}(1 - e^{-\dot{c}/2})^{-1}\right),
\end{aligned}
\tag{58}
$$

*with $\bar{D}_{2,1}$, $\bar{D}_{2,2}$ given in (47), $\hat{c}$, $\dot{c}$ given in Lemma D.9, $\alpha_3$ given in (44), and $\bar{M}_4$ given in Lemma D.3.*

*Proof.* This follows using Proposition D.8, Lemma D.7, Corollary D.10, and Lemma D.11 in the proof of Corollary 4.9 of (Zhang et al., 2023). □

### D.2.3. BOUND FOR $W_p(\mathcal{L}(Y_t^{\lambda,\text{FSGLD}}), \pi_\beta^{\text{FSGLD}})$

In this section, we derive a non-asymptotic bound for the third term on the right-hand side of (42).

**Theorem D.14.** *Let Assumptions 3.1, 3.2, and 3.3 hold. Then, there exist constants $\dot{c}, \hat{c} > 0$ such that, for every $\beta > 0$, for $0 < \lambda < \lambda_{max}$, any $t \in (kT, (k+1)T]$, and $k \in \mathbb{N}$,*

$$W_1(\mathcal{L}(Y_t^{\lambda,f\text{SGLD}}), \pi_\beta^{f\text{SGLD}}) \leq \hat{c}e^{-\dot{c}\lambda t}\left(1 + \mathbb{E}[\widetilde{V}_2(\theta_0)] + \int_{\mathbb{R}^d}\widetilde{V}_2(\theta)\pi_\beta^{f\text{SGLD}}(\mathrm{d}\theta)\right),$$

*with $\hat{c}, \dot{c}$ given in Lemma D.9.*

*Proof.* Using $W_1(\mu, \nu) \leq w_{1,2}(\mu, \nu)$, where the latter is defined in (51), and Proposition D.8, we have for $t \in (kT, (k+1)T]$

$$\begin{aligned}
W_1(\mathcal{L}(Y_t^{\lambda,\text{fSGLD}}), \pi_\beta^{\text{fSGLD}}) &\leq w_{1,2}(\mathcal{L}(Y_t^{\lambda,\text{fSGLD}}), \pi_\beta^{\text{fSGLD}}) \\
&\leq \hat{c}e^{-\dot{c}\lambda t}w_{1,2}(\mathcal{L}(\theta_0), \pi_\beta^{\text{fSGLD}}) \\
&\leq \hat{c}e^{-\dot{c}\lambda t}\left(1 + \mathbb{E}[\widetilde{V}_2(\theta_0)] + \int_{\mathbb{R}^d}\widetilde{V}_2(\theta)\pi_\beta^{\text{fSGLD}}(\mathrm{d}\theta)\right).
\end{aligned}$$

□

**Corollary D.15.** *Let Assumptions 3.1, 3.2, and 3.3 hold. Then, there exist constants $\dot{c}, \hat{c} > 0$ such that, for every $\beta > 0$, for $0 < \lambda < \lambda_{max}$, any $t \in (kT, (k+1)T]$, and $k \in \mathbb{N}$,*

$$W_2(\mathcal{L}(Y_t^{\lambda,f\text{SGLD}}), \pi_\beta^{f\text{SGLD}}) \leq \sqrt{2}\hat{c}^{1/2}e^{-\dot{c}\lambda t/2}\left(1 + \mathbb{E}[\widetilde{V}_2(\theta_0)] + \int_{\mathbb{R}^d}\widetilde{V}_2(\theta)\pi_\beta^{f\text{SGLD}}(\mathrm{d}\theta)\right)^{1/2},$$

*with $\hat{c}, \dot{c}$ given in Lemma D.9.*

*Proof.* Using $W_2(\mu, \nu) \leq \sqrt{2w_{1,2}(\mu, \nu)}$, and Proposition D.8, we have for $t \in (kT, (k+1)T]$

$$\begin{aligned}
W_2(\mathcal{L}(Y_t^{\lambda,\text{fSGLD}}), \pi_\beta^{\text{fSGLD}}) &\leq \sqrt{2w_{1,2}(\mathcal{L}(Y_t^{\lambda,\text{fSGLD}}), \pi_\beta^{\text{fSGLD}})} \\
&\leq \sqrt{2}\hat{c}^{1/2}e^{-\dot{c}\lambda t/2}\sqrt{w_{1,2}(\mathcal{L}(\theta_0), \pi_\beta^{\text{fSGLD}})} \\
&\leq \sqrt{2}\hat{c}^{1/2}e^{-\dot{c}\lambda t/2}\left(1 + \mathbb{E}[\widetilde{V}_2(\theta_0)] + \int_{\mathbb{R}^d}\widetilde{V}_2(\theta)\pi_\beta^{\text{fSGLD}}(\mathrm{d}\theta)\right)^{1/2}.
\end{aligned}$$

□

### D.2.4. FINAL BOUND FOR $W_p(\mathcal{L}(\theta_k^{\text{FSGLD}}), \pi_{\beta,\sigma}^\star)$

*Proof of Theorem 3.5.* Using Lemma D.7, Lemma D.12, Theorem D.14, and Proposition 3.4 in (42), we obtain for $t \in (kT, (k+1)T]$,

$$\begin{aligned}
W_1(\mathcal{L}(\theta_k^{\text{fSGLD}}), \pi_{\beta,\sigma}^\star) &\leq W_1(\mathcal{L}(\bar{\theta}_t^{\text{fSGLD}}), \mathcal{L}(\bar{\Phi}_t^{\lambda,k,\text{fSGLD}})) + W_1(\mathcal{L}(\bar{\Phi}_t^{\lambda,k,\text{fSGLD}}), \mathcal{L}(Y_t^{\lambda,\text{fSGLD}})) \\
&\quad + W_1(\mathcal{L}(Y_t^{\lambda,\text{fSGLD}}), \pi_\beta^{\text{fSGLD}}) + W_1(\pi_\beta^{\text{fSGLD}}, \pi_{\beta,\sigma}^\star) \\
&\leq (\bar{D}_{2,1}^{1/2} + \bar{D}_{2,2}^{1/2} + \bar{D}_{2,3} + \bar{D}_{2,4})\sqrt{\lambda}[(e^{-\dot{c}k/2}\mathbb{E}[\widetilde{V}_4(\theta_0)] + 1)] \\
&\quad + \hat{c}e^{-\dot{c}\lambda t}\left(1 + \mathbb{E}[\widetilde{V}_2(\theta_0)] + \int_{\mathbb{R}^d}\widetilde{V}_2(\theta)\pi_\beta^{\text{fSGLD}}(\mathrm{d}\theta)\right) + W_2(\pi_\beta^{\text{fSGLD}}, \pi_{\beta,\sigma}^\star) \\
&\leq 2e^{-\dot{c}k/2}(\lambda_{max}^{1/2}(\bar{D}_{2,1}^{1/2} + \bar{D}_{2,2}^{1/2} + \bar{D}_{2,3} + \bar{D}_{2,4}) + \hat{c})(1 + \mathbb{E}[|\theta_0|^4]) \\
&\quad + \hat{c}e^{-\dot{c}k/2}\left(1 + \int_{\mathbb{R}^d}\widetilde{V}_2(\theta)\pi_\beta^{\text{fSGLD}}(\mathrm{d}\theta)\right)(1 + \mathbb{E}[|\theta_0|^4]) \\
&\quad + \sqrt{\lambda}(\bar{D}_{2,1}^{1/2} + \bar{D}_{2,2}^{1/2} + \bar{D}_{2,3} + \bar{D}_{2,4}) + \underline{D},
\end{aligned} \tag{59}$$

where $\bar{D}_{2,1}$, $\bar{D}_{2,2}$ are given in (47) (Lemma D.7), $\bar{D}_{2,3}$, $\bar{D}_{2,4}$ are given in (57) (Lemma D.12), and $\underline{D}$ is given in (35). The above result implies, for any $k \in \mathbb{N}$,

$$W_1(\mathcal{L}(\theta^{\text{fSGLD}}_{(k+1)T}), \pi^\star_{\beta,\sigma}) \leq D_1 e^{-\dot{c}(k+1)/2}(1 + \mathbb{E}[|\theta_0|^4]) + (D_2 + D_3)\sqrt{\lambda} + \underline{D}, \tag{60}$$

where, for $\sigma = \beta^{-\frac{1+\eta}{4}}$, the constants are

$$\begin{aligned}
D_1 &:= 2e^{\dot{c}/2}\left[(\lambda^{1/2}_{\max}(\bar{D}^{1/2}_{2,1} + \bar{D}^{1/2}_{2,2} + \bar{D}_{2,3} + \bar{D}_{2,4}) + \hat{c}) + \hat{c}\left(1 + \int_{\mathbb{R}^d} \widetilde{V}_2(\theta)\pi^{\text{fSGLD}}_\beta(\mathrm{d}\theta)\right)\right] \\
&= O\left(e^{D_\star(\beta+d+d\beta^{(1-\eta)/2}+d^2\beta^{-\eta}+1)}\left(1 + \frac{1}{1 - e^{-\dot{c}}}\right)\right), \\
D_2 &:= \bar{D}^{1/2}_{2,1} + \bar{D}^{1/2}_{2,2} = O\left(1 + \sqrt{\frac{d}{\beta^{(1+\eta)/2}}}\right), \\
D_3 &:= \bar{D}_{2,3} + \bar{D}_{2,4} = O\left(e^{D_\star(\beta+d+d\beta^{(1-\eta)/2}+d^2\beta^{-\eta}+1)}\left(1 + \frac{1}{1 - e^{-\dot{c}}}\right)\right),
\end{aligned} \tag{61}$$

with $\hat{c}$, $\dot{c}$ given in Lemma D.9 and $D_\star > 0$ is independent of $d$, $\beta$, $k$. To obtain a non-asymptotic error bound for $(\bar{\theta}^{\text{fSGLD}}_{(k+1)})_{k\in\mathbb{N}}$, we set $(k+1)T$ to $k+1$ on the left-hand side of (60), and set $k+1$ to $(k+1)/T$ on the right-hand side of (60). By using $\lambda(k+1) \leq (k+1)/T$, it follows that, for any $k \in \mathbb{N}$,

$$W_1(\mathcal{L}(\theta^{\text{fSGLD}}_{k+1}), \pi^\star_{\beta,\sigma}) \leq D_1 e^{-\dot{c}\lambda(k+1)/2}(1 + \mathbb{E}[|\theta_0|^4]) + (D_2 + D_3)\sqrt{\lambda} + \underline{D}. \tag{62}$$

In addition, for any $\bar{\delta} > 0$, if we choose $\lambda$, $k$ and $\beta$ in (11) such that $\lambda \leq \lambda_{\max}$, and

$$D_1 e^{-\dot{c}\lambda k/2}(1 + \mathbb{E}[|\theta_0|^4]) \leq \frac{\bar{\delta}}{3}, \qquad (D_2 + D_3)\sqrt{\lambda} \leq \frac{\bar{\delta}}{3}, \qquad \underline{D} \leq \frac{\bar{\delta}}{3},$$

then $W_1(\mathcal{L}(\theta^{\text{fSGLD}}_k), \pi^\star_{\beta,\sigma}) \leq \bar{\delta}$. This yields

$$\beta \geq \beta_{\bar{\delta}} := \underline{D}^0 \max\left(\left(\frac{9\sqrt{d}}{\bar{\delta}}\right)^{\frac{4}{\eta}}, \left(\frac{9d}{\bar{\delta}}\right)^{\frac{2}{\eta}}, \left(\frac{9d^2}{\bar{\delta}}\right)^{\frac{2}{1+\eta}}\right), \tag{63}$$

where $\underline{D}^0 > 0$ is a constant collecting all the terms in (35) with no dependence on $d$ or $\beta$, and

$$\lambda \leq \lambda_{\bar{\delta}} := \frac{\bar{\delta}^2}{9(D_2 + D_3)^2} \wedge \lambda_{\max}, \tag{64}$$

and $\lambda k \geq \frac{2}{\dot{c}} \ln\left(\frac{3D_1(1+\mathbb{E}[|\theta_0|^4])}{\bar{\delta}}\right)$. From (61), it follows that

$$\begin{aligned}
k \geq k_{\bar{\delta}} := &\frac{D_\star e^{D_\star(\beta+d+d\beta^{(1-\eta)/2}+d^2\beta^{-\eta}+1)}}{\bar{\delta}^2\dot{c}}\left(1 + \frac{1}{(1 - e^{-\dot{c}})^2}\right) \\
&\times \ln\left(\frac{D_\star e^{D_\star(\beta+d+d\beta^{(1-\eta)/2}+d^2\beta^{-\eta}+1)}}{\bar{\delta}}\left(1 + \frac{1}{1 - e^{-\dot{c}}}\right)\right).
\end{aligned} \tag{65}$$

$\square$

**Corollary D.16.** *Let Assumption 3.1, 3.2 and 3.3 hold, and let $\sigma = \beta^{-\frac{1+\eta}{4}}$ for $\eta \in (0,1)$. Then, there exists constants $\dot{c}, D_4, D_5, D_6, \underline{D} > 0$ such that, for every $\beta > 0$, $0 < \lambda \leq \lambda_{max}$ with $\lambda_{max}$ given in (36), and $k \in \mathbb{N}$,*

$$W_2(\mathcal{L}(\theta^{\text{fSGLD}}_k), \pi^\star_{\beta,\sigma}) \leq D_4 e^{-\dot{c}\lambda k/4}(\mathbb{E}[|\theta_0|^4] + 1) + (D_5 + D_6)\lambda^{1/4} + \underline{D}, \tag{66}$$

*where $\dot{c}$ is given in Lemma D.9, $\underline{D}$ is the same as in Theorem 3.5 and is given explicitly in (35) and,*

$$D_4 = O\left(e^{D_\star(\beta+d+d\beta^{(1-\eta)/2}+d^2\beta^{-\eta}+1)}\left(1+\frac{1}{1-e^{-\dot{c}/2}}\right)\right),$$

$$D_5 = O\left(1+\sqrt{\frac{d}{\beta^{(1+\eta)/2}}}\right),$$

$$D_6 = O\left(e^{D_\star(\beta+d+d\beta^{(1-\eta)/2}+d^2\beta^{-\eta}+1)}\left(1+\frac{1}{1-e^{-\dot{c}/2}}\right)\right),$$

*with $D_\star > 0$ independent of $d$, $\beta$, $k$. The explicit expressions of $D_4$, $D_5$, $D_6$ are given in (68). In addition, let $\beta_{\widetilde{\delta}}$, $\lambda_{\widetilde{\delta}}$, $k_{\widetilde{\delta}}$ be as in (69), (70), and (71) respectively. For any $\widetilde{\delta} > 0$, if we choose $\beta \geq \beta_{\widetilde{\delta}}$, $\lambda \leq \lambda_{\widetilde{\delta}}$, and $k \geq k_{\widetilde{\delta}}$, then*

$$W_2(\mathcal{L}(\theta_k^{fSGLD}), \pi_{\beta,\sigma}^\star) \leq \widetilde{\delta}.$$

*Remark* D.17. The convergence rate in Corollary D.16 can be improved to $O(\lambda^{\frac{1}{2}})$ under substantially stronger assumptions than Assumption 3.1, 3.2 and 3.3. For example, one may assume that the $\pi_\beta^{fSGLD}$ satisfies a log-Sobolev inequality, as in (Huang et al., 2025). However, such assumptions go beyond the scope of our work.

*Proof of Corollary D.16.* Using Lemma D.7, Corollary D.13, Corollary D.15, and Proposition 3.4 in (42), and $\lambda T > 1/2$, we obtain for $t \in (kT, (k+1)T]$,

$$\begin{aligned}
W_2(\mathcal{L}(\theta_k^{fSGLD}), \pi_{\beta,\sigma}^\star) &\leq W_2(\mathcal{L}(\bar{\theta}_t^{fSGLD}), \mathcal{L}(\bar{\Phi}_t^{\lambda,k,fSGLD})) + W_2(\mathcal{L}(\bar{\Phi}_t^{\lambda,k,fSGLD}), \mathcal{L}(Y_t^{\lambda,fSGLD})) \\
&\quad + W_2(\mathcal{L}(Y_t^{\lambda,fSGLD}), \pi_\beta^{fSGLD}) + W_2(\pi_\beta^{fSGLD}, \pi_{\beta,\sigma}^\star) \\
&\leq \sqrt{\lambda}\left(e^{-ak/4}\bar{D}_{2,1}\mathbb{E}[\widetilde{V}_2(\theta_0)] + \bar{D}_{2,2}\right)^{1/2} + \lambda^{1/4}(e^{-\dot{c}/4}\bar{D}_{2,3}^\star(\mathbb{E}[\widetilde{V}_4(\theta_0)])^{1/2} + \bar{D}_{2,4}^\star) \\
&\quad + \sqrt{2}\hat{c}^{1/2}e^{-\dot{c}\lambda t/2}\left(1+\mathbb{E}[\widetilde{V}_2(\theta_0)]+\int_{\mathbb{R}^d}\widetilde{V}_2(\theta)\pi_\beta^{fSGLD}(\mathrm{d}\theta)\right)^{1/2} + \underline{D} \quad\quad (67) \\
&\leq 2e^{-\dot{c}k/4}(\lambda_{\max}^{1/2}(\bar{D}_{2,1}^{1/2}+\bar{D}_{2,2}^{1/2}) + \lambda_{\max}^{1/4}(\bar{D}_{2,3}^\star+\bar{D}_{2,4}^\star) + \sqrt{2}\hat{c}^{1/2})(1+\mathbb{E}[|\theta_0|^4]) \\
&\quad + \sqrt{2}\hat{c}^{1/2}e^{-\dot{c}k/4}\left(1+\int_{\mathbb{R}^d}\widetilde{V}_2(\theta)\pi_\beta^{fSGLD}(\mathrm{d}\theta)\right) \\
&\leq D_4e^{-\dot{c}\lambda k/4}(\mathbb{E}[|\theta_0|^4]+1) + (D_5+D_6)\lambda^{1/4} + \underline{D},
\end{aligned}$$

where

$$\begin{aligned}
D_4 &:= 2(\lambda_{\max}^{1/2}(\bar{D}_{2,1}^{1/2}+\bar{D}_{2,2}^{1/2}) + \lambda_{\max}^{1/4}(\bar{D}_{2,3}^\star+\bar{D}_{2,4}^\star) + \sqrt{2}\hat{c}^{1/2}) \\
&\quad + \sqrt{2}\hat{c}^{1/2}\left(1+\int_{\mathbb{R}^d}\widetilde{V}_2(\theta)\pi_\beta^{fSGLD}(\mathrm{d}\theta)\right) \\
&= O\left(e^{D_\star(\beta+d+d\beta^{(1-\eta)/2}+d^2\beta^{-\eta}+1)}\left(1+\frac{1}{1-e^{-\dot{c}/2}}\right)\right) \\
D_5 &:= \lambda_{\max}^{1/4}\bar{D}_{2,1}^{1/2} + \lambda_{\max}^{1/4}\bar{D}_{2,2}^{1/2} = O\left(1+\sqrt{\frac{d}{\beta^{(1+\eta)/2}}}\right) \quad\quad (68) \\
D_6 &:= \bar{D}_{2,3}^\star + \bar{D}_{2,4}^\star = O\left(e^{D_\star(\beta+d+d\beta^{(1-\eta)/2}+d^2\beta^{-\eta}+1)}\left(1+\frac{1}{1-e^{-\dot{c}/2}}\right)\right),
\end{aligned}$$

where $\hat{c}$, $\dot{c}$ given in Lemma D.9, and $D_\star > 0$ is independent of $d$, $\beta$, $k$. In addition, for any $\widetilde{\delta} > 0$, $\lambda$, $k$ and $\beta$ such that $\lambda \leq \lambda_{\max}$, and

$$D_4e^{-\dot{c}\lambda k/4}(\mathbb{E}[|\theta_0|^4]+1) \leq \frac{\widetilde{\delta}}{3}, \qquad (D_5+D_6)\lambda^{1/4} \leq \frac{\widetilde{\delta}}{3}, \qquad \underline{D} \leq \frac{\widetilde{\delta}}{3},$$

then $W_2(\mathcal{L}(\theta_k^{\text{fSGLD}}), \pi_{\beta,\sigma}^\star) \leq \widetilde{\delta}$. This yields

$$\beta \geq \beta_{\widetilde{\delta}} := \underline{D}^0 \max\left(\left(\frac{9\sqrt{d}}{\widetilde{\delta}}\right)^{\frac{4}{\eta}}, \left(\frac{9d}{\widetilde{\delta}}\right)^{\frac{2}{\eta}}, \left(\frac{9d^2}{\widetilde{\delta}}\right)^{\frac{2}{1+\eta}}\right), \tag{69}$$

where $\underline{D}^0 > 0$ is the same as in the proof of Theorem 3.5,

$$\lambda \leq \lambda_{\widetilde{\delta}} := \frac{\widetilde{\delta}^4}{81(D_5 + D_6)^4} \wedge \lambda_{\max}, \tag{70}$$

and $\lambda k \geq \frac{4}{\dot{c}} \ln\left(\frac{3D_4(1+\mathbb{E}[|\theta_0|^4])}{\widetilde{\delta}}\right)$. From (68), it follows that

$$\begin{aligned}
k \geq k_{\widetilde{\delta}} &:= \frac{D_\star e^{D_\star(\beta+d+d\beta^{(1-\eta)/2}+d^2\beta^{-\eta}+1)}}{\widetilde{\delta}^4 \dot{c}}\left(1 + \frac{1}{(1-e^{-\dot{c}/2})^4}\right) \\
&\quad \times \ln\left(\frac{D_\star e^{D_\star(\beta+d+d\beta^{(1-\eta)/2}+d^2\beta^{-\eta}+1)}}{\widetilde{\delta}}\left(1 + \frac{1}{1-e^{-\dot{c}/2}}\right)\right).
\end{aligned} \tag{71}$$

$\square$

### D.2.5. NON-ASYMPTOTIC BOUND FOR THE EXPECTED EXCESS RISK

*Proof of Theorem 3.8.* We begin by decomposing the expected excess risk using the random variable $Y_\infty^{\text{fSGLD}}$, for which $\mathcal{L}(Y_\infty^{\text{fSGLD}}) = \pi_\beta^{\text{fSGLD}}$, and obtain

$$\begin{aligned}
&\mathbb{E}[g_\epsilon(\theta_k^{\text{fSGLD}})] - \inf_{\theta \in \mathbb{R}^d} g_\epsilon(\theta) \\
&= (\mathbb{E}[g_\epsilon(\theta_k^{\text{fSGLD}})] - \mathbb{E}[g_\epsilon(Y_\infty^{\text{fSGLD}})]) + (\mathbb{E}[g_\epsilon(Y_\infty^{\text{fSGLD}})] - \inf_{\theta \in \mathbb{R}^d} g_\epsilon(\theta)).
\end{aligned} \tag{72}$$

We proceed by controlling the two terms on the right-hand side of (72) separately. By using Lemma 6 of (Raginsky et al., 2017), Remark C.1 with $\sigma^2 = \beta^{-\frac{1+\eta}{2}}$ for $\eta \in (0,1)$, Lemma D.2, Lemma D.7, Corollary D.13, and Corollary D.15, the first term on the RHS of (72) can be bounded by

$$\begin{aligned}
&\mathbb{E}[g_\epsilon(\theta_k^{\text{fSGLD}})] - \mathbb{E}[g_\epsilon(Y_\infty^{\text{fSGLD}})] \\
&\leq \left(L_1\mathbb{E}[\varphi(X_0)](\mathbb{E}[|\theta_0|^2] + \alpha_1(\lambda_{\max} + a^{-1})) + L_1\mathbb{E}[\varphi(X_0)]\sigma\sqrt{d} + L_2\mathbb{E}[\bar{\varphi}(X_0)] + |\nabla U(0,0)|\right) \\
&\quad \times W_2(\mathcal{L}(\theta_k^{\text{fSGLD}}), \pi_\beta^{\text{fSGLD}}) \\
&\leq D_1^\Diamond e^{-\dot{c}\lambda k/4} + D_2^\Diamond \lambda^{1/4},
\end{aligned} \tag{73}$$

where

$$\begin{aligned}
D_1^\Diamond &:= D_4\left(L_1\mathbb{E}[\varphi(X_0)](\mathbb{E}[|\theta_0|^2] + \alpha_1(\lambda_{\max} + a^{-1})) + L_1\mathbb{E}[\varphi(X_0)]\sigma\sqrt{d} + L_2\mathbb{E}[\bar{\varphi}(X_0)] + |\nabla U(0,0)|\right) \\
&\quad \times (\mathbb{E}[|\theta_0|^4] + 1), \\
D_2^\Diamond &:= (D_5 + D_6) \\
&\quad \times \left(L_1\mathbb{E}[\varphi(X_0)](\mathbb{E}[|\theta_0|^2] + \alpha_1(\lambda_{\max} + a^{-1})) + L_1\mathbb{E}[\varphi(X_0)]\sigma\sqrt{d} + L_2\mathbb{E}[\bar{\varphi}(X_0)] + |\nabla U(0,0)|\right),
\end{aligned} \tag{74}$$

with $\dot{c}$ given in (53), $D_4, D_5, D_6$ given in (68), and $\alpha_1$ given in (43). The second term on the RHS of (72) can be controlled using Proposition 11 of (Raginsky et al., 2017) together with (52), which leads to

$$\mathbb{E}[g_\epsilon(Y_\infty^{\text{fSGLD}})] - \inf_{\theta \in \mathbb{R}^d} g_\epsilon(\theta) \leq D_\#^\Diamond, \tag{75}$$

where

$$D_\#^\Diamond := \frac{d}{2\beta}\log\left(\frac{eL_1\mathbb{E}[\varphi(X_0)]}{a}\left(\frac{b\beta}{d} + 1\right)\right), \tag{76}$$

with $b$ defined in (21). Using the estimates from (73) and (75) in (72), we obtain

$$\mathbb{E}[g_\epsilon(\theta_k^{\text{fSGLD}})] - \inf_{\theta \in \mathbb{R}^d} g_\epsilon(\theta) \le D_1^\Diamond e^{-\dot{c}\lambda k/4} + D_2^\Diamond \lambda^{1/4} + D_\#^\Diamond. \tag{77}$$

Applying (14) on the LHS of (77), along with (15), and choosing $\sigma^4 = \beta^{-(1+\eta)}$, it follows that

$$\mathbb{E}[v(\theta_k^{\text{fSGLD}})] - \inf_{\theta \in \mathbb{R}^d} v(\theta) \le D_1^\# e^{-\dot{c}\lambda k/4} + D_2^\# \lambda^{1/4} + D_3^\#, \tag{78}$$

where

$$
\begin{aligned}
D_1^\Diamond &= O\left( e^{D_\star(\beta+d+d\beta^{(1-\eta)/2}+d^2\beta^{-\eta}+1)} \left(1 + \frac{1}{1 - e^{-\dot{c}/2}}\right) \right), \\
D_2^\Diamond &= O\left( e^{D_\star(\beta+d+d\beta^{(1-\eta)/2}+d^2\beta^{-\eta}+1)} \left(1 + \frac{1}{1 - e^{-\dot{c}/2}}\right) \right), \\
D_3^\Diamond &:= D_\#^\Diamond + 2\beta^{-(1+\eta)}d^2(C_\mathcal{K} + C_\triangle) \\
&= O\left( \frac{d}{\beta} \log\left( D_\star\left(\frac{\beta}{d} + \beta^{(1-\eta)/2} + 1\right)\right) + \beta^{-(1+\eta)}d^2 \right),
\end{aligned} \tag{79}
$$

with $D_\star > 0$ a constant independent of $d$, $\beta$, $k$. In addition, for $\underline{\delta} > 0$, if we choose $\beta$ such that $D_3^\Diamond \le \underline{\delta}/3$, then choose $\lambda$ such that $\lambda \le \lambda_{\max}$ and $D_2^\Diamond \lambda^{1/4} \le \underline{\delta}/3$, and choose $k$ such that $D_1^\Diamond e^{-\dot{c}\lambda k/4} \le \underline{\delta}/3$, we obtain

$$\mathbb{E}[g_\epsilon(\theta_k^{\text{fSGLD}})] - \inf_{\theta \in \mathbb{R}^d} v(\theta) \le \underline{\delta}.$$

This yields

$$\beta \ge \beta_{\underline{\delta}} := \beta_\Diamond \vee \frac{6d}{\underline{\delta}} \log\left( \frac{eL_1\mathbb{E}[\varphi(X_0)]}{ad}(d+1)\right) \vee \left[\frac{12d^2(C_\mathcal{K} + C_\triangle)}{\underline{\delta}}\right]^{\frac{1}{1+\eta}}, \tag{80}$$

where $\beta_\Diamond$ is the root of the function $f^\Diamond(\beta) = \frac{\log(\beta+1)}{\beta} + \log\left(\frac{eL_1\mathbb{E}[\varphi(X_0)]}{ad}\left(1 + \frac{d}{\beta^{(1+\eta)/2}}\right)\right) - \frac{\underline{\delta}}{3d}$. Since

$$D_3^\Diamond \le \frac{d}{2\beta} \log\left( \frac{eL_1\mathbb{E}[\varphi(X_0)]}{ad}(b+1)(d+1)(\beta+1)\right) + \beta^{-(1+\eta)}d^2(C_\mathcal{K} + C_\triangle),$$

we can ensure $D_3^\Diamond \le \underline{\delta}/3$ by imposing

$$
\begin{aligned}
\frac{d}{2\beta}\log\left(\frac{eL_1\mathbb{E}[\varphi(X_0)]}{ad}(b+1)\right) &\le \frac{\delta}{12}, & \frac{d}{2\beta}\log\left(\frac{eL_1\mathbb{E}[\varphi(X_0)]}{ad}(d+1)\right) &\le \frac{\delta}{12}, \\
\frac{d}{2\beta}\log(\beta+1) &\le \frac{\delta}{12}, & \beta^{-(1+\eta)}d^2(C_\mathcal{K} + C_\triangle) &\le \frac{\delta}{12}.
\end{aligned}
$$

Moreover, one can verify that

$$\lambda \le \lambda_{\underline{\delta}} := \frac{\underline{\delta}^4}{81(D_2^\Diamond)^4} \wedge \lambda_{\max}, \tag{81}$$

and $\lambda k \ge \frac{4}{\dot{c}} \ln \frac{3D_1^\Diamond}{\underline{\delta}}$ which leads to

$$
\begin{aligned}
k \ge k_{\underline{\delta}} :={}& \frac{D_\star e^{D_\star(\beta+d+d\beta^{(1-\eta)/2}+d^2\beta^{-\eta}+1)}}{\underline{\delta}^4 \dot{c}} \left(1 + \frac{1}{(1 - e^{-\dot{c}/2})^4}\right) \\
&\times \ln\left( \frac{D_\star e^{D_\star(\beta+d+d\beta^{(1-\eta)/2}+d^2\beta^{-\eta}+1)}}{\underline{\delta}} \left(1 + \frac{1}{1 - e^{-\dot{c}/2}}\right)\right).
\end{aligned} \tag{82}
$$

$\square$

### D.2.6. TABLE OF CONSTANTS

Table 5 and Table 6 summarize the constants appearing in Theorem 3.5 and Theorem 3.8, together with their roles in the bounds and references to their explicit definitions in the main text.

*Table 5.* Summary of the constants appearing in Theorem 3.5. The table lists the role of each constant together with the equation or lemma in the main text where its explicit definition is provided.

| Constants | Role | Reference in the main text |
|---|---|---|
| $\dot{c}$ | Contraction constant of the overdamped Langevin diffusion (37) | Lemma D.9 |
| $D_1$ | Constant in the bound for the exponential mixing of the overdamped Langevin diffusion (37) | Eq. (61) |
| $D_2$ | Upper bound constant for $W_2(\mathcal{L}(\bar{\theta}_t^{\text{fSGLD}}), \mathcal{L}(\bar{\Phi}_t^{\lambda,k,\text{fSGLD}}))$ | Eq. (61) |
| $D_3$ | Upper bound constant for $W_1(\mathcal{L}(\bar{\Phi}_t^{\lambda,k,\text{fSGLD}}), \mathcal{L}(Y_t^{\lambda,\text{fSGLD}}))$ | Eq. (61) |
| $\underline{D}$ | Upper bound constant for $W_2(\pi_\beta^{\text{fSGLD}}, \pi_{\beta,\sigma}^\star)$ | Eq. (35) |
| $\beta_{\bar{\delta}}$ | Inverse temperature threshold ensuring $\underline{D} \leq \bar{\delta}/3$ | Eq. (63) |
| $\lambda_{\bar{\delta}}$ | Stepsize threshold ensuring $(D_2 + D_3)\sqrt{\lambda} \leq \bar{\delta}/3$ | Eq. (64) |
| $k_{\bar{\delta}}$ | Iteration threshold ensuring $D_1 e^{-\dot{c}\lambda k/2}(1 + \mathbb{E}[|\theta_0|^4]) \leq \bar{\delta}/3$ | Eq. (65) |

*Table 6.* Summary of the constants appearing in Theorem 3.8. The table lists the role of each constant together with the equation or lemma in the main text where its explicit definition is provided.

| Constants | Role | Reference in the main text |
|---|---|---|
| $\dot{c}$ | Contraction constant of the overdamped Langevin diffusion (37) | Lemma D.9 |
| $D_1^\diamond$ | Constant in the bound for the exponential mixing of the overdamped Langevin diffusion (37) | Eq. (74) |
| $D_2^\diamond$ | Upper bound constant for the discretization error contained in $\mathbb{E}[g_\epsilon(\theta_k^{\text{fSGLD}})] - \mathbb{E}[g_\epsilon(Y_\infty^{\text{fSGLD}})]$ | Eq. (74) |
| $D_3^\diamond$ | Upper bound constant for $\mathbb{E}[g_\epsilon(Y_\infty^{\text{fSGLD}})] - \inf_{\theta \in \mathbb{R}^d} v(\theta)$ | Eq. (79) |
| $\beta_{\underline{\delta}}$ | Inverse temperature threshold ensuring $D_3^\diamond \leq \underline{\delta}/3$ | Eq. (80) |
| $\lambda_{\underline{\delta}}$ | Stepsize threshold ensuring $D_2^\diamond \lambda^{1/4} \leq \underline{\delta}/3$ | Eq. (81) |
| $k_{\underline{\delta}}$ | Iteration threshold ensuring $D_1^\diamond e^{-\dot{c}\lambda k/4} \leq \underline{\delta}/3$ | Eq. (82) |

# E. Experimental details

## E.1. Software and hardware environments

We conduct all experiments with PYTHON 3.10.9 and PYTORCH 1.13.1, CUDA 11.6.2, NVIDIA Driver 510.10 on Ubuntu 22.04.1 LTS server which equipped with AMD Ryzen Threadripper PRO 5975WX, NVIDIA A100 GPUs.

## E.2. Details for Section 4.1 and 4.2

We follow the experimental protocol of Entropy-MCMC (Li & Zhang, 2024) for both image classification and out-of-distribution (OOD) detection. Here, Entropy-SGLD refers to the learning algorithm introduced in (Dziugaite & Roy, 2018).

**Datasets.** For in-distribution classification, we use CIFAR-10 and CIFAR-100. For OOD evaluation, we adopt SVHN as the out-of-distribution dataset. Following the evaluation setup of (Li & Zhang, 2024), the OOD test set is formed by concatenating the CIFAR test split with the SVHN test split.

**Models.** All experiments are conducted using a ResNet-18 backbone, aligned with the Entropy-MCMC configuration. We keep the backbone architecture, data preprocessing, and augmentation strategies identical across all compared methods to ensure controlled comparisons.

**Implementation details.** Consistent with the Bayesian marginalization procedure in (Li & Zhang, 2024), we construct the predictive distribution by averaging network outputs over multiple parameter snapshots. For a given input $x$, the predictive distribution is defined as

$$p(y \mid x) = \frac{1}{S} \sum_{s=1}^{S} p(y \mid x, \theta^{(s)}),$$

where $(\theta^{(s)})_{s=1}^{S}$ denotes the set of parameter snapshots, and we use $S = 16$ in all experiments. For fSGLD, we adopt the same learning rate schedule and temperature settings as used in the Entropy-MCMC implementation to maintain strict comparability. The noise scale is fixed to $\sigma = 10^{-3}$.

### E.3. Details for Section 4.3

**Datasets.** We evaluate our method on three noisy label datasets including CIFAR-10N and CIFAR-100N (Wei et al., 2022), and WebVision (Li et al., 2017). CIFAR-10N and CIFAR-100N include real-world annotation errors introduced by human annotators, offering realistic yet standardized benchmarks for noisy label learning. For CIFAR-10N, we use the aggregate noise setting. WebVision is a large-scale, in-the-wild benchmark, consisting of more than 2.4 million images with labels automatically collected from Google and Flickr based on the 1,000 ImageNet ILSVRC2012 categories. Following standard protocol (Li et al., 2020; Ortego et al., 2021; Li et al., 2022), we use the first 50 classes from its Google image subset and report Top-1 (WV-1) and Top-5 (WV-5) accuracy on the official validation set.

We follow standard data preprocessing and augmentation strategies as adopted in prior work (Li et al., 2017; Wei et al., 2022) on noisy-label benchmarks. For CIFAR-10N and CIFAR-100N, we apply random cropping with padding, random horizontal flipping, and normalization using dataset-specific statistics. For WebVision, we follow the preprocessing protocol of (Kodge, 2024). We note that Noisy-label benchmarks and ViT fine-tuning are widely used in evaluating the optimizer's generalization ability (Luo et al., 2024; Baek et al., 2024; Tan et al., 2025).

**Models.** We use ResNet-34 and ResNet-50 for training from scratch. For fine-tuning experiments, we use the pre-trained ViT-B/16 (Dosovitskiy et al., 2021) architecture, which has been trained on the ImageNet-1K (Deng et al., 2009) dataset as the backbone on CIFAR-10N and CIFAR-100N.

Regarding model architectures, we employ the CIFAR-specific variants of ResNet-34 and ResNet-50 when training on CIFAR-10N and CIFAR-100N, where the first convolution layer is replaced by a $3 \times 3$ kernel with stride 1 (instead of the $7 \times 7$ stride-2 convolution and max pooling used in ImageNet models) to accommodate the smaller $32 \times 32$ resolution. For WebVision, we adopt the standard ResNet implementations as provided for ImageNet-scale data.

**Baselines and Implementation Details.** We compare fSGLD against four baselines: SGD with momentum, AdamW (Loshchilov & Hutter, 2019), SAM (Foret et al., 2021), and ASAM (Kwon et al., 2021). To ensure a fair comparison, all optimizer hyperparameters are tuned using Optuna (Akiba et al., 2019) with 20 trials of Bayesian optimization. For each optimizer, the search spaces were carefully chosen to include previously reported optimal hyperparameters from the literature, ensuring that all baselines are strongly tuned. For experiments with training from scratch, all experiments are trained for 150 epochs with a batch size of 128. The learning rate decays by a factor of 0.1 in the 50th and 100th epochs. For fine tuning, models are trained for 75 epochs with a batch size of 128, decaying the rate by a factor of 0.1 at the 50th epoch. The detailed hyperparameter search spaces for each optimizer and experimental settings are provided in Table 7.

For both training-from-scratch and fine-tuning experiments, we use the same hyperparameter search spaces. Table 7 summarizes the ranges considered for each optimizer. We do not employ any early stopping or pruning strategy during the Optuna-based hyperparameter tuning, ensuring that each trial is fully evaluated to its final epoch. We performed the same number of hyper-parameter trials for all methods so that the search-space exploration budget (number of trials) was identical. Because each SAM and ASAM update requires two gradient evaluations, this design implies that, for the same number of trials and training epochs, SAM and ASAM consumed roughly twice the wall-clock compute time of the other baselines. Thus our tuning protocol is at least as favorable to SAM and ASAM as to the proposed fSGLD, ensuring that our reported improvements are not due to weaker tuning of SAM and ASAM.

For fSGLD, we search over inverse temperature, ranging from $\beta = 10^7$ to $\beta = 10^9$. We only search for the optimal inverse

temperature $\beta$ and then deterministically set perturbation scale $\sigma$ via our theoretically-derived relationship, $\sigma = \beta^{-\frac{1+\eta}{4}}$ with $\eta = 0.1$. This is a practical choice, as a larger $\eta$ would cause $\sigma$ to become too small, causing the perturbation to lose its intended significance. A small $\eta$ thus ensures stable optimization. This principled approach significantly simplifies the search space.

*Table 7.* Hyperparameter search spaces for different optimizers.

| Optimizer | Learning rate | Momentum | Weight decay | Other hyperparameters |
|---|---|---|---|---|
| SGD | $10^{[-2,0]}$ | $\{0.1, 0.9\}$ | $5 \times 10^{-4}$ | – |
| AdamW | $10^{[-4,-2]}$ | – | $10^{-2}$ | $[\beta_1, \beta_2] \in \{[0.8, 0.95], [0.99, 0.999]\}$ |
| SAM | $10^{[-2,0]}$ | $\{0.1, 0.9\}$ | $5 \times 10^{-4}$ | $\rho \in 10^{[-3,-1]}$ |
| ASAM | $10^{[-2,0]}$ | $\{0.1, 0.9\}$ | $5 \times 10^{-4}$ | $\rho \in 10^{[-3,-1]}$ |
| fSGLD | $10^{[-2,0]}$ | – | $5 \times 10^{-4}$ | $\beta \in 10^{[7,9]}$, $\sigma = \beta^{-\frac{1+0.1}{4}}$ |

### E.4. Details for Section 4.4

Figure 1 presents a sensitivity analysis of $\eta$ for a ResNet-34 trained on CIFAR-10N, with the initial learning rate fixed to $0.5$. In Figure 1a and Figure 1b, we fix $\beta = 10^8$ and $\sigma = 10^{-3}$, respectively. Each point corresponds to the mean classification accuracy over three random seeds, and error bars represent the associated standard deviations.

### E.5. Details for Section 4.5

For the Hessian spectrum analysis, we use the best-performing ResNet-34 model trained on CIFAR-10N under each optimizer setting. Given a trained network $f_\theta$ and loss function $L$, we compute Hessian-vector products (HVPs) by applying automatic differentiation to the scalar product $\nabla_\theta L^\top \bar{v}$ for a random vector $\bar{v}$. For eigenvalue computation, we adopt the Lanczos algorithm (Lin et al., 2016) as implemented in `scipy.sparse.linalg.eigsh`, which allows us to approximate the top-$k$ eigenvalues without explicitly forming the Hessian. In all reported results, we compute up to the top 50 eigenvalues. As a complementary measure of curvature, we estimate the trace of the Hessian using Hutchinson's stochastic estimator (Avron & Toledo, 2011) with Rademacher random vectors:

$$\operatorname{tr}(H(\theta)) \approx \frac{1}{m} \sum_{i=1}^{m} z_i^\top H(\theta) z_i, \quad z_i \sim \operatorname{Unif}\{\pm 1\}^d,$$

where $m = 100$ in our experiments and $d$ denotes the number of model parameters. Eigenvalue computations are performed with a tolerance of $10^{-4}$ and a maximum of 500 iterations for the Lanczos solver.

