# OpenReview forum: "Flatness-Aware Stochastic Gradient Langevin Dynamics"
_ICML.cc/2026/Conference — ICML 2026 regular_

### Official Review · Reviewer_AQav · 2026-03-11

**Soundness:** 4
**Presentation:** 4
**Significance:** 4
**Originality:** 3
**Overall Recommendation:** 5
**Confidence:** 2

**Summary:**

The paper introduces Flatness-Aware Stochastic Gradient Langevin Dynamics (fSGLD), a first-order optimization algorithm designed to improve generalization by biasing the search toward flat regions of the loss landscape. Existing methods like SAM exists, they focus on local flatness so that they often struggle to escape sharp local minima in multi-modal landscapes.

fSGLD addresses this by integrating Langevin dynamics with a modified objective function that accounts for the local curvature of the loss geometry. The paper provide a non-asymptotic theoretical analysis proving that fSGLD converges to a flatness-biased Gibbs distribution when a specific coupling between the perturbation scale and the inverse temperature is maintained.

Empirically, fSGLD converges to substantially flatter minima and outperforms standard optimisers and Bayesian baselines, while maintaining a computational cost comparable to that of SGD.

**Compliance With Llm Reviewing Policy:**

Affirmed.

**Key Questions For Authors:**

1. How does the non-asymptotic excess risk bound derived in Theorem 3.8 compare quantitatively with the classic SGLD bounds? Specifically, does the flatness-aware term lead to a tighter bound in high-dimensional settings, or is the improvement primarily qualitative?

**Limitations:**

Yes

**Strengths And Weaknesses:**

(1) Soundness: The paper is technically rigorous. The theoretical results provide explicit convergence rates and excess risk guarantees under reasonable assumptions. The experimental setup is comprehensive and covers diverse metrics.

(2) Presentation: The narrative is logical and well-structured. It was very easy to follow.

(3) Significance: Generalization and uncertainty estimation are central problems in machine learning. By providing a method that is as computationally efficient as SGLD but with better generalization properties, the paper offers high practicality.

(4) Originality: While the flat minima concept and SGLD are not new, the specific combination and the resulting flatness-biased Gibbs distribution provide novel perspective.

---

> ### Author Rebuttal · Authors · 2026-03-30
>
> Thank you very much for your time and effort to write the review. Below, we address the point raised in the  questions to authors.
>
> > **Q1.**
>
>
> We thank the reviewer for this important question.  The reviewer is right to focus on the quantitative form of the bound in Theorem 3.8:
> \begin{equation}
> \mathbb{E}[g_{\epsilon}(\theta_k^{\mathrm{fSGLD}})] - \inf_{\theta \in \mathbb{R}^d} v(\theta)
> \le D_1^{\Diamond} e^{- \dot{c} \lambda k/4} + D_2^{\Diamond} \lambda^{1/4} + D_3^{\Diamond}.
> \end{equation}
>
>
> Under the same dissipativity and Lipschitz assumptions as in Zhang et al. (2023) [1], the discretization rate $\lambda^{1/4}$ matches the best-known SGLD excess-risk rate, and the overall structure of the bound is analogous.
>
> The main difference lies in the objective:
>
> - Vanilla SGLD targets $\min u(\theta)$,
> - fSGLD targets $\min \big(u(\theta) + \frac{\sigma^2}{2}\mathrm{tr}(H(\theta))\big)$.
>
>
> Since the objectives are different, a direct comparison of the “tightness” of the bounds may not be meaningful. Instead, the key point is that fSGLD retains the optimal non-asymptotic rate with respect to the step size, while targeting a flatness-aware objective.
>
> Moreover, to target the Hessian-regularized objective, fSGLD introduces an additional error bound that does not appear in standard SGLD, namely the discrepancy between its invariant distribution and the flatness-biased target distribution. This term is captured by $\underline{D}$ in Theorem 3.5. Importantly, $\underline{D}$ depends polynomially on the dimension $d$, and involves terms of the form $d^2 \beta^{-(1+\eta)/2}$ , $d \beta^{-\eta/2}$, and $\beta^{-\eta/4} \sqrt{d}$, which decrease as $\beta$ grows. This ensures that the bound remains scalable in high‑dimensional settings.
>
> **References**
>
> [1] Ying Zhang, Omer Deniz Akyildiz, Theodoros Damoulas, and Sotirios Sabanis. Nonasymptotic estimates for stochastic gradient langevin dynamics under local conditions in nonconvex optimization. Applied Mathematics \& Optimization, 2023.

---

> > ### Author Rebuttal · Reviewer_AQav · 2026-04-03
> >
> > I thank the authors for their detailed answer to my question on comparison with classical SGLD bound.

---

> > > ### Author Response · Authors · 2026-04-07
> > >
> > > We thank the reviewer for the positive evaluation and important questions.

---

### Official Review · Reviewer_RDsx · 2026-03-13

**Soundness:** 3
**Presentation:** 4
**Significance:** 3
**Originality:** 3
**Overall Recommendation:** 5
**Confidence:** 3

**Summary:**

This paper proposes a variant of SGLD, fSGLD, that finds flatter minima of SGLD.

**Compliance With Llm Reviewing Policy:**

Affirmed.

**Final Justification:**

This paper provides solid contributions to flatness-aware optimization with sampling-based methods. The proposed method is elegant and fundamentally different from methods in optimization methods, and its effectiveness is supported by theory and experiments. Finally, the paper is presented in good form, with adequate discussions and justifications. Therefore, I would strongly recommend acceptance.

**Key Questions For Authors:**

1. What do the three terms in $\underline{D}$ mean intuitively?
2. In Figure 1, the performance of fSGLD seems to drop significantly out of the theoretical coupling regime. Why?
3. In Figure 2, SGLD already seems to finding flatter minima than SGD. Is there an intuition?

**Limitations:**

Yes.

**Strengths And Weaknesses:**

## Strengths

1. Discussions about theoretical results are comprehensive (except for Weakness 1).
2. fSGLD is an effective algorithms in practice, and theoretical results of $\beta$-$\sigma$ coupling are well verified by experiments.

## Weaknesses

1. Some discussions are expected about fSGLD convergeing to $\pi^\ast_{\beta, \sigma}$ in the comparative rate as SGLD converging to $\pi^\ast_\beta$.
2. It would be better to change Figure 2 to a bar chart with the eigenvalues binned.

---

> ### Author Rebuttal · Authors · 2026-03-30
>
> Thank you very much for your time and effort to write the review. Below, we address each point raised in the weaknesses and questions to authors. The cited reference is provided at the end.
>
> **Response to Weaknesses**
>
> **W1.** In Remark 3.6 (lines 223–234), we have mentioned the convergence rate of fSGLD in Wasserstein‑1 matches the best‑known rate for SGLD under comparable Lipschitz and dissipative assumptions. The $W_1$-rate is $\lambda^{1/2}$, which is the optimal rate for SGLD in this setting (see, e.g., Theorem 2.4 in Zhang et al, 2023 [1]).
>
> The error bound in Equation (11) is:
> $$W_1(\mathcal{L}(\theta_k), \pi_{\beta,\sigma}^{\star})
> \le D_1 e^{- \dot{c} \lambda k/2 } (1+ \mathbb{E}[|\theta_{0}|^4]) + (D_2 + D_3) \sqrt{\lambda} + \underline{D}.$$
> - The term $D_1 e^{- \dot{c} \lambda k/2 }$ captures exponential mixing of the overdamped Langevin diffusion, with rate $O(\lambda k)$, leading to a step-size exponent $1/2$ in the discretization term.
> - The term $(D_2 + D_3) \sqrt{\lambda}$ corresponds to the (Euler–Maruyama) discretization error, which is of order $O(\lambda^{1/2})$.
> - The term $\underline{D}$ bounds the distance between the invariant measure of fSGLD and the flatness-aware target, and depends on $\beta$, $d$, and $\eta$.
>
> Therefore, the convergence rate is governed by the discretization error $O(\lambda^{1/2})$, matching the best-known SGLD dependencies under our Assumptions 3.2–3.3 (see, e.g., Theorem 2.4 in [1]).
>
> The excess risk bound in Equation (12),
> \begin{equation*}
> \begin{split}
> \mathbb{E} [g_{\epsilon}(\theta_k^{\text{fSGLD}})] - \inf_{\theta \in \mathbb{R}^d} v(\theta)
> \le D_1^{\Diamond} e^{-  \dot{c} \lambda k/4} +  D_2^{\Diamond} \lambda^{1/4} +  D_3^{\Diamond},
> \end{split}
> \end{equation*}
> admits a similar interpretation, with the convergence rate governed by a discretization term of order $O(\lambda^{1/4})$, matching the best-known SGLD dependencies (see. e.g., Corollary 2.8 in [1]).
>
> We will make this interpretation more explicit in the revised manuscript and highlight that the exponential and $\sqrt{\lambda}$ (or $\lambda^{1/4}$) dependencies match those of Zhang et al. (2023) [1].
>
>  **W2.** Thank you for the suggestion. We will make this change in the revised version of the manuscript.
>
> **Response to Questions**
>
>  **Q1.** Thank you for the clarifying question. Intuitively, the three terms in $\underline{D}$ reflect different sources of error in approximating the flatness-biased target distribution. The first two terms, $O(\beta^{-\frac{\eta}{4}}\sqrt{d} + \beta^{-\frac{\eta}{2}} d)$, come from controlling the discrepancy between the invariant measure $\pi_{\beta}^{\mathrm{fSGLD}}$ and the flatness-biased target $\pi_{\beta,\sigma}^{\star}$, and capture how this gap scales with the temperature $\beta$ and the dimension $d$. The third term, $O(\beta^{-(1+\eta)/2} d^2)$, arises from the regularity of the target measure itself and reflects how the curvature and dissipativity (Assumption 3.3) influence the Wasserstein bound.
>
> **Q2.** Thank you for this question. Figure 1 shows that the prescribed $\beta$–$\sigma$ coupling is not merely a theoretical condition, but also the regime where the perturbation and Langevin noise are well balanced.
> When moving away from this regime, this balance breaks. For instance, as $\beta$ increases, $\sigma = \beta^{-(1+\eta)/4}$ decreases, weakening the flatness bias, while for smaller $\beta$ (larger $\sigma$), the Langevin noise dominates the gradient direction in each iteration, leading to overly noisy updates. Both cases result in performance degradation. This supports that the proposed $\beta$–$\sigma$ coupling is not only theoretically motivated, but also practically beneficial.
>
> **Q3.** Thank you for this insightful observation. Indeed, SGLD can be viewed as SGD with an additional scaled Gaussian perturbation, namely Langevin noise (i.e., $\sqrt{2 \beta^{-1}} \xi_{k+1}$ in eq. (5)), injected at each iteration. Due to this extra noise, SGLD exhibits stronger stochastic exploration than SGD.
>
> Intuitively, the injected Gaussian noise helps the iterates escape sharp and narrow minima, while flatter regions, which occupy a larger volume in the parameter space, are more likely to retain the dynamics. As a result, the algorithm tends to spend more time in low-curvature basins. This is consistent with the broader understanding that noisy gradient methods favor flatter minima compared to deterministic updates. In this light, the behavior observed in Figure 2 is expected. Our fSGLD further strengthens this effect via perturbed gradients and the $\beta$–$\sigma$ coupling, so that the algorithm directly targets a flatness-biased Gibbs measure rather than relying on the weaker flatness bias of vanilla SGLD.
>
> Reference
>
> [1] Zhang, et al Nonasymptotic estimates for stochastic gradient langevin dynamics under local conditions in nonconvex optimization. Applied Mathematics & Optimization 2023.

---

> > ### Author Rebuttal · Reviewer_RDsx · 2026-04-02
> >
> > Many thanks for the response. I have also read the other reviews and rebuttals, and I believe that the paper is a strong submission. I will increase my rating to 5.

---

> > > ### Author Response · Authors · 2026-04-07
> > >
> > > We thank the reviewer for the positive evaluation and interesting questions. We also appreciate the updated score after the rebuttal and are glad that our responses addressed the concerns.

---

### Official Review · Reviewer_WZNX · 2026-03-13

**Soundness:** 4
**Presentation:** 4
**Significance:** 3
**Originality:** 3
**Overall Recommendation:** 5
**Confidence:** 3

**Summary:**

The article considers a flatness-aware algorithm designed to converge to minima of a non-convex function $u$ that exhibit flatter curvature in their neighborhood. Such minima are attractive because they are believed to provide better generalization.

The problem is addressed by minimizing the penalized objective
$$
v = u + \lambda \operatorname{Tr}(H),
$$
where $H$ is the Hessian of $u$ and $\lambda$ is a regularization parameter.

The article proposes a first-order method that avoids the explicit computation of the Hessian, which is computationally expensive. Instead, it considers minimizing
$$
g(\theta) = \mathbb{E} (u(\theta + \varepsilon)) \simeq u(\theta)) + \sigma^2/2 \operatorname{Tr}(H(\theta)),
$$
where $\varepsilon \sim \mathcal{N}(0, \sigma I_d)$, and optimizes this objective using a stochastic gradient Langevin algorithm.

The authors then obtain theoretical guarantees and derive an optimal coupling between $\sigma$ and the inverse temperature parameter of the Gibbs target distribution.

Finally, a practical experimental section illustrates the performance of the proposed algorithm.

**Compliance With Llm Reviewing Policy:**

Affirmed.

**Final Justification:**

I believe this is a strong paper that is well written, introduces a novel method with improved practical performance, and provides solid theoretical results.

My only concern lies in the proofs, which I have not examined in detail.

Overall, I recommend acceptance.

**Key Questions For Authors:**

1: Theorem 3.8 seems to suggest that $\eta \simeq 1$ is optimal. Why did you choose $\eta = 0.1$ in the experiments?

2: In Eq. (12), it seems to me that bounding  $\mathbb{E}[v(\theta_k)] - \inf v $
would provide more information. Why did you choose to bound  $\mathbb{E}[g_\varepsilon(\theta_k)] - \inf v$  instead?

3: How do you compare your method to ASAM [2]?


[2]: Kwon, Jungmin, et al. "Asam: Adaptive sharpness-aware minimization for scale-invariant learning of deep neural networks." International conference on machine learning. PMLR, 2021.

**Limitations:**

yes

**Strengths And Weaknesses:**

**Strengths:**
- The paper is very well written and technically solid.
- The paper proposes a novel algorithm with both theoretical guarantees and empirical demonstrations of its performance on real applications.

**Weaknesses:**
- The algorithm solves a weaker form of flatness-aware minimization, which was first introduced as
$$
\min_{\theta} \max_{\|\varepsilon\|\leq \rho} u(\theta + \varepsilon) + \lambda \|\theta\|^2 .
$$

- Linking the sharpness-aware parameter $\sigma$ to the inverse temperature parameter $\beta$ limits the convergence of the algorithm, since $\beta$ cannot be increased during training to ensure convergence to the exact minimizer, as done in simulated annealing [1].

- fSGLD is not compared with more competitive methods such as ASAM [2].


[1]: Pelletier, Mariane. "Weak convergence rates for stochastic approximation with application to multiple targets and simulated annealing." Annals of Applied Probability (1998): 10-44.

[2]: Kwon, Jungmin, et al. "Asam: Adaptive sharpness-aware minimization for scale-invariant learning of deep neural networks." International conference on machine learning. PMLR, 2021.

---

> ### Author Rebuttal · Authors · 2026-03-31
>
> Thank you for your time and effort to write the review.
>
> **W1.** We would like to clarify that our method does not aim to approximate or relax the SAM objective, but instead targets a different notion of flatness: $v(\theta)=u(\theta)+\frac{\sigma^2}{2}\text{tr}(H(\theta)).$ Through randomized smoothing and the coupling scheme, fSGLD optimizes this curvature-regularized objective and its corresponding Gibbs measure, rather than solving a min–max problem. Importantly, these formulations are not strictly ordered, but reflect different inductive biases: SAM emphasizes worst-case robustness, while fSGLD promotes low average curvature. Moreover, by avoiding the min–max structure, fSGLD achieves this at the same computational cost as SGD/SGLD, without requiring double gradient evaluations. For these reasons, we would describe our method as addressing a complementary formulation of flatness-aware learning, rather than a weaker approximation of the SAM objective.
>
> **W2.** We thank the reviewer for this insightful comment. fSGLD is not designed as a simulated annealing scheme, nor does it aim to drive $\beta\to\infty$ to concentrate on the set of global minimizers. Instead, we fix $\beta$ and couple $\sigma$ to $\beta$ (via $\sigma=\beta^{-(1+\eta)/4}$), so that the algorithm targets a flatness-biased Gibbs distribution. While we do not consider an annealed regime, our result provides non-asymptotic guarantees for any finite $\beta$ and $\sigma$, and in particular characterizes convergence to the target distribution for sufficiently large $\beta$. Nevertheless, we agree this is an interesting future direction.
>
> **W3/Q3.** We added comparisons with ASAM. fSGLD matches or outperforms ASAM on in-distribution, OOD, and noisy-label benchmarks, while reducing runtime from 662.5s/epoch to 345.8s/epoch.
> | Method | C10 ACC | C10 NLL | C100 ACC | C100 NLL | C10-SVHN AUROC/AUPR | C100-SVHN AUROC/AUPR | R34 C10N/C100N | R50 C10N/C100N | ViT C10N/C100N | sec/epoch |
> |---|---:|---:|---:|---:|---:|---:|---:|---:|---:|---:|
> | SAM | 95.25 | 0.166 | 78.41 | 0.876 | 94.23 / 95.67 | 74.56 / 84.61 | 91.53 / 59.18 | 90.88 / 59.01 | **96.75** / 74.66 | 656.7 |
> | ASAM | 95.34 | 0.150 | 78.26 | 0.814 | 97.24 / 98.26 | 79.86 / 87.93 | **91.73** / 60.79 | **91.25** / 60.47 | 96.25 / 74.86 | 662.5 |
> | fSGLD | **95.73** | **0.144** | **78.53** | **0.810** | **98.91 / 99.44** | **80.52 / 88.01** | 91.37 / **61.51** | 90.86 / **61.26** | 96.45 / **75.67** | **345.8** |
>
> **Q1.** The reviewer is correct that, from the asymptotic perspective of Theorem 3.8, taking $\eta \backsimeq 1$ yields the fastest decay of the flatness‑related terms. However, this also makes the perturbation scale $\sigma = \beta^{-(1+\eta)/4}$ very small, effectively diminishing the flatness-inducing effect in practice. In the fixed‑temperature setting, the perturbation scale $\sigma$ influences the effective dissipativity and linear-growth constants of the  drift of the underlying overdamped Langevin dynamics (39), which enter the bounds through $D_1,D_2,D_3$ (Theorem 3.5) and $D_1^{\Diamond},D_2^{\Diamond},D_3^{\Diamond}$ (Theorem 3.8), and depend through the coupling $\sigma=\beta^{-(1+\eta)/4}$. As a result, the theory does not identify a single optimal $\eta$, but instead guarantees validity for any $\eta \in (0,1)$. In practice, this leads to a trade-off: larger $\eta$ improves asymptotic decay, while smaller $\eta$ keeps $\sigma$ sufficiently large to maintain a meaningful flatness bias. We therefore fix $\eta = 0.1$ to preserve the flatness effect while retaining acceptable decay as $\beta$ increases. Our empirical results show that fSGLD is relatively insensitive to the choice of $\eta$, with slightly better performance observed for smaller values of $\eta$ (See Figure 1). We will clarify this trade-off in the revised manuscript.
>
> **Q2.** Thank you for the excellent comment. This is indeed a point we carefully considered when formulating Theorem 3.8. There are two natural and valid options for the excess-risk bound:
> (i) $\mathbb{E}[g_\epsilon(\theta_k)] - \inf_{\theta}v(\theta)$ (current version)
> (ii) $\mathbb{E}[v(\theta_k)] - \inf_{\theta}v(\theta)$ (suggested version).
> The current formulation emphasizes that fSGLD operates on the randomized-smoothing surrogate $g_\epsilon$, and that under the prescribed coupling, this surrogate aligns with the Hessian-regularized objective $v$. In contrast, the suggested formulation more directly highlights that the iterates of fSGLD solve the Hessian-regularized objective $v$. Both formulations are valid and emphasize different aspects of the method. Since $g_\epsilon(\theta)=v(\theta)+\mathbb{E}[\mathcal R(\theta,\epsilon)]$ one can pass from the current bound to the suggested one by controlling the expected remainder term under Assumption 3.1. We agree that the suggested formulation may provide a more direct interpretation, and we would be happy to revise the statement accordingly to improve clarity.

---

> > ### Author Rebuttal · Reviewer_WZNX · 2026-04-03
> >
> > I thank the authors for their clear response and the new numerical experiments.

---

> > > ### Author Response · Authors · 2026-04-07
> > >
> > > We sincerely thank the reviewer for the positive evaluation and thoughtful comments.

---

### Official Review · Reviewer_Th4B · 2026-03-16

**Soundness:** 1
**Presentation:** 2
**Significance:** 1
**Originality:** 2
**Overall Recommendation:** 2
**Confidence:** 4

**Summary:**

The paper introduces Flatness-aware Stochastic Gradient Langevin Dynamics (fSGLD), a sampling-based method designed to favor flatter minima by incorporating a randomized smoothing approximation of the Hessian trace into the objective. The authors provide a theoretical analysis showing that the sampling distribution converges to a flatness-aware Gibbs distribution, along with non-asymptotic guarantees under certain assumptions. Empirical results on several datasets suggest improved generalization compared to standard SGLD and related optimization methods.

**Compliance With Llm Reviewing Policy:**

Affirmed.

**Final Justification:**

I thank the authors for their further reply, which reduces my previous concern regarding Assumption 3.1.

However, all the other main concerns remain, including

1) It is still unclear why the coupling between $\beta$ and $\sigma$ can guarantee the good performance of fSGLD. While I understand the goal of this paper is to reveal relationship between these two parameters, it is not justified why such coupling in the form of $\sigma = \beta^{-(1+\eta)/4}$ is principle. Such an issue is further exacerbated by a technical flaw in the main result, Theorem 3.5, to be explained below.

2) The presentation issues remain.

There is a technical flaw in the main result, Theorem 3.5, where the upper bound for $\underline{D}$ is at least $O(\beta^{-\eta/4} \sqrt{d})$ by Proposition 3.4. As a result, the bound in Theorem 3.5 for the Wasserstein-1 distance between the law of the fSGLD iterates and the target Gibbs measure, only converges to $0$ if $\beta \to \infty$, which in turn results in $\sigma \to 0$ so that there is no flatness effect in fSGLD (also mentioned by Reviewer WZNX). If the authors do not intend to make the RHS of the bound (11) converge to $0$, then claiming such a result as a convergence result is misleading. Also, it is not clear how the convergence rate of $O(\lambda^{1/2})$ is obtained in the authors' rebuttal so that the RHS of (11) is bounded by $O(\lambda^{1/2})$ thus "matching the best‑known SGLD dependencies". In fact, if the RHS of (11) is bounded by $O(\lambda^{1/2})$, then we must have $\beta^{-\eta/4} \sqrt{d} = O(\lambda)$, so there is a further coupling between $\beta$ and the step size $\lambda$.

In summary, due to the above concerns, including the concern about the factually flawed claim in the main result (Theorem 3.5) as a convergence result "matching the best‑known SGLD dependencies", I cannot recommend acceptance for this paper in its current form.

**Key Questions For Authors:**

See weaknesses

**Limitations:**

See weaknesses

**Strengths And Weaknesses:**

Strengths:

(1) The paper addresses an important topic by explicitly incorporating flatness into stochastic gradient-based sampling. Optimizing a Hessian-trace-regularized objective is a principled way to encourage solutions that may generalize better.

(2) The proposed fSGLD method is supported by theoretical analysis, which attempts to characterize its limiting distribution and provides non-asymptotic guarantees under the stated assumptions.

(3) The empirical results indicate consistent improvements over baseline methods such as standard SGLD, suggesting that the approach can be beneficial in practice.

Weaknesses:

(1) The theoretical guarantees rely on very strong assumptions. In particular, Assumption 3.1 requires a uniform bound on the expectation of the remainder term $\mathcal{R}$ that scales with $\sigma^4 d^2$, which is quite restrictive. The comparison to assumptions in prior work is not clearly justified, and the claim that such higher-order assumptions are standard in implicit regularization analysis appears misleading. For instance, assumptions such as third-order smoothness in prior work, e.g., Zhang et al., 2025, are significantly weaker than the condition imposed here.

(2) The convergence results are not presented in a sufficiently informative way. It is unclear how restrictive the constant $C_{\Delta}$ must be for the theoretical guarantees to hold, and the bounds in Theorems 3.5 and 3.8 do not explicitly reflect its influence. In addition, the expressions in Eq. (11) and Eq. (12) are difficult to interpret, as they do not clearly reveal the convergence rates. This makes it hard to assess the claim that the results match the best known guarantees for SGLD.

(3) The role of the smoothing parameter $\sigma$ in practical training is not well explained. Although $\sigma$ is coupled with $\beta$ in the formulation, it directly controls the degree of flatness in the objective. The paper does not provide guidance on how to select or tune $\sigma$, nor does it clarify what range of values leads to desirable generalization behavior.

(4) The presentation could be improved. Some equations, especially in the appendix, extend beyond page margins and are difficult to read. Several notations introduced in Section 3.3 are only defined later without sufficient explanation in the main text. The bounds in Eq. (11) and Eq. (12) are also not broken down or interpreted, which further reduces clarity.

---

> ### Author Rebuttal · Authors · 2026-03-31
>
> Thank you very much for your time and effort to write the review.
>
> **W1** Zhang et al '25 [5] analyzes convergence to approximate flat minima under the assumptions that the objective is a convex, globally $C^3$ objective with bounded third derivative, while we work in a non‑convex setting with a four-times differentiable objective and Assumption 3.2-3.3 (Lipschitz gradient, dissipativity) and use a localized (outside‑compact) control of the fourth‑order term induced by randomized smoothing (Assumption 3.1), tailored to the smoothing framework. The two are designed for different regimes and are not directly comparable. Higher‑order regularity (e.g., $C^3/C^4$ and Lipschitz higher derivatives) is standard in the flatness/implicit‑bias literature [5,6,7]; and all impose such assumptions to characterize flatness or second‑order behavior, and our Assumption 3.1 aligns with this practice while avoiding global third‑derivative boundedness.
>
> **W2** The convergence rates in Theorems 3.5 and 3.8 are fully explicit, though we agree that the discussion can be made more accessible. We clarify that $C_{\vartriangle}$ is a problem-dependent regularity constant of the objective $u$, and that it appears with polynomial (mild) dependence into the bounds in Theorem 3.5 and Theorem 3.8 via the explicit expressions of $\underline{D}$ and $D_3^{\Diamond}$ in Eq. (35) and Eq. (79), respectively. In the main text following standard practice in the SGLD literature [1,2,3,4], we highlight the key dependencies on dimension $d$ and inverse temperature $\beta$ using big-$O$ notation, while referring detailed expressions of other constants to the appendix. The bound in Equation (11) is $W_1(\mathcal{L}(\theta_k),\pi_{\beta,\sigma}^\star) \le D_1 e^{-\dot{c}\lambda k/2}(1+\mathbb{E}[|\theta_0|^4]) + (D_2+D_3)\sqrt{\lambda} + \underline{D}$
>
> The term $D_1e^{-\dot{c}\lambda k/2}$ captures exponential mixing of the overdamped Langevin diffusion, with rate $O(\lambda k)$  leading to a step-size exponent 1/2 in the discretization term;  the term $(D_2+D_3)\sqrt{\lambda}$ corresponds to the (Euler–Maruyama) discretization error, which is of order $O(\lambda^{1/2})$, and the term $\underline{D}$ bounds the distance between the invariant measure of fSGLD and the flatness-aware target, and depends on $\beta$, $d$, and $\eta$. Therefore, the convergence rate is governed by the discretization error $O(\lambda^{1/2})$, matching the best‑known SGLD dependencies under our Assumptions 3.2-3.3 (see, e.g., Theorem 2.4 in [3]).  Equation (12) has a discretization term $\lambda^{1/4}$, again matching SGLD dependencies (e.g., Corollary 2.8 in [3]). We will explicitly break down the rate interpretation in the revision.
>
> **W3** In practice, $\beta$ and $\sigma$ are coupled via $\sigma=\beta^{-(1+\eta)/4}$, so tuning $\beta$ uniquely determines $\sigma$. We tune $\beta$ in the standard SGLD range (e.g., $10^7–10^9$), where these algorithms are known to perform well (e.g., under fixed temperature in [8,9] and annealed temperature in [10]), and this implicitly controls $\sigma$. Figure 1 shows that performance is maximized for the theoretically prescribed regime $\eta\in(0,1)$, and we provide the coupling to remove the need for independent tuning of $\sigma$ while still enabling effective flatness control through $\beta$.
>
> **W4** We agree that presentation can be improved. We will reformat long equations in the appendix, including (34)–(35), so they fit within margins. The constants $\dot{c}, D_1,D_2,D_3,\underline{D},D_1^\Diamond,D_2^\Diamond,D_3^\Diamond$ are defined in Section 3.3 and detailed in the appendix; in the revision, we will add a short table listing each constant, its definition, and role in the bounds to make the notation and structure easier to cross‑check.
>
> Due to the length limit, we had to shorten W1–W2. Happy to discuss further during the discussion period.
>
> [1] Raginsky et al Non-convex learning via stochastic gradient langevin dynamics: a nonasymptotic analysis PMLR'17.
>
> [2] Chau et al On stochastic gradient langevin dynamics with dependent data streams: The fully nonconvex case SIMODS'21.
>
> [3] Zhang et al Nonasymptotic estimates for stochastic gradient langevin dynamics under local conditions in nonconvex optimization AMO'23.
>
> [4]  Xu et al Global Convergence of Langevin Dynamics Based Algorithms for Nonconvex Optimization NeurIPS'18.
>
> [5] Zhang et al Zeroth-order optimization finds flat minima NeurIPS'25.
>
> [6] Ahn et al How to escape sharp minima with random perturbations ICML'24.
>
> [7] Arora et al Understanding Gradient Descent on the Edge of Stability in Deep Learning ICML'22.
>
> [8]  Lim et al Polygonal Unadjusted Langevin Algorithms: Creating stable and efficient adaptive algorithms for neural networks JMLR'24.
>
> [9] Lovas et al Taming neural networks with TUSLA: Non-convex learning via adaptive stochastic gradient Langevin algorithms SIMODS'23.
>
> [10] Deng et al Non-convex Learning via Replica Exchange Stochastic Gradient MCMC ICML'20.

---

> > ### Author Rebuttal · Reviewer_Th4B · 2026-04-04
> >
> > Thank you for your detailed rebuttal. While some of my original concerns were addressed, there are still major concerns unsolved in the current paper with rebuttal.
> >
> > 1. The main results still require very restrictive assumptions, such as Assumption 3.1, compared to the literature. Even though the globally bounded third derivative in Zhang et al '25 [5]  is avoided, the current justifications for Assumption 3.1 is insufficient. In particular,  the role of the compact $K$ is unclear, and a comparison between condition (19) and the uniformly bounded third derivative in Zhang et al '25 [5]  is needed to understand this assumption.
> >
> > 2. It is well known in the literature that the strength of the flatness measure, $\sigma$, influences the performance, which is also shown in Fig. 1. The coupling between $\beta$ and $\sigma$ further complicates the performance of fSGLD, as there is no principled method and formal analysis provided to give a justified way of tuning $\beta$ to render a $\sigma$ with good performance. The tuning of $\beta$ would not only influence $\sigma$ (thus the generalization capability of the model), but also the optimization trajectory (as the Langevin noise scale). The current finding that $\eta \in (0,1)$ is a highly heuristic result on very limited choices of simple models (e.g., ResNet-18, ResNet-50) and small-scaled data (e.g., CIFAR10, CIFAR100), which does not universally guarantee that such a choice of $\eta$ can generalize to general models/data on large-scale setups.
> >
> > 3. The current presentation of the technical results and their organization are still below the bar of ICML. Even in the rebuttal, the authors mentioned wrongly formatted equations (not only (34)–(35)). There are many wrongly formatted equations and grammatical errors in the appendix. Readers have to refer to equations in the appendix frequently to verify the technical results, including the statements in Theorem 3.5 and 3.8, which are the key results of this paper. The definitions deferred to the appendix do not have clear physical meanings in the main paper, which make the parsing of the main paper difficult. For example, there is nearly no way for one to check the convergence result in the bound of Theorem 3.5, with so many definitions either deferred to the appendix or not explained clearly. In summary, a thorough revision is required before this paper can be accepted.

---

> > > ### Author Response · Authors · 2026-04-07
> > >
> > > **Q1** [5] gives convergence toward flat minima for zeroth‑order optimizers in a convex setting under global smoothness assumptions up to order 3, where the remainder in the Taylor expansion is controlled via globally bounded third derivatives. Both [5] and our analysis use a Taylor expansion of $g_\epsilon$, but the way the remainder is controlled differs: Eq (2) in [5] uses a Taylor expansion of $g_\epsilon(\theta)=u(\theta)+\frac{\sigma^2}{2}tr(H(\theta))+o(\sigma^2)$ with remainder controlled by a **globally bounded third derivative**. In contrast, we expand one order higher to exploit the vanishing third moment and fourth‑order moments of $\epsilon$: $g_\epsilon(\theta)=u(\theta)+\frac{\sigma^2}{2}tr(H(\theta))+\mathbb{E}[R(\theta,\epsilon)]=u(\theta)+\frac{\sigma^2}{2}tr(H(\theta))+O(\sigma^4)$ under a $C^4$ condition. This gives a different remainder control mechanism tailored to Gaussian smoothing, which is essential for quantifying the gap between the invariant measure and the flatness-aware measure. We do not claim that our assumptions are weaker; the two works operate under different regimes (convex vs nonconvex, $o(\sigma^2)$ vs $O(\sigma^4)$, global Lipschitz up order 3 vs order 1).
> > >
> > > **Role of Compact Set $K$** We believe the reviewer may be referring to condition (9) in Assumption 3.1, not (19). The compact set $K$ is used to separate the localized remainder control from the dissipativity region (Lemma D.1): outside $K$, Assumption 3.1 controls $\mathbb{E}[R(\theta,\epsilon)]$; dissipativity (Assumption 3.3) guarantees the existence of a compact set where the minimizers of $g_\epsilon$ lie, see Section 4 in Raginsky [1].
> > >
> > > **Q2** **Coupling \& Tuning** Our proposed coupling is not intended as a task-optimal tuning rule, but to ensure that fSGLD targets the Hessian-trace regularized objective. As the reviewer notes, $\beta$ affects both the Langevin noise scale and the flatness scale $\sigma$. Our contribution is to characterize a principled relationship between these two, which aligns the algorithm with the desired flatness-aware objective. In this sense, the coupling is theoretically grounded for its intended role, rather than heuristic. This coupling removes the need to tune $\sigma$ independently, so that $\beta$ remains the primary parameter, as in standard SGLD.
> > >
> > > **Choice of $\eta$**
> > > We fix $\eta=0.1$ in our experiments and do not tune it. While tuning $\eta$ could further improve performance, we do not believe that (as mentioned by the reviewer) a universally optimal choice of $\eta$ that guarantees generalization performance across all problems is attainable. Our focus is on validating the theoretical range of $\eta$ and demonstrating that performance is robust and relatively insensitive to its choice within this range. Figure 1 shows that performance degrades outside this proposed regime, supporting the validity of $\eta\in(0,1)$. Although we observe that performance is relatively insensitive to the choice of $\eta$ within this range, we do not claim that a single value universally guarantees performance, as such guarantees are generally unrealistic in high-dimensional nonconvex settings. In practice, if one wishes to tune $\eta$, it should be done within the theoretically prescribed range $(0,1)$. To further address the reviewer’s concern, we additionally experimented on a larger-scale model-dataset (ViT-B/16 on WebVision), where we observe consistent behavior (Table 1 $\beta=10^8$, Table 2 $\sigma=10^{-3}$):
> > >
> > > |$\sigma$|$10^{1}$|$10^{-2}$|$10^{-2.2}$|$10^{-3}$|$10^{-4}$|$10^{-6}$
> > > |---|---:|---:|---:|---:|---:|---:|
> > > |$\eta$|$-1$|$0$|$0.1$|$0.5$|$1$|$2$|
> > > |Acc(\%)|$2.00$|$86.76$|$87.04$|$86.32$|$86.04$|$86.04$|
> > >
> > > |$\beta$|$10^{-12}$|$10^{12}$|$10^{10.9}$|$10^{8}$|$10^{6}$|$10^{4}$|
> > > |---|---:|---:|---:|---:|---:|---:|
> > > |$\eta$|$-2$|$0$|$0.1$|$0.5$|$1$|$2$|
> > > |Acc(\%)|$2.00$|$85.00$|$86.2$|$86.32$|$86.44$|$24.00$|
> > >
> > > The role of $\eta$ is **explicitly characterized in our theory**: Proposition 3.4 shows how the discrepancy term (via $\underline{D}$) depends on $\eta$, quantifying how $g_\epsilon$ approaches the flatness-aware target. This reveals a clear trade-off: larger $\eta$ improves the approximation $g_\epsilon \approx v$, while inducing a smaller $\sigma$, potentially weakening the flatness effect.
> > >
> > > **Q3** The overall structure of our presentation is standard in non‑asymptotic SGLD analyses, where the main text states the key scalings in $d,\beta,\eta$ and the full constant expressions are deferred to the appendix. For a wider audience, we will revise the work to make the structure of the bounds more explicit: we will add a short table summarizing all constants, their definitions, and roles in Theorem 3.5-3.8, and we will carefully reformat long equations in the appendix. We are happy to fix any grammatical issues; if the reviewer can indicate specific lines that they consider problematic, we will address them individually in the revision.

---

### Decision · Program_Chairs · 2026-04-30

**Decision:**

Accept (regular)

**Comment:**

The paper proposes fSGLD, a neat sampling-based optimizer that targets flat minima at the same cost as SGLD. Three out of four reviewers recommend acceptance, praising the novelty, solid theory, and strong empirical results across multiple benchmarks. The main concern from Reviewer Th4B is about whether the convergence guarantee and the flatness effect can truly coexist, plus some presentation issues. While this is a fair point, the overall contributions outweigh the weaknesses. I recommend acceptance.